# Evaluating Judges as Evaluators: The JETTS Benchmark of LLM-as-Judges as Test-Time Scaling Evaluators

**Yilun Zhou** [* 1]  **Austin Xu** [* 1]  **Peifeng Wang** [1]  **Caiming Xiong** [1]  **Shafiq Joty** [1]

https://github.com/SalesforceAIResearch/jetts-benchmark

## Abstract

Scaling test-time computation, or affording a generator large language model (LLM) extra compute during inference, typically employs the help of external non-generative evaluators (i.e., reward models). Concurrently, LLM-judges, models trained to generate evaluations and critiques (explanations) in natural language, are becoming increasingly popular in automatic evaluation. Despite judge empirical successes, their effectiveness as evaluators in test-time scaling settings is largely unknown. In this paper, we introduce the *Judge Evaluation for Test-Time Scaling* (JETTS) benchmark, which evaluates judge performance in three domains (math reasoning, code generation, and instruction following) under three task settings: response reranking, step-level beam search, and critique-based response refinement. We evaluate 10 different judge models (7B-70B parameters) for 8 different base generator models (6.7B-72B parameters). Our benchmark shows that while judges are competitive with outcome reward models in reranking, they are consistently worse than process reward models in beam search procedures. Furthermore, though unique to LLM-judges, their natural language critiques are currently ineffective in guiding the generator towards better responses.

## 1 Introduction

For the last several years, the rapid increase in various capabilities of large language models (LLMs) is mostly attributed to scaling – the corresponding increase in model and training data sizes (Kaplan et al., 2020; Hoffmann et al., 2022). Nonetheless, this effort has gradually saturated due to lack

*Equal contribution  [1]Salesforce AI Research. Correspondence to: Yilun Zhou <yilun.zhou@salesforce.com>, Austin Xu <austin.xu@salesforce.com>.

*Proceedings of the 42$^{nd}$ International Conference on Machine Learning*, Vancouver, Canada. PMLR 267, 2025. Copyright 2025 by the author(s).

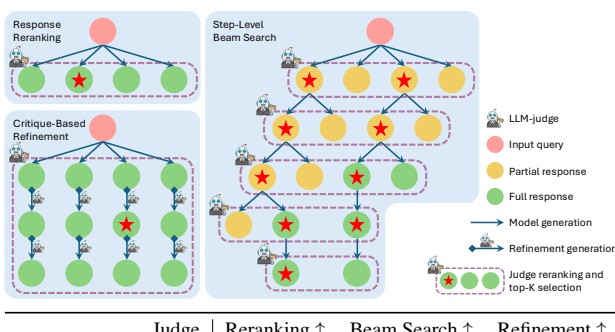

| Judge | Reranking ↑ | Beam Search ↑ | Refinement ↑ |
|---|---|---|---|
| Prometheus-2 7B | -0.107 | -0.100 | 0.976 |
| SFR-Judge 8B | 0.035 | 0.001 | 0.941 |
| Skywork-Critic 8B | 0.039 | 0.046 | - |
| OffsetBias 8B | -0.020 | 0.015 | - |
| Themis 8B | -0.149 | -0.021 | 0.996 |
| SFR-Judge 12B | 0.013 | 0.047 | 0.943 |
| Prometheus-2 8x7B | -0.079 | -0.083 | 0.966 |
| SFR-Judge 70B | 0.171 | 0.138 | 0.951 |
| Skywork-Critic 70B | 0.177 | 0.132 | - |
| Self-Taught Eval. 70B | 0.095 | 0.072 | - |
| Best RM | 0.113 | - | - |
| Best PRM | - | 0.195 | - |
| Random | -0.193 | -0.139 | - |
| Baseline Value | 0.000 | 0.000 | 1.000 |

Figure 1: Top: a graphical description of the three test-time compute tasks in JETTS used to evaluate LLM-judges. Bottom: the leaderboard of judges on the three benchmark tasks, along with outcome/process reward models and random choice. "-" means that the model cannot be used for the task. Achieving a value less than the baseline (grayed out) means that the judge is worse than some less compute-intensive approach (e.g., greedy decoding, see Sec. 3.5 for details).

of additional data and prohibitive cost of model training and serving. Recently, test-time scaling is being considered as an alternative solution, in which more compute is dedicated at the test time in the hope that better model responses are generated (Snell et al., 2024; Jaech et al., 2024).

A common component in test-time compute is the evaluator model, which gives a quality signal to various generated responses. Scalar reward models (RMs) are predominantly used to rerank either complete responses in best-of-N selection or partial responses in step-wise generation (e.g., beam search). By contrast, generative LLM judges are often favored in model evaluation, e.g., AlpacaEval (Dubois et al.,

2024b) and MT-Bench (Zheng et al., 2023), since they can adapt to custom evaluation criteria and can (often) produce natural language critiques (or explanations) of the response.

Despite LLM-judges achieving competitive performance on reward model benchmarks (e.g., RewardBench (Lambert et al., 2024)) and PPE (Frick et al., 2024)), they are less commonly used in test-time scaling scenarios compared to scalar RMs (Ji et al., 2025; Ke et al., 2025). This gap is perhaps surprising given the potential benefits of LLM-judges. Namely, judge models are often trained to engage in chain-of-thought reasoning before giving a judgment, which could not only benefit for reasoning-intensive domains such as math and code, but also serve as effective critiques for the model to refine its generation, such as in Reflexion (Shinn et al., 2024) or SCORE (Zhang et al., 2024b). Furthermore, LLM-judges could seamlessly integrate tool use. Last, LLM-judges are inherently "instructable", capable of giving different types of judgment like binary yes/no answers, numerical scores, fine-grained feedback, etc., simply with different prompts.

In this paper, we propose the first systematic benchmark of LLM-judges for model's test-time scaling. This benchmark consists of three tasks (Fig. 1): (1) response reranking, where the judge picks the best response out of several ones, (2) step-level beam search, where the judge guides the model to generate the response step-by-step, and (3) critique-based refinement, where the judge offers natural language critique to the response for the model to refine. We evaluate 10 different judge models, ranging in size from 7B to 70B parameters on these three test-time scaling tasks. We cover 3 different domains (math reasoning, code generation, and instruction following) and generate responses from up to 8 unique LLMs, ranging in size from 6.7B to 72B parameters.

The JETTS setup enables us to analyze the effects of judge-guided test-time scaling: Can a "weak" (e.g., 8B) judge actually help a "strong" (e.g., 72B) generator? How beneficial are strong judges to weak generators? In what domains are the current crop of judge models best suited for test-time scaling? How useful are judge critiques in practice? We find that weak judges can help strong generators in easier tasks, such as instruction following, but not in reasoning-intensive tasks like coding or math. Larger judges bring the most benefit for math and instruction following tasks, but no evaluated judges are able to reliably improve generator performance for coding. Lastly, while natural language critiques are touted as a defining advantage of judges over RMs, we find that such critiques have significant room for improvement in terms of utility. We present these results with many additional analyses in Sec. 4.

**Comparison with RewardBench.** In Fig. 2, we present a preview of our results by comparing performance on JETTS Best-of-N Reranking and Beam Search tasks against RewardBench (Lambert et al., 2024), a popular RM and judge

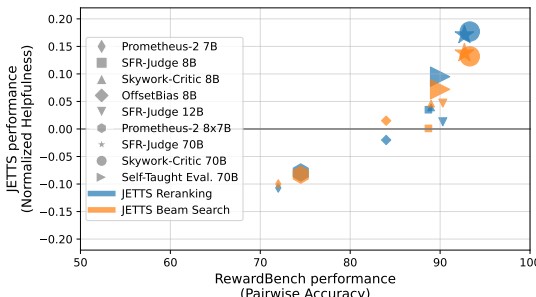

Figure 2: Judge performance on RewardBench vs. JETTS, where marker size signifies judge model size (7B-70B). Small judges perform comparably to large judges on RewardBench, but lag large judges on JETTS, which simulates test-time scaling settings.

model benchmark. While judge performance between the two benchmarks is generally correlated, JETTS reveals a difference in the "fundamental judging ability" between judges of different scales. For example, if deciding between Skywork-Critic-8B and 70B, the practical choice based on RewardBench is to use the 8B model, trading off 4% in accuracy for an order of magnitude fewer parameters. JETTS, which simulates test-time scaling scenarios, reveals that this choice is suboptimal for reranking: The 8B judge yields substantively lower improvements over the greedy response than the 70B judge. The benchmark construction approach likely contributes to the difficulty gap: RewardBench forms pairs of responses from different generators, meaning judges can arrive at the right outcome via stylistic factors, like formatting (Tan et al., 2024). JETTS, however, requires judges to compare responses sampled from the same generator, limiting stylistic factors that lead to correct outcomes.

## 2 Background and Related Work

The concept of using more computation at test time to increase model performance is not new. Chain-of-thought (CoT) reasoning (Nye et al., 2021; Kojima et al., 2022; Wei et al., 2022) can be considered as one of the earliest such techniques, where the model is encouraged to generate a reasoning chain before producing the final response. When the query asks for a "single-target" response (e.g., number, True/False, or name), self-consistency (Wang et al., 2022) samples multiple CoT chains and returns the majority vote answer. Orthogonally, tree-of-thoughts (Yao et al., 2024) and graph-of-thoughts (Besta et al., 2024) break down the chain into individual steps to search over.

While some studies (Madaan et al., 2024; Saunders et al., 2022) find that a model could improve or correct its own response, others highlight various issues and limitations in these approaches based on the fundamental paradox that if a model knows where it gets wrong, why it does not simply output the correct response in the beginning (Huang et al., 2023; Stechly et al., 2023; Valmeekam et al., 2023; Shridhar et al., 2024). Thus, a new trend in test-time compute is the

use of an evaluator to help with model response, commonly known as the reward model (RM) (Liu et al., 2025; Cobbe et al., 2021; Ouyang et al., 2022). Beyond RMs that operate on complete responses, known as outcome RMs (ORMs), for reasoning tasks it is also possible to train *process* RMs (PRMs) to provide step-level rewards (Wang et al., 2023a; Lightman et al., 2023; Zhang et al., 2025b; Luo et al., 2024; Dai et al., 2024).

Mostly parallel to this effort is the development of LLM-as-judges (Zheng et al., 2023), i.e., specializing LLMs into judges of model responses. While vanilla models are originally used (e.g., GPT-4 in AlpacaEval (Dubois et al., 2024b)), finetuned models are often found to deliver superior performance and be less susceptible to bias (Park et al., 2024). To further leverage the reasoning abilities of LLMs and improve explainability, recent judges are finetuned to generate reasoning or critiques for the response, often before the final judgment. Since then, a long line of work has been proposed in developing specialized LLM-judge models, focusing both on data curation (Wang et al., 2024b; Kim et al., 2023; 2024; Wang et al., 2023b; Li et al., 2023; Vu et al., 2024) and training methodologies (Wang et al., 2024a; Hu et al., 2024; Ye et al., 2024; Saad-Falcon et al., 2024; Deshpande et al., 2024; Wang et al., 2024b).

As RMs and LLM-judges are both evaluators in nature, they can often be directly compared, most notably on the Reward-Bench leaderboard (Lambert et al., 2024), showing close performance among the best RMs and LLM-judges. Due to saturating performance on RewardBench, with top models achieving over 95% accuracy, new benchmarks have been proposed to evaluate the efficacy of evaluators (Feuer et al., 2024; Liu et al., 2024b). Notably, ProcessBench (Zheng et al., 2024) was proposed to evaluate how process reward models (PRMs) perform in identifying step-level mistakes in math reasoning. PPE (Frick et al., 2024) and RMB (Zhou et al., 2024) both evaluate RM and LLM-judge efficacy in pairwise preference and Best-of-N settings for alignment, focusing on chat quality and safety. JudgeBench (Tan et al., 2024) identifies shortcomings with RewardBench's reasoning samples, and proposes a more difficult pairwise evaluation set for reasoning tasks. Complementing outcome-based judge benchmarks, Lan et al. (2024); Lin et al. (2024) evaluate the quality of critiques for single-round refinement.

JETTS, unlike recent judge benchmarks, does not assess judges using fixed pairwise test sets. Instead, it simulates three distinct test-time scaling scenarios (reranking, beam search, and *multi-round* refinement) and evaluates the quality of LLM judges by measuring the improvement they bring to the generator model. Snell et al. (2024) orthogonally investigate the efficacy of different test-time-compute scaling approaches (Best-of-N reranking and step-by-step beam search) with a *fixed* RM. Similarly, Zhang et al. (2024a) in-

| Task | Dataset | Size | Evaluation metric |
|------|---------|------|-------------------|
| Math Reasoning | GSM8k | 1319 | Accuracy via Math-Verify |
| | MATH (Lvl 5) | 1324 | Accuracy via Math-Verify |
| | CHAMP | 270 | Accuracy via GPT-4o grading |
| Code Generation | HumanEval+ | 164 | Pass @ 1 |
| | MBPP+ | 378 | Pass @ 1 |
| | BigCodeBench | 1140 | Pass @ 1 |
| Instruction Following | AlpacaEval | 805 | Win rate vs. GPT-4 Turbo, with GPT-4 Turbo as judge |
| | IFEval | 541 | Prompt-level strict accuracy |

Table 1: List of datasets and associated metrics in JETTS.

vestigates different types of verifiers for best-of-N reranking, considering only one evaluation protocol (binary correctness) for one prompted LLM-judge baseline per domain.

## 3 The JETTS Benchmark

In this section, we present our proposed benchmark: Judge Evaluation for Test-Time Scaling (JETTS). As previewed in Sec. 1, JETTS consists of three tasks: response reranking, step-level beam search, and critique-based refinement.

### 3.1 Task and Dataset Selection

Judge evaluation has typically focused on judging *instruction following* as a proxy for chat quality (Zeng et al., 2023). However, most recent efforts in test-time scaling have focused on math and code (e.g., Snell et al. (2024); Brown et al. (2024)). To cover both typical judge and test-time scaling domains, we consider three types of tasks: math reasoning, code generation and instruction following; we list the datasets in Tab. 1. In order to minimize the effect of randomness, we pre-compute the model responses and release them as part of the benchmark wherever possible.

For math, we use GSM8k (Cobbe et al., 2021), MATH (Hendrycks et al., 2021) and CHAMP (Mao et al., 2024). For code, we use HumanEval+ (Liu et al., 2023a), MBPP+ (Liu et al., 2023a) and BigCodeBench (Zhuo et al., 2024). HumanEval+ and MBPP+ are further refined versions of HumanEval (Chen et al., 2021) and MBPP (Austin et al., 2021), respectively. For instruction following, we use AlpacaEval (Dubois et al., 2024b) and IFEval (Zhou et al., 2023), with the former being LLM-evaluated and latter being algorithmically evaluated. App. A contains additional details about evaluation protocols.

### 3.2 Task 1: Response Reranking

The most straightforward and used inference-time compute technique is response reranking, where the generator model samples multiple responses with stochastic decoding and the judge reranks them to select the best one. In our benchmark, recognizing that the greedy decoding often produces higher quality responses, for each input query, we generate the greedy response and 9 sampled responses with a temperature of 1.0 and top-$p$ with $p = 0.95$, for a total of 10 responses.

We ask LLM-judge to rerank and select the top response for evaluation. Since no current judge supports one-shot direct reranking of multiple responses, we consider two protocols: pairwise round-robin and single-instance rating. In the former, for the $N$ responses, we construct the $N(N-1)/2$ response pairs, query the judge on each pair, assign the winner a score of 1 and the loser a score of 0, and select the response with the highest score (random tie-breaking if necessary)[1]. In the latter, we ask the judge to generate a score (an integer between 1 and 5), and the select the response with the highest score. Since ties are much more common in the single-instance rating setting, we keep all tied responses and report min, average and max performance; e.g., if for a test sample, a judge rates 3 of 10 responses as the highest score, and those 3 responses have accuracy labels $(1, 0, 1)$, then we use accuracy values $(0, 2/3, 1)$ for this sample when aggregating (min, average, max) performance.

Since the base performance is very different across datasets and generators, we define and report the normalized helpfulness for the judge as

$$h = \frac{p_{\text{judge}} - p_{\text{greedy}}}{p_{\text{oracle}} - p_{\text{greedy}}}, \tag{1}$$

where $p_{\text{greedy}}$ is the average performance of the greedily decoded responses across the dataset, $p_{\text{oracle}}$ is the oracle judge performance over all response selections (e.g., selecting the correct response for a math problem whenever there is one), and $p_{\text{judge}}$ is the average performance of top responses reranked by the judge. A negative value indicates strict harm as $h = 0$ can be trivially achieved by selecting the greedy response. We also include random baseline (i.e., the expected helpfulness of a judge that randomly reranks the responses) to compare each judge against an "average" reranker. For GSM8k and MATH, we also compute the performance of the majority-vote answer for each problem[2].

### 3.3  Task 2: Step-Level Beam Search

Using an evaluator to control the flow of generation is increasingly popular. For example, when generating the response line-by-line to a coding problem, an evaluator could decide that newly generated line is bad and hence asks the generator to resample a new line. In this task, we generalize this procedure to beam search. Since this procedure requires the notion of steps in a response, we consider only math and code tasks, and define each step as a line of response from our observation that the generated math solutions often put each step on its own line and the fact that each line of code naturally represents a programmatic operation.

---

[1]To account for positional bias, we employ additional consistency checks which may lead to ties; See App. B.1 for details.

[2]Majority-vote aggregation is not feasible for CHAMP since GPT-4o directly grades natural language responses against the ground truth without extracting the "final answer".

In a $(N, M)$-beam search with beam width of $M$, the generator samples $N$ first steps, which are reranked by the judge to keep the top-$N/M$. Then, for each chosen step, the generator samples $M$ next steps, for a total of $N$ responses to rerank again. When a generation ends (due to the EOS token or length limit), its current step is not expanded, resulting in less than $N$ responses. This process repeats until all selected steps are finished, at which time a final reranking selects the top response. Similar to before, in each sampling, one response is greedily decoded and the rest are sampled with temperature of 1.0 and top-$p$ of 0.95. We use $(10, 2)$-beam search in the benchmark and limit the tree depth to 10.

Notably, the beam search requires LLM-judges to reason about *partial* responses, a use case not specifically designed for any of the current judge models. Nonetheless, since they are often trained with flexible judging criteria, we append the following to the existing ones in this task:

> Important note: the model response is generated with streaming, meaning that it may not be finished yet but we still want to get a sense of the partial progress so far. If this is the case, you should provide judgment and feedback on the currently generated part, focusing on its correctness and promise of leading to a correct final solution.

Finally, we consider a *lookahead* variant of beam search similar to Snell et al. (2024), where instead of sending partial responses to the judge, we complete them with greedy decoding to finish and send these "lookahead" versions (without adding the instruction above). To contextualize judge performances, we again compute the greedy, random and oracle performance and derive normalized helpfulness.

### 3.4  Task 3: Critique-Based Refinement

A unique feature of LLM-judges is their ability to generate natural language critiques, or explanations, for their judgments. The critique-based refinement task evaluates whether these critiques are helpful for the model to improve their response. Specifically, we define a $(N, M)$-refinement as the setup where we start with $N$ seed responses, and each response is iteratively refined $M$ times, for a total of $N(M + 1)$ responses, which is reranked at the end to selected the top response. Compared to reranking and beam search, in which the judge reweighs the output distribution, in refinement the judge directly modifies it with its critique.

Since most judges are finetuned on fixed judging templates, they may not be able to directly revise responses. We instead provide the generator model the judge rating and critique, and ask it to revise its response using the feedback with the prompt shown in Fig. 18 of App. A.3. In our benchmark, we consider $(1, 9)$-refinement, which yields 10 responses in total to make it directly comparable to the reranking task.

While reranking and beam search have oracle performance as a natural upper-bound to define normalized helpfulness,

refinement has no such oracle. Therefore, we use improvement ratio over reranking and greedy responses as measure of judge efficacy, defined as

$$\delta^{(\text{RR})} = p_{\text{judge}}^{(\text{Ref})}/p_{\text{judge}}^{(\text{RR})}, \quad \delta^{(\text{G})} = p_{\text{judge}}^{(\text{Ref})}/p_{\text{greedy}}, \quad (2)$$

where, for $(N, M)$-refinement, $p_{\text{judge}}^{(\text{Ref})}$ is the average performance of the refined-and-reranked response across the dataset, and $p_{\text{judge}}^{(\text{RR})}$ is that of the reranked response among $N(M+1)$ sampled ones (including one greedy) and $p_{\text{greedy}}$ is that of the greedy response. In our $(1, 9)$-refinement setup, the latter two quantities are directly taken from the results in Task 1 of reranking on 10 responses.

In order for refinement to be a meaningful test-time scaling procedure, we need it to outperform both reranking and greedy. So we define the effective improvement ratio as

$$\delta^{(\text{Eff})} = \min\left(\delta^{(\text{RR})}, \delta^{(\text{G})}\right). \quad (3)$$

### 3.5 The JETTS Leaderboard

Finally, we compute a leaderboard (Fig. 1) for three tasks. We use *normalized helpfulness* $h$ (Eq. 1) to measure judge performance on reranking and refinement, and *effective improvement ratio* $\delta^{(\text{Eff})}$ (Eq. 3) on refinement. Each judge's score is computed by averaging the task-specific metric over all datasets and generators. If a judge supports multiple protocols (e.g., both single rating and pairwise comparisons in reranking), we report the maximum aggregate performance among all protocols. For convenience, we include per-dataset and per-generator metrics for each of the three tasks for all evaluated judges in App. C.

**Practitioner note.** Beam search evaluation requires significantly more compute than reranking evaluation. While Task 3 reveals interesting findings, e.g., judges generalize to some degree to assessing partial responses, we find reranking and beam search performance to be correlated; see Fig. 2 and analysis in Sec. 4.3. Our recommendation for practitioners is to use reranking performance as a proxy for beam search performance when evaluating future judge models, unless working specifically on process-based judging.

## 4 Experimental Results

### 4.1 Experiment Setup

As introduced in Sec. 3.1, JETTS consists of eight datasets from three task categories: math reasoning, code generation and instruction following. Models listed in Tab. 2 (top) generate the responses. We use Llama 3.1 (Dubey et al., 2024) and Qwen 2.5 (Yang et al., 2024) models in all tasks. For math and code tasks, we also consider four domain-specific models and intentionally choose newer models,

| Generator Model | # Params | Generator Model | #Params |
|---|---|---|---|
| Llama-3.1-Instruct | 8, 70B | Qwen-2.5-Instruct | 32, 72B |
| Deepseek-Math-Instruct | 7B | Qwen-2.5-Math | 7B |
| Deepseek-Coder-Instruct | 6.7B | Qwen-2.5-Coder | 7B |

| Judge Model | | Abbr. | # Params | Supported Capability | | |
|---|---|---|---|---|---|---|
| | | | | Pairwise | Single | Critique |
| SFR-Judge | (Wang et al., 2024a) | SFR | 8, 12, 70B | ✓ | ✓ | ✓ |
| Skywork-Critic | (Shiwen et al., 2024) | SC | 8, 70B | ✓ | ✗ | ✗ |
| OffsetBias | (Park et al., 2024) | OB | 8B | ✓ | ✗ | ✗ |
| Prometheus | (Kim et al., 2024) | Prom | 7, 8x7B | ✓ | ✓ | ✓ |
| Self-Taught-Eval. | (Wang et al., 2024b) | STE | 70B | ✓ | ✗ | ✗ |
| Themis | (Hu et al., 2024) | Thm | 8B | ✗ | ✓ | ✓ |

Table 2: Top: the generator models to produce the model responses. The models on the first row are used in all tasks, and those on the second and third rows are used only in math and code tasks respectively. Bottom: The judge models used in our experiments.

Qwen math (Yang et al., 2024) and coder (Yang et al., 2024), and older models, DeepSeek math (Shao et al., 2024) and coder (Guo et al., 2024), to analyze how judge-based test time scaling improves models of different strengths.

Tab. 2 (bottom) lists the LLM-judge models that we benchmark. We additionally include several baseline models:

- the vanilla Llama-3.1-8B-Instruct model with SFR-Judge prompts for response reranking;
- three ORMs for response reranking: Llama-3-OffsetBias-RM-8B ($OB_{\text{RM}}$), Skywork-Reward-Llama-3.1-8B-v0.2 and Skywork-Reward-Gemma-2-27B-v0.2 ($S_{\text{RM}}$); and
- PRM Qwen2.5-Math-PRM-7B ($Q_{\text{PRM}}$) for beam search.

App. A.2 includes more details on judges and RMs.

For all results presented below, unless otherwise noted, we perform statistical analyses of reported quantity differing from the baseline value (0 for reranking and beam-search, and 1 for refinement) using a two-sided one-sample $t$-test and indicate the significance as "$\star$" for $p \leq 0.05$, "$\star\star$" for $p \leq 0.01$, "$\star\star\star$" for $p \leq 0.001$, and "n.s." or no marker for not significant ($p > 0.05$).

### 4.2 Response Reranking

As discussed in Sec. 3.2, since no judge supports direct reranking of $N$ responses, we use either pairwise round-robin comparison or single-instance rating and evaluate each judge on the protocol(s) it supports. Furthermore, we experiment with to judging prompts for single rating: Likert scale, where judges are not given fine-grained criteria and simply asked to rate on a 1-5 scale, and additive criteria (Yuan et al., 2024), which instructs the model incrementally increase score based based on given criteria, with a maximum score of 5. The full results are presented in Tab. 3-5 in App. B.1. Below, we highlight notable findings.

**Reranking helpfulness varies across protocols and datasets.** Fig. 3 plots normalized helpfulness per dataset averaged across judges and generators, along with the respective random performance. There are two notable trends.

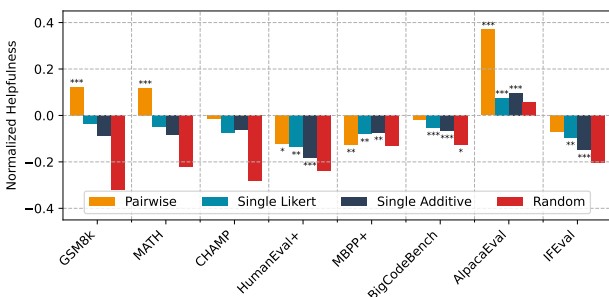

Figure 3: Normalized helpfulness of judge reranking per dataset. Asterisks on the bars denote statistical significance described at the end of Sec. 4.1.

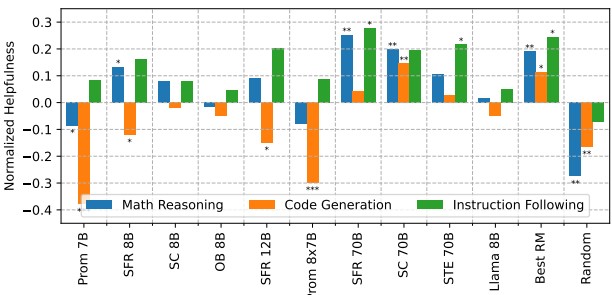

Figure 4: Normalized helpfulness of the pairwise protocol across task categories of each judge, compared to Llama-3.1 8B with judge prompt, best reward model and random reranking, averaged across generator models and datasets. A violin plot depicting finer-grained distributions is in Fig. 21 of App. B.1.

First, the pairwise protocol typically performs better than the single-rating protocol with either Likert or additive rubric, with the only exception of MBPP+, with all protocols worse than the greedy baseline. This is consistent with past findings, where both humans (Shah et al., 2016) and LLM-judges (Liu et al., 2023b) are better at providing pairwise comparison responses than single-rating responses. Given this finding, for the next two benchmarking tasks, step-level beam search and critique-based refinement, we use the pairwise protocol in reranking whenever possible. However, this comes at a significant inference cost, as the judge inference for pairwise reranking takes $O(N^2)$ time for $N$ responses to rerank, compared to $O(N)$ for single-instance rating.

Second, there are also significant variations across datasets, even within the same task category: pairwise protocols help on GSM8k and MATH but not CHAMP, and all protocols help on AlpacaEval (for both win rate and length-controlled win, as we show in App. B.1) but none on IFEval. Overall, judge helpfulness highly depends on the dataset, and further fine-grained analyses to understand the strengths and weaknesses of judgment would be valuable[3].

**Inter-task helpfulness variation is consistent across dif-**

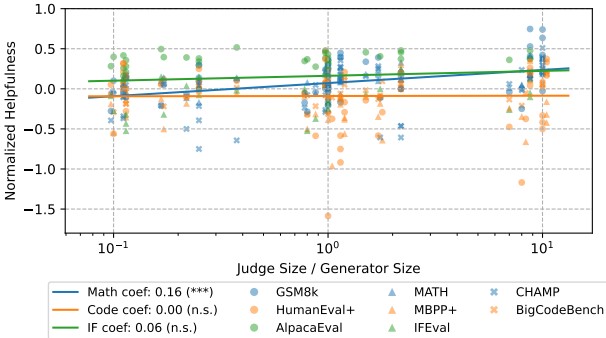

Figure 5: Normalized helpfulness under the pairwise protocol for different judge-to-generator size ratios, along with the best-fit lines for each task in log scale.

**ferent judges.** Fig. 4 plots the normalized helpfulness of each judge on each task category for the pairwise protocol, aggregating over respective datasets and all generator models. With few exceptions, all judges, including the vanilla Llama-3.1-8B Instruct with judge prompt and the reward models, demonstrate highest helpfulness for instruction following, less but mostly positive helpfulness for math reasoning, and mostly negative helpfulness for code generation. For single-rating protocols, consistent with Fig. 3, the helpfulness is much lower, but the relative ranking is still generally the same, as shown in Fig. 22 of App. B.1.

The consistency of inter-task helpfulness among judges coincides with the instruction-following training (and prompting) emphasis of current LLM-judges (Park et al., 2024; Tan et al., 2024). Despite training data including both math and code samples (e.g., Vu et al. (2024); Shiwen et al. (2024)), judge-specific finetuning seems to primarily boost instruction-following evaluation abilities, sometimes at the cost of other capabilities. This effect is shown in Fig. 4, where the vanilla Llama 3.1 8B model, prompted as a judge, exhibits a baseline level of competency as a judge, performing roughly at the greedy response level. Comparably sized 7B/8B judge models outperform it significantly on instruction-following and moderately on math, but performance degrades, sometimes significantly, on coding. Moreover, task-specific judge prompting does not appear to mitigate this task gap, as we show in Fig. 23 in App. B.1.

**Larger judge-to-generator size ratio under the pairwise protocol increases helpfulness for math and instruction following, but not code.** For each pair of judge and generator, Fig. 5 plots the ratio of the number of parameters in the judge to that in the generator (on log-scale) vs. the helpfulness achieved by the judge, color-coded by task category. A linear regression is plotted for each task on the log-scale size ratio. For math reasoning, a larger size ratio statistically significantly increases helpfulness, indicating the promise of larger judges for small generators, (which, nonetheless, may not be practical). At the lower end of

---

[3]Although different sets of judges are used to compute the average for pairwise vs. single-instance protocols due to different supported capabilities, Fig. 20 of App. B.1 shows that these trends largely preserve within judges that support both capabilities.

size ratio around 0.1 (e.g., 8B judge for 70B generator), normalized helpfulness is negative on average, indicating the absence of "weak-to-strong" guidance ability for LLM-judges. For instruction following, all size ratios achieve positive helpfulness but the effect of ratio increase is less notable. For code generation, the size ratio does not have an effect on the helpfulness, which is negative at all size ratios. For single-ratings protocol with Likert and additive rubrics (Fig. 24 of App. B.1), the effect of the size ratio is mildly positive only on math datasets, at 0.07 and 0.03 respectively, and negative on both of the other two tasks.

**No judge can reliably outperform majority-vote on math datasets.** The output format specification of GSM8k and MATH allows us to easily extract the final answer from the generated response, and compute the majority-vote (i.e., self-consistency (Wang et al., 2022)) performance. In Fig. 6, we plot the difference in the normalized helpfulness between each judge and this heuristic and mark all positive cells with purple stars. On GSM8k, only SFR 70B and SC 70B offer substantive improvement over majority-vote, but only for relatively weak generator models. For MATH, only SFR 70B is able to meaningfully beat majority-vote on small-to-medium generators only. Thus, in theory, an LLM that simply extracts the majority-vote answer could serve as a very strong reranker for these datasets.

**Judges are over-lenient under the single rating protocol, resulting in a large number of top-ranked responses.** Despite more time-efficient, single-rating evaluation protocol performance results in evaluation that is too *lenient*. Judges often rate a significant fraction of the $N$ responses a top score (i.e., a 4 or 5), with the most lenient models consistently rating all 10 responses equally high (Fig. 25 of App. B.1). Multiple responses tied as the "best" response leads to large ranges of possible reranking performance. Fig. 7 shows the average performance, as well as the mini-

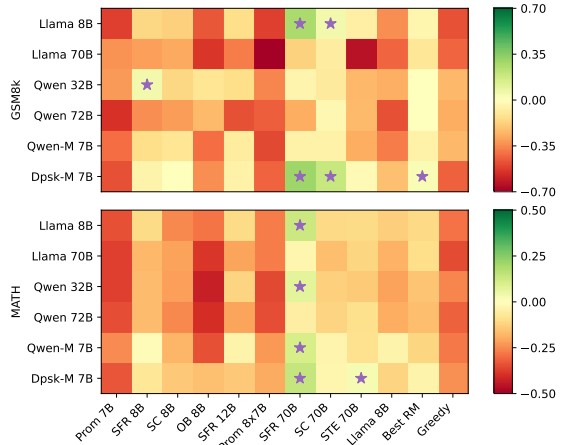

Figure 6: Normalized helpfulness difference between each judge and majority-vote for each generator on GSM8k and MATH. Positive value cells are marked with purple stars.

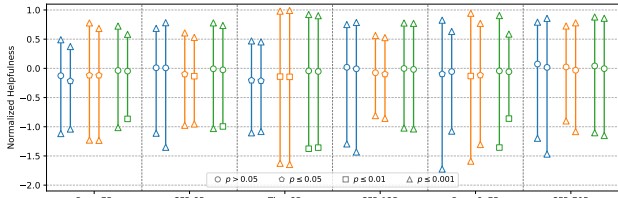

Figure 7: Normalized helpfulness ranges for single-rating protocols, showing minimum and maximum performance achievable from tied top-rated responses, along with average performance.

mum and maximum achievable performances from the tied responses. If one were to employ random tie-breaking, performance can swing from extremely poor to near oracle.

### 4.3 Step-Level Beam Search

As mentioned in Sec. 3.3, we focus on math and code tasks since their solutions can naturally be decomposed down into steps. We consider the (10, 2)-beam search, where the judge needs to rerank 10 candidates at each step and select the top-5, which get expanded to 10 next-step candidates. We use the pairwise protocol for the all rerankings, except for Themis 8B, a single instance rating judge, where we use the additive prompt. The full results are presented in Tab. 9 of App. B.2, and we highlight notable findings below.

**The effects of judge-to-generator size ratio are similar to that of reranking.** Fig. 8 shows the judge helpfulness vs. the ratio of the number of parameters in the judge to that in the generator (on log-scale), along with a linear regression for each task. The effects of judge-to-generator are remarkably similar to that in reranking: for math reasoning, a larger size ratio results in statistically significant helpfulness gains, with generally negative helpfulness for smaller size ratios. Like reranking, small judges are unable to provide "weak-to-strong" improvements for large generators, revealing the need for large judges for beam search. This requirement may be impractical due to the significant computational overhead required to judge outputs at each "step". For code, scaling judge size relative to the generator produces no statistically significant gains: for all size ratios, the helpfulness is negative.

**LLM-judges lag significantly behind the small $Q_{PRM}$.** Fig. 9 compares the normalized helpfulness of non-lookahead vs. lookahead beam search for different judges, along with reranking, random and $Q_{PRM}$ performance. For math reasoning, while some judges can achieve positive helpfulness, they struggle to outperform reranking even with much more judge inference. More notably, by far the best result is achieved by the 7B parameter $Q_{PRM}$. For code generation, only large judges achieve positive helpfulness, with only SC 70B providing statistically significant improvement. $Q_{PRM}$, although finetuned on math data, still achieves decent helpfulness at much smaller scale. These results suggest that even though some judges are finetuned

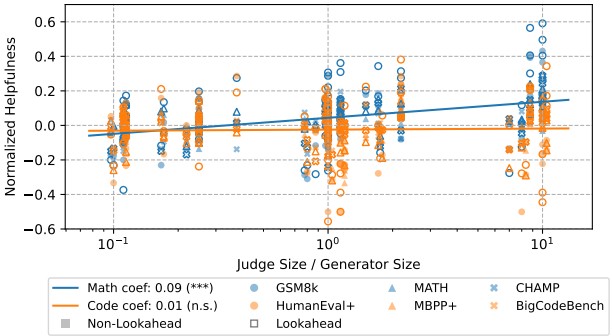

Figure 8: Normalized helpfulness of beam search under for different judge-to-generator size ratios, along with the best-fit lines for each task in log scale.

to work with flexible rubrics, process judgment is likely too out-of-distribution for small judges to be effective.

**The benefit of lookahead is inconclusive.** In the lookahead mode, rather than incomplete responses, judges evaluate full, rolled out responses, more aligned with judge training distributions. However, somewhat surprisingly, lookahead is not universally beneficial. For math reasoning, Fig. 9 shows that it benefits seven judges and $Q_{PRM}$, sometimes by two to three times, while slightly hurting the other three judges. For code generation, lookahead benefits five judges while hurting the other five and $Q_{PRM}$, with much milder effects. Overall, judges that perform well on reranking full responses tend to benefit from lookahead, whereas weaker ones tend to benefit from non-lookahead. This suggests that splitting full responses into shorter chunks makes evaluation *easier* for them, despite the out-of-distribution nature of step-level evaluation.

**There is a high correlation among non-lookahead, lookahead and reranking performance.** Fig. 10 plots the correlation between each judge's reranking performance and beam search performance (left), and the correlation between its lookahead and non-lookahead performance (right), with each scatter point representing a pair of dataset and generator. We observe high correlation among the three quantities, suggesting that they are all likely to be tied under the same underlying "fundamental judging capability" of the judge.

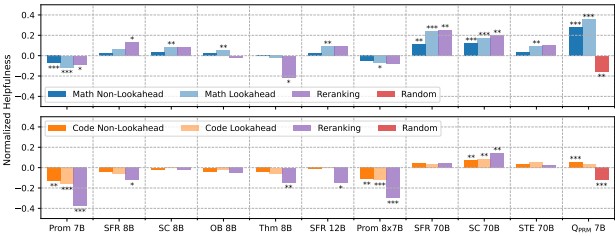

Figure 9: Normalized helpfulness of beam search by each judge and Qwen-2.5-Math-PRM 7B. The judge's reranking performance on the task category is plotted for comparison, as well as the random beam search baseline.

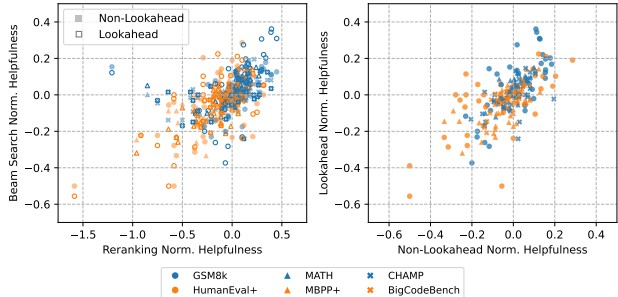

Figure 10: Relationship of judge's reranking vs. beam search performance, and non-lookahead vs. lookahead performance.

## 4.4 Critique-Based Refinement

Since this setup requires the judge to generate a critique on its judgment of an individual sample, we only evaluate judges with both single-instance rating and critique-generation capabilities in Tab. 2: SFR-Judge, Prometheus and Themis. In addition, when reranking the final candidate responses, as the reranking performance is much better with pairwise round-robin than single-instance rating (Sec. 4.2), we use the former whenver the judge supports it (i.e., for SFR-Judge and Prometheus) and fall back to the latter otherwise (i.e., for Themis). The full results are presented in Tab. 10 of App. B.3, with notable findings discussed below.

**No judge yields a beneficial performance on refinement across task category.** For individual judges, Fig. 11 plots the average effective improvement ratio $\delta^{(Eff)}$ for each judge, with no judge achieving a ratio better than 1.0. Instruction following experiences the largest loss, with up to 10% performance degradation. Thus, we conclude that all are incapable at this task.

We now focus our analysis on (1) quantifying the utility of refinement relative to greedy decoding and reranking and (2) precisely quantifying the behavior of the refine-then-rerank process. To do so, we define the following metrics. Towards the quantifying refinement utility, denote $p_{rand.}^{(Ref)}$ and $p_{orac.}^{(Ref)}$ to be the performance of a randomly selected response and the best performing response from all responses produced in the refinement process across the dataset, and analogously $p_{rand.}^{(RR)}$ and $p_{orac.}^{(RR)}$ on responses in reranking. We define $\delta_{rand.}^{(RR)} = p_{rand.}^{(Ref)}/p_{rand.}^{(RR)}$, $\delta_{orac.}^{(RR)} = p_{orac.}^{(Ref)}/p_{orac.}^{(RR)}$, $\delta_{rand.}^{(G)} = p_{rand.}^{(Ref)}/p_{greedy}$, $\delta_{orac.}^{(G)} = p_{orac.}^{(Ref)}/p_{greedy}$. To gain deeper insights into the refinement process, we examine the performance using the last refined response without rerank-

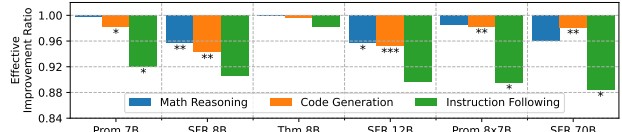

Figure 11: Effective improvement ratio $\delta^{(Eff)}$ for each judge and task category.

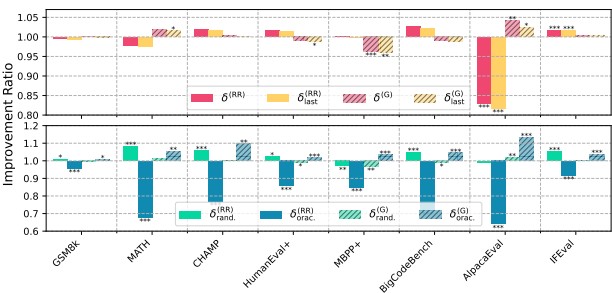

Figure 12: Various diagnostic metrics in refinement.

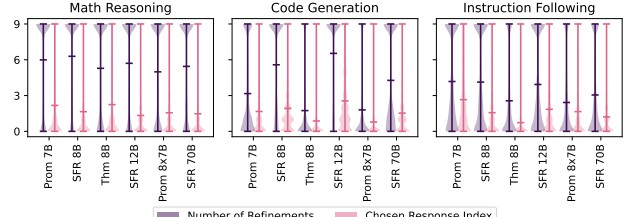

Figure 14: The distribution of number of refinement rounds conducted by the Llama 3.1 8B generator, and the index of the chosen response by the final reranking.

ing. We denote $p_{\text{last}}^{(\text{Ref})}$ to be the performance of the last response from the refinement process across the dataset and define $\delta_{\text{last}}^{(\text{RR})} = p_{\text{last}}^{(\text{Ref})}/p_{\text{judge}}^{(\text{RR})}$ and $\delta_{\text{last}}^{(\text{G})} = p_{\text{last}}^{(\text{Ref})}/p_{\text{greedy}}$ to be analogous to $\delta^{(\text{RR})}$ and $\delta^{(\text{G})}$ defined in Eq. 2. Fig. 26 in App. B.3 illustrates these newly defined $p$ metrics.

**The refinement process improves upon either reranking or greedy decoding, but never both.** In Fig. 12 (top), we see that $\delta^{(\text{RR})}$ and $\delta^{(\text{G})}$ are never meaningfully greater than 1.0 simultaneously on any dataset across all judges. As such, refinement is typically a sub-optimal choice of test-time scaling method, with greedy decoding or reranking leading to better performance.

**The benefit of final reranking is small.** While Fig. 12 (top) shows that $\delta$ is consistently better than $\delta_{\text{last}}$, the difference is small. Given the $O(N^2)$ complexity of pairwise reranking, just using the last refinement may be practically favorable.

**Generators and judges rarely engage in extensive refinement processes.** A notable trend from Fig. 12 (bottom) is $\delta_{\text{rand.}}^{(\text{G})} \approx 1$ and $\delta_{\text{orac.}} < 1$ for all datasets, suggesting that the refined responses are very similar to the original greedy seed and fail to discover much better solutions. It turns out, in fact, that the refinement process is rarely extensive. In Fig. 13, we plot the total number of generated refinements (excluding the original greedy response) across the three task categories. Among the four generator models, Llama 8B revised for significantly more rounds than other generators. This may suggest an intrinsic pliability to taking feedback, and/or result from its initial greedy response being much worse. Among judges, Prometheus 7B is much better

at giving actionable advice to encourage refinement behavior, which unfortunately does not translate to meaningful benefits in the final result (Fig. 11).

**The final selected response is often generated early on.** For Llama 8B, Fig. 14 plots the distribution of number of refinements (excluding the greedy seed response) and distribution of the chosen response index (0 for greedy). Llama 8B tends to either refine very minimally or use all nine rounds. However, the response actually chosen by the final reranking tends to be generated much earlier in the process, with the seed response (index 0) still most likely to be chosen, revealing limited utility of the critiques.

**Critiques tend to over-focus on stylistic features than response quality.** We conduct a qualitative study on the critiques in App. B.3. We inspect over 100 refinements and find two major issues. First, the judge gives high scores to responses with actual mistakes for stylistic reasons such as it being easy to follow (Fig. 27), thus failing to lead the generator to correct the error. Second, the judge is overly critical about minor stylistic issues on fully correct responses (Fig. 28) so that the generator engages in meaningless refinements that may even flip correct answers.

## 5 Conclusion and Future Work

In this paper, we seek to understand the feasibility of using LLM-judges in place of typically used RMs in test-time compute procedures. To this end, we propose JETTS, the first such systematic benchmark with three tasks: response reranking, step-level beam search and critique-based refinement. Our benchmarking results reveal several limitations and corresponding future research directions for LLM-judges. First, the reranking task faces a performance-efficiency dilemma: the pairwise protocol shows promise in certain domains but runs in $O(N^2)$, while the $O(N)$ single-instance protocol performs much worse. Improving single-rating ability of judges is essential for compute-efficient test-time scaling. Further, current chain-of-thought reasoning generated by the judges is insufficient, both for themselves to arrive at better judgments, especially in math and code domains, and for generators as critiques to refine responses. As such, a key next-step for judge development is to produce better and more useful reasoning.

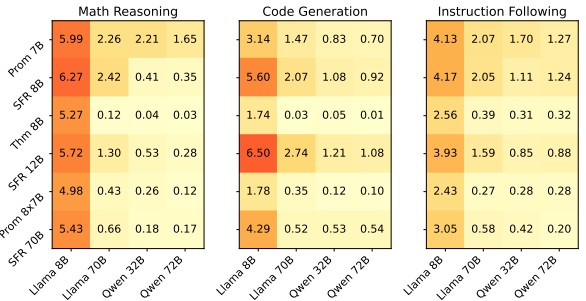

Figure 13: The average number of refinement rounds carried out by each (judge, generator) pair.

## Impact Statement

This paper presents work whose goal is to advance the field of Machine Learning. There are many potential societal consequences of our work, none which we feel must be specifically highlighted here.

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

# A Additional benchmark details: Evaluation procedures, judge models, and prompts

In this section, we provide additional details about benchmark evaluation procedures and the judge models evaluated.

## A.1 Benchmark evaluation procedures.

**Math reasoning.** For GSM8K, we prompt generator models with a 0-shot version of the CoT prompt used by Llama-3.1 evaluations[4]. To parse and grade results, we use Math-Verify (Kydlíček, 2025), comparing the model responses against the list of accepted responses provided by Llama-3.1 evaluations.

For MATH, we prompt the generator with 0-shot CoT prompt used in Shepherd-Math (Wang et al., 2023a). To parse and grade results, we again use Math-Verify, comparing model responses against the original solution in MATH.

For CHAMP, we utilize the default 0-shot prompt released by CHAMP and their suggested evaluation pipeline: The generator produces a response and is then asked to summarize its response into a final answer. A frontier LLM is then prompted to evaluate the final answer and the ground-truth. We utilize GPT-4o for this evaluation. We evaluate *all responses* generated across all tasks, regardless of if they are chosen or not, and store the results to be re-used for different judges. This ensures that judge evaluation is not influenced by potentially stochastic GPT-4o inference.

**Code generation.** We utilize the code generation and evaluation scripts provided by BigCodeBench (Zhuo et al., 2024) and EvalPlus (Liu et al., 2023a), with minor modifications to accommodate grading arbitrary subsets of the full evaluation set. For fast evaluation, similar to CHAMP evaluation, we evaluate all responses once and store the results.

**Instruction following.** We directly prompt the generator model with the prompts provided by both AlpacaEval (Liu et al., 2023a) and IFEval (Zhou et al., 2023) and utilize the evaluation scripts. We again making minimal changes to accommodate grading arbitrary subsets of the full evaluation set. For AlpacaEval, we follow the same procedure as CHAMP and evaluate all responses, storing results to be re-used for different judges. This again eliminates judge-by-judge stochasticity in GPT-4 evaluations.

## A.2 Additional details about evaluated judge baselines.

As noted in Tab. 2, we benchmark 10 judges of varying capabilities. Here, we provide a more comprehensive overview of each judge evaluated. We run inference with vLLM (Kwon et al., 2023), with 1 A100 40GB GPU for judges with sizes 12B and smaller, and 8xA100 40GB GPUs for judges with sizes 8x7B and larger.

- **SFR-Judge** (Wang et al., 2024a): SFR-Judge is a family of three judge models trained from Llama-3.1-8B-and 70B-instruct and Mistral-NeMo-12B-Instruct using direct preference optimization (DPO). These judges are trained with data samples from a diverse array of source datasets to provide natural language critiques and perform three evaluation tasks: Pairwise comparisons, single-rating evaluation, and binary yes/no classification. Notably, SFR-Judge training data preserves original human evaluation instructions, meaning the models are amenable to flexible prompting.

- **Skywork-Critic** (Shiwen et al., 2024): Skywork-Critic is a family of two judge models trained from Llama-3.1-8B-and 70B-instruct using supervised finetuning (SFT). These judges are trained on a smaller, curated preference dataset to perform pairwise comparison evaluation, without providing natural language critiques. Skywork-Critic models are trained with a fixed prompt template.

- **OffsetBias** (Park et al., 2024): OffsetBias is a pairwise comparison (without natural language critiques) model trained from Llama-3-8B-Instruct using SFT. It is trained with data specifically aimed at mitigating common LLM-as-judge biases, such as length bias. OffsetBias is trained with a fixed prompt template.

- **Prometheus-v2.0** (Kim et al., 2024): Prometheus-v2.0 is a family of two judge models trained from Mistral-7B-and 8x7B-instruct. These judges are trained on purely synthetic data synthesized from GPT-4 to perform pairwise comparisons and single-rating evaluation, and to provide natural language feedback. Notably, Prometheus specifically generates per-instance fine-grained evaluation criteria, making it amenable to flexible prompting.

---

[4]Meta releases evaluation data publicly. For example, Llama-3.1-8B-Instruct results: https://huggingface.co/datasets/meta-llama/Llama-3.1-8B-Instruct-evals

- **Self-taught-evaluator** (Wang et al., 2024b): Self-taught-evaluator is a model trained from Llama-3.1-70B-Instruct to perform pairwise comparison evaluation and to provide natural language critiques. It is trained using iterative DPO self-teaching, where adversarially generated samples from the model are produced at each training round, classified into correct or incorrect judgments, and used to update the model with DPO. Self-taught-evaluator is trained with a fixed prompt template.

- **Themis** (Hu et al., 2024): Themis is a single-rating specific model (with natural language critiques) trained from Llama-3-8B-Instruct using a single-rating margin-based version of DPO. It is trained from a large mixture of preference data from a diverse array of sources. Notably, original instructions given to human annotators is preserved in Themis training data, making it amenable to flexible prompting.

For each judge, we adhere to prompt templates and judgment parsing code provided by authors, minimally updating the evaluation criteria if necessary. We provide examples of prompts in the next section. We additionally evaluate four reward model baselines: three ORMs for reranking and one PRM for beam search.

- **Skywork-Reward-v0.2** (Liu et al., 2024a): A family of sequence classifier outcome reward models trained from Llama-3.1-8B and Gemma-2-27B using the same dataset as the Skywork-Critic series.

- **OffsetBias-RM** (Park et al., 2024): A sequence classifier outcome reward model trained from Llama-3-8B using the same dataset as the OffsetBias generative judge.

- **Qwen2.5-Math-PRM** (Zhang et al., 2025a): A process reward model specific for math domain trained from Qwen2.5.

### A.3 Prompts used in JETTS.

**Judge prompts.** Here, we provide an example of the pairwise (Fig. 15), Likert (Fig. 16), and additive rating (Fig. 17) prompts used for evaluation. We show the prompts used for SFR-Judge, with other prompts differing only in the template (e.g., the expected format of outputs, whether or not to elicit CoT, order of input elements, etc). In all of the prompts, as mentioned in Sec. 3.3, the message indicating a partial solution is inserted in the `{partial_response_note}` portion of the prompt, or is left empty (with surrounding newlines removed) for non-beam-search tasks.

**Generator refinement prompt.** We provide the prompt given to the generator for critique-based refinement in Fig. 18. As mentioned in Sec. 3.4, the generator is allowed to interpret the judge feedback and decide that no further refinement is necessary.

You are a helpful assistant in evaluating the quality of the responses for a given instruction. Your goal is to select the best response for the given instruction.

Select Response A or Response B, that is better for the given instruction. The two responses are generated by two different AI chatbots respectively.
Do NOT say both / neither are good.

Here are some rules of the evaluation:
(1) You should prioritize evaluating whether the output honestly/precisely/closely executes the instruction, then consider its helpfulness, accuracy, level of detail, harmlessness, etc.
(2) Responses should NOT contain more/less than what the instruction asks for, as such responses do NOT precisely execute the instruction.
(3) You should avoid any potential bias and your judgment should be as objective as possible. Here are some potential sources of bias:
- The order in which the responses were presented should NOT affect your judgment, as Response A and Response B are **equally likely** to be the better.
- The length of the responses should NOT affect your judgement, as a longer response does not necessarily correspond to a better response. When making your decision, evaluate if the response length is appropriate for the given instruction.

{partial_response_note}

Your reply should strictly follow this format:
**Reasoning:** <feedback evaluating the responses>

**Result:** <A or B>

Here is the data.

Instruction:
'''
{query_text}
'''

Response A:
'''
{response_A}
'''

Response B:
'''
{response_B}
'''

Figure 15: The pairwise evaluation prompt given to SFR-Judge models. Blue text are changed to the corresponding actual value when rendering the template. Everything else is presented literally. The partial response note slot holds the beam search judging note of Sec. 3.3 is only used in the non-lookahead mode of beam search. The partial response note slot holds the beam search judging note of Sec. 3.3 is only used in the non-lookahead mode of beam search.

You are tasked with evaluating a response based on a given instruction (which may contain an Input). Provide a comprehensive feedback on the response quality based on the rules for evaluation. Follow this with a score between 1 and 5. Avoid generating any additional opening, closing, or explanations.

Here are some rules of the evaluation:
(1) You should prioritize evaluating whether the output honestly/precisely/closely executes the instruction, then consider its helpfulness, accuracy, level of detail, harmlessness, etc.
(2) Responses should NOT contain more/less than what the instruction asks for, as such responses do NOT precisely execute the instruction.
(3) You should avoid any potential bias and your judgment should be as objective as possible. Here is a potential source of bias:
- The length of the response should NOT affect your judgement, as a longer response does not necessarily correspond to a better response. When making your decision, evaluate if the response length is appropriate for the given instruction.

{partial_response_note}

Your reply should strictly follow this format:
**Reasoning:** <Your feedback>

**Result:** <an integer between 1 and 5>

Here is the data:

Instruction:
```
{query_text}
```

Response:
```
{response}
```

Figure 16: The Likert single-rating evaluation prompt given to SFR-Judge models. Blue text are changed to the corresponding actual value when rendering the template. Everything else is presented literally.

You are tasked with evaluating a response based on a given instruction (which may contain an Input). Provide a comprehensive feedback on the response quality based on the rules for evaluation. Follow this with a score between 1 and 5. Avoid generating any additional opening, closing, or explanations.

Here are some rules of the evaluation:
(1) You should prioritize evaluating whether the response satisfies the provided rubric. The basis of your score should depend exactly on the rubric.
(2) You should avoid any potential bias and your judgment should be as objective as possible. Here is a potential source of bias:
- The length of the response should NOT affect your judgement, as a longer response does not necessarily correspond to a better response. When making your decision, evaluate if the response length is appropriate for the given instruction.

{partial_response_note}

Your reply should strictly follow this format:
**Reasoning:** <Your feedback>

**Result:** <an integer between 1 and 5>

Here is the data:

Instruction:
```
{query_text}
```

Response:
```
{response_note}
```

Score rubrics:
- Add one point if the response is relevant and provides some information related to the user's inquiry, even if it is incomplete or contains some irrelevant content.
- Add a second point if the response addresses a substantial portion of the user's question, but does not completely resolve the query or provide a direct answer.
- Add a third point if the response answers the basic elements of the user's question in a useful way, regardless of whether it seems to have been written by an AI Assistant or if it has elements typically found in blogs or search results.
- Add a fourth point if the response is clearly written from an AI Assistant's perspective, addressing the user's question directly and comprehensively, and is well-organized and helpful, even if there is slight room for improvement in clarity, conciseness or focus.
- Add a fifth point for a response that is impeccably tailored to the user's question by an AI Assistant, without extraneous information, reflecting expert knowledge, and demonstrating a high-quality, engaging, and insightful answer.

Figure 17: The additive single-rating evaluation prompt given to SFR-Judge models, adapted from Yuan et al. (2024). Blue text are changed to the corresponding actual value when rendering the template. Everything else is presented literally. The partial response note slot holds the beam search judging note of Sec. 3.3 is only used in the non-lookahead mode of beam search.

You help revise a machine-generated response to a user query. Below, you will be provided with the user query and the machine-generated response. You will also be provided with the output of an evaluator model, which gives a score (max being {max_score}) and an explanation for the score.

You should revise and improve the current response, following the evaluator's recommendation. If the evaluator does not identify any area of improvement, you should output "No revision needed." Otherwise, you should output the revised response surrounded by the <revised_response> and </revised_response> tags. You do not need to output anything else.

<query>
{query}
</query>

<original_response>
{response}
</original_response>

<score>
{score} out of {max_score}.
</score>

<explanation>
{explanation}
</explanation>

Your revision (or "No revision needed."):

Figure 18: The refinement prompt given to the generator. Blue text are changed to the corresponding actual value when rendering the template. Everything else is presented literally.

You are a helpful assistant in evaluating the quality of the responses for a given instruction, which is a math problem. Your goal is to select the best response for the given instruction.
Select Response A or Response B, that is better for the given math problem. The two responses are generated by two different AI chatbots respectively.
Do NOT say both / neither are good.

Here are some rules of the evaluation:
(1) You should prioritize evaluating if the output arrives at the correct solution for the given math problem and if the logical reasoning to derive the solution is sound.
(2) If both responses arrive at the correct solution, choose the response that contains the better logical reasoning. For reasoning, prioritize correctness, then completeness and conciseness.
(3) Responses should NOT contain more/less than what the question asks for, as such responses do NOT precisely solve the problem.
(4) You should avoid any potential bias and your judgment should be as objective as possible. Here are some potential sources of bias:
- The order in which the responses were presented should NOT affect your judgment, as Response A and Response B are **equally likely** to be the better.
- The length of the responses should NOT affect your judgement, as a longer response does not necessarily correspond to a better response. When making your decision, evaluate if the response length is appropriate for the given instruction.

---

You are a helpful assistant in evaluating the quality of the responses for a given instruction, which is a coding problem. Your goal is to select the best response for the given instruction.
Select Response A or Response B, that is better for the given coding problem. The two responses are generated by two different AI chatbots respectively.
Do NOT say both / neither are good.

Here are some rules of the evaluation:
(1) You should prioritize evaluating if the output response code correctly implements the desired functionality in the instruction.
(2) If both responses correctly implement the desired functionality, choose the response that contains the better written code. Prioritize conciseness and readibility.
(3) Responses should NOT contain more/less than what the question asks for, as such responses do NOT precisely solve the problem.
(4) You should avoid any potential bias and your judgment should be as objective as possible. Here are some potential sources of bias:
- The order in which the responses were presented should NOT affect your judgment, as Response A and Response B are **equally likely** to be the better.
- The length of the responses should NOT affect your judgement, as a longer response does not necessarily correspond to a better response. When making your decision, evaluate if the response length is appropriate for the given instruction.

Figure 19: Domain-specific evaluation criteria given to SFR-Judge-8B for math reasoning (top) and code generation (bottom) evaluation.

# B    Complete benchmark results and additional experiments

In this section, we present the raw benchmark performance as well as additional visualizations for all judges and reward model baselines across the three tasks.

## B.1    Response reranking

**Pairwise round-robin reranking details.** Because judge models are prone to positional biases (Li et al., 2023), we employ a *consistency*-based pairwise evaluation setup. We form all $N(N-1)/2$ pairs of responses. Then, we run each pairwise comparison twice, swapping the order of responses in the second run. If the same response is chosen for both runs, then we assign the selected response 1 "point" and the rejected response 0 "points". If the judge is inconsistent, i.e., selects different a different response when the order changes, we consider this a tie and assign both response 0.5 "points". The round-robin tournament winner is selected based on the response that has the most points after all $N(N-1)/2$ comparisons are completed.

**Full experimental results.** We present the full raw benchmark performance for all judges, generators, and dataset combinations for the pairwise comparison protocol in Tab. 3, Likert rating protocol in Tab. 4, and additive rating protocol in Tab. 5. We also report the performance of Llama-3.1-8B-Instruct prompted with SFR-Judge prompts and three reward models to contextualize judge performance.

|  |  | Prom 7B | SFR 8B | SC 8B | OB 8B | SFR 12B | Prom 8x7B | SFR 70B | SC 70B | STE 70B | Llama 8B | Best RM | Greedy | Random | Oracle | Maj |
|---|---|---|---|---|---|---|---|---|---|---|---|---|---|---|---|---|
| GSM8k | Llama 8B | 85.22 | 89.08 | 88.86 | 85.82 | 89.31 | 85.22 | 93.71 | 91.36 | 90.07 | 87.11 | 90.37 | 85.67 | 82.08 | 96.44 | 90.83 |
| | Llama 70B | 95.83 | 95.91 | 95.98 | 95.22 | 95.68 | 94.69 | 96.29 | 96.59 | 94.92 | 95.53 | 96.51 | 95.53 | 95.11 | 98.48 | 96.82 |
| | Qwen 32B | 95.00 | 96.21 | 95.53 | 95.68 | 95.60 | 94.84 | 95.91 | 95.75 | 95.22 | 95.15 | 96.06 | 95.22 | 95.22 | 98.56 | 96.06 |
| | Qwen 72B | 95.07 | 95.53 | 95.60 | 95.75 | 95.22 | 95.30 | 95.68 | 96.21 | 95.75 | 95.22 | 96.29 | 95.68 | 95.52 | 97.88 | 96.29 |
| | Qwen-M 7B | 94.69 | 95.45 | 95.53 | 94.69 | 95.60 | 94.47 | 95.68 | 95.68 | 95.07 | 94.77 | 95.68 | 95.15 | 91.65 | 97.88 | 95.83 |
| | Dpsk-M 7B | 80.44 | 86.73 | 87.49 | 82.41 | 86.66 | 81.05 | 91.74 | 90.22 | 87.11 | 84.15 | 87.83 | 80.97 | 79.75 | 95.53 | 87.49 |
| MATH | Llama 8B | 23.19 | 29.98 | 25.76 | 24.85 | 29.98 | 25.15 | 36.71 | 29.83 | 29.98 | 29.15 | 29.83 | 24.70 | 21.98 | 53.47 | 33.08 |
| | Llama 70B | 43.35 | 48.11 | 47.21 | 42.98 | 47.28 | 45.77 | 51.81 | 48.41 | 49.55 | 47.81 | 50.04 | 43.81 | 42.07 | 68.35 | 52.49 |
| | Qwen 32B | 54.76 | 59.06 | 57.93 | 53.55 | 60.05 | 55.06 | 64.05 | 59.82 | 59.59 | 58.01 | 59.34 | 57.10 | 52.24 | 78.17 | 62.61 |
| | Qwen 72B | 62.31 | 65.86 | 64.20 | 61.25 | 65.03 | 62.31 | 68.20 | 66.84 | 67.37 | 65.33 | 66.01 | 62.99 | 61.37 | 82.78 | 69.26 |
| | Qwen-M 7B | 66.77 | 70.09 | 67.75 | 65.48 | 69.71 | 67.07 | 71.68 | 69.79 | 68.96 | 69.71 | 68.58 | 66.39 | 55.28 | 80.14 | 70.24 |
| | Dpsk-M 7B | 16.01 | 23.56 | 21.45 | 21.07 | 21.53 | 19.94 | 29.68 | 24.85 | 26.28 | 22.13 | 24.58 | 18.66 | 17.04 | 47.21 | 25.68 |
| CHAMP | Llama 8B | 30.74 | 30.37 | 30.74 | 31.11 | 34.07 | 31.11 | 34.07 | 32.96 | 33.33 | 30.37 | 31.85 | 29.26 | 27.30 | 60.00 | - |
| | Llama 70B | 45.93 | 51.48 | 50.74 | 48.52 | 48.89 | 45.56 | 48.15 | 52.22 | 47.78 | 48.52 | 52.22 | 47.41 | 46.07 | 71.48 | - |
| | Qwen 32B | 70.00 | 71.11 | 67.41 | 72.96 | 68.52 | 68.89 | 70.37 | 68.89 | 70.37 | 67.78 | 71.30 | 75.19 | 70.67 | 85.56 | - |
| | Qwen 72B | 65.93 | 66.30 | 70.00 | 66.67 | 68.52 | 68.15 | 67.41 | 70.37 | 67.78 | 68.52 | 68.70 | 71.48 | 67.11 | 85.56 | - |
| | Qwen-M 7B | 65.19 | 70.37 | 68.15 | 67.04 | 67.78 | 67.41 | 72.59 | 70.00 | 70.37 | 66.67 | 70.00 | 62.96 | 46.26 | 81.85 | - |
| | Dpsk-M 7B | 27.04 | 42.59 | 38.52 | 34.44 | 39.26 | 25.56 | 42.96 | 40.00 | 39.63 | 30.74 | 41.48 | 26.30 | 28.78 | 72.22 | - |
| HumanEval+ | Llama 8B | 53.66 | 58.54 | 64.02 | 61.59 | 60.98 | 55.49 | 67.68 | 66.46 | 64.02 | 61.59 | 67.07 | 63.35 | 56.70 | 79.88 | - |
| | Llama 70B | 67.07 | 71.34 | 78.05 | 75.61 | 77.43 | 70.73 | 75.61 | 79.27 | 77.44 | 75.61 | 78.66 | 75.61 | 73.23 | 90.85 | - |
| | Qwen 32B | 79.88 | 82.32 | 81.10 | 84.15 | 82.32 | 79.88 | 81.10 | 83.54 | 81.10 | 81.10 | 85.37 | 81.10 | 82.56 | 93.29 | - |
| | Qwen 72B | 82.93 | 84.76 | 85.98 | 85.98 | 82.93 | 81.71 | 86.59 | 87.80 | 82.93 | 82.32 | 87.80 | 82.32 | 82.13 | 93.90 | - |
| | Qwen-C 7B | 75.00 | 82.32 | 81.10 | 79.88 | 82.32 | 78.05 | 84.15 | 83.54 | 82.93 | 82.32 | 82.32 | 86.59 | 81.52 | 93.90 | - |
| | Dpsk-C 6.7B | 66.46 | 69.51 | 75.00 | 67.68 | 67.68 | 66.46 | 77.43 | 76.83 | 74.39 | 69.51 | 77.44 | 71.95 | 67.62 | 86.59 | - |
| MBPP+ | Llama 8B | 47.62 | 54.23 | 57.41 | 58.99 | 52.38 | 49.21 | 56.88 | 62.70 | 62.17 | 56.61 | 60.85 | 54.50 | 55.21 | 76.46 | - |
| | Llama 70B | 55.29 | 58.73 | 64.29 | 65.34 | 56.08 | 56.08 | 60.32 | 69.31 | 66.14 | 65.34 | 68.39 | 65.08 | 63.68 | 83.07 | - |
| | Qwen 32B | 73.81 | 75.93 | 75.93 | 76.46 | 74.87 | 73.81 | 75.66 | 78.04 | 75.40 | 75.13 | 76.06 | 75.40 | 75.45 | 84.13 | - |
| | Qwen 72B | 74.60 | 76.72 | 76.46 | 76.19 | 75.13 | 75.40 | 77.25 | 76.98 | 76.46 | 78.04 | 76.72 | 76.19 | 75.71 | 84.66 | - |
| | Qwen-C 7B | 66.40 | 69.05 | 70.11 | 70.11 | 66.93 | 66.93 | 72.49 | 75.13 | 73.28 | 71.43 | 72.22 | 71.69 | 66.51 | 85.45 | - |
| | Dpsk-C 6.7B | 53.44 | 58.73 | 61.11 | 61.38 | 57.67 | 57.41 | 60.58 | 64.02 | 61.90 | 65.87 | 65.87 | 66.14 | 62.12 | 79.37 | - |
| BigCodeBench | Llama 8B | 26.23 | 32.46 | 33.25 | 32.72 | 31.23 | 28.33 | 36.05 | 37.19 | 36.23 | 30.35 | 35.09 | 31.67 | 28.03 | 56.84 | - |
| | Llama 70B | 40.44 | 42.19 | 42.89 | 42.28 | 41.67 | 40.00 | 42.63 | 43.51 | 43.60 | 43.33 | 43.90 | 45.44 | 41.92 | 62.63 | - |
| | Qwen 32B | 43.51 | 46.23 | 45.26 | 44.91 | 45.79 | 43.86 | 47.72 | 46.84 | 46.67 | 45.18 | 46.23 | 45.53 | 44.66 | 65.18 | - |
| | Qwen 72B | 45.79 | 46.84 | 47.46 | 47.89 | 47.63 | 45.70 | 47.54 | 47.28 | 47.63 | 45.79 | 48.60 | 46.30 | 46.30 | 60.18 | - |
| | Qwen-C 7B | 36.32 | 40.26 | 39.74 | 41.23 | 39.91 | 38.33 | 43.33 | 43.86 | 42.63 | 40.18 | 42.37 | 42.37 | 37.89 | 61.58 | - |
| | Dpsk-C 6.7B | 32.28 | 36.14 | 38.42 | 37.54 | 37.19 | 33.60 | 40.18 | 40.09 | 39.30 | 36.84 | 38.99 | 34.91 | 32.49 | 58.95 | - |
| AlpacaEval | Llama 8B | 34.94 | 37.15 | 36.98 | 34.23 | 39.93 | 37.31 | 40.54 | 37.86 | 37.63 | 32.00 | 38.98 | 27.26 | 26.80 | 55.07 | - |
| | Llama 70B | 46.02 | 44.88 | 42.88 | 40.99 | 45.92 | 45.20 | 46.87 | 43.90 | 45.69 | 40.17 | 45.08 | 35.48 | 34.27 | 61.90 | - |
| | Qwen 32B | 44.18 | 43.82 | 42.70 | 41.36 | 47.78 | 44.59 | 45.97 | 43.66 | 46.70 | 41.84 | 45.17 | 32.92 | 35.80 | 61.69 | - |
| | Qwen 72B | 58.94 | 62.22 | 55.98 | 58.35 | 64.16 | 60.51 | 63.69 | 61.43 | 63.03 | 58.73 | 62.56 | 52.01 | 56.76 | 76.52 | - |
| IFEval | Llama 8B | 67.28 | 70.24 | 70.06 | 71.90 | 71.90 | 68.95 | 74.68 | 71.34 | 72.09 | 69.87 | 73.01 | 72.83 | 67.79 | 87.80 | - |
| | Llama 70B | 82.07 | 81.70 | 80.78 | 80.04 | 81.70 | 80.04 | 82.99 | 82.26 | 82.99 | 80.22 | 82.62 | 84.29 | 81.21 | 92.42 | - |
| | Qwen 32B | 80.60 | 81.70 | 80.41 | 79.48 | 80.04 | 80.22 | 82.62 | 82.62 | 82.26 | 79.85 | 82.38 | 80.22 | 79.44 | 90.02 | - |
| | Qwen 72B | 82.81 | 84.66 | 84.47 | 83.73 | 84.84 | 83.55 | 85.03 | 85.40 | 83.92 | 85.21 | 86.04 | 83.55 | 83.42 | 92.05 | - |

Table 3: Performance of the top response after LLM-judge pairwise comparison-based reranking, in the context of the vanilla Llama 3.1 8B model using the SFR-Judge pairwise comparison prompt, the best reward model (using single-instance score to rerank), and various baselines. "-M" and "-C" suffixes represent the math and code domain-specific models.

| | | Prom 7B | SFR 8B | Thm 8B | SFR 12B | Prom 8x7B | SFR 70B | Llama 8B | OB$_{RM}$ 8B | S$_{RM}$ 8B | S$_{RM}$ 27B | Greedy | Random | Oracle | Maj |
|---|---|---|---|---|---|---|---|---|---|---|---|---|---|---|---|
| GSM8k | Llama 8B | 84.32 | 86.86 | 81.92 | 86.16 | 83.95 | 88.33 | 84.70 | 90.03 | 88.78 | 90.37 | 85.67 | 82.08 | 96.44 | 90.83 |
| | Llama 70B | 95.80 | 95.77 | 95.38 | 95.60 | 95.42 | 95.47 | 95.59 | 95.87 | 96.06 | 96.51 | 95.53 | 95.11 | 98.48 | 96.82 |
| | Qwen 32B | 95.45 | 95.65 | 95.07 | 95.34 | 95.30 | 95.47 | 95.35 | 96.06 | 95.91 | 95.79 | 95.22 | 95.22 | 98.56 | 96.06 |
| | Qwen 72B | 95.85 | 95.55 | 95.41 | 95.58 | 95.56 | 95.62 | 95.70 | 95.83 | 96.29 | 96.13 | 95.68 | 95.52 | 97.88 | 96.29 |
| | Qwen-M 7B | 93.72 | 95.36 | 91.86 | 95.35 | 94.19 | 95.57 | 94.74 | 95.45 | 95.53 | 95.68 | 95.15 | 91.65 | 97.88 | 95.83 |
| | Dpsk-M 7B | 80.89 | 82.74 | 79.60 | 82.77 | 80.86 | 87.13 | 81.26 | 85.90 | 87.72 | 87.83 | 80.97 | 79.75 | 95.53 | 87.49 |
| MATH | Llama 8B | 24.33 | 26.69 | 21.30 | 26.35 | 24.65 | 29.42 | 24.34 | 29.25 | 28.44 | 29.83 | 24.70 | 21.98 | 53.47 | 33.08 |
| | Llama 70B | 45.05 | 45.67 | 41.59 | 46.23 | 44.71 | 46.15 | 45.14 | 46.98 | 47.73 | 50.04 | 43.81 | 42.07 | 68.35 | 52.49 |
| | Qwen 32B | 54.79 | 54.53 | 52.11 | 55.84 | 53.97 | 54.99 | 55.05 | 55.97 | 59.34 | 57.78 | 57.10 | 52.24 | 78.17 | 62.61 |
| | Qwen 72B | 62.73 | 63.61 | 61.60 | 63.16 | 62.06 | 63.78 | 62.84 | 64.09 | 66.01 | 65.94 | 62.99 | 61.37 | 82.78 | 69.26 |
| | Qwen-M 7B | 55.81 | 67.35 | 56.26 | 67.33 | 61.71 | 67.78 | 64.14 | 67.86 | 67.71 | 68.58 | 66.39 | 55.28 | 80.14 | 70.24 |
| | Dpsk-M 7B | 17.22 | 21.76 | 16.33 | 19.83 | 18.11 | 24.87 | 20.30 | 23.79 | 24.58 | 24.43 | 18.66 | 17.04 | 47.21 | 25.68 |
| CHAMP | Llama 8B | 27.13 | 24.92 | 28.41 | 30.22 | 29.74 | 31.57 | 30.94 | 31.85 | 30.00 | 30.74 | 29.26 | 27.30 | 60.00 | - |
| | Llama 70B | 47.02 | 50.30 | 46.99 | 50.90 | 47.95 | 48.94 | 47.67 | 52.22 | 50.74 | 50.56 | 47.41 | 46.07 | 71.48 | - |
| | Qwen 32B | 70.17 | 69.68 | 70.81 | 71.06 | 70.30 | 70.81 | 69.36 | 69.63 | 68.70 | 71.30 | 75.19 | 70.67 | 85.56 | - |
| | Qwen 72B | 64.95 | 68.09 | 67.42 | 67.40 | 66.43 | 66.99 | 68.50 | 68.15 | 68.70 | 67.41 | 71.48 | 67.11 | 85.56 | - |
| | Qwen-M 7B | 64.47 | 65.88 | 66.41 | 67.26 | 64.20 | 68.44 | 66.08 | 67.78 | 70.00 | 68.15 | 62.96 | 46.26 | 81.85 | - |
| | Dpsk-M 7B | 26.70 | 31.83 | 29.15 | 33.02 | 30.52 | 39.11 | 31.96 | 39.44 | 41.48 | 34.81 | 26.30 | 28.78 | 72.22 | - |
| Humaneval+ | Llama 8B | 58.61 | 61.54 | 57.03 | 61.47 | 57.49 | 64.87 | 57.55 | 60.37 | 64.33 | 67.07 | 63.35 | 56.70 | 79.88 | - |
| | Llama 70B | 74.76 | 72.88 | 73.63 | 73.17 | 73.37 | 74.11 | 73.84 | 77.44 | 78.66 | 77.44 | 75.61 | 73.23 | 90.85 | - |
| | Qwen 32B | 82.90 | 81.30 | 82.29 | 82.53 | 82.50 | 84.85 | 81.80 | 84.76 | 81.71 | 85.37 | 81.10 | 82.56 | 93.29 | - |
| | Qwen 72B | 82.01 | 85.51 | 82.17 | 84.51 | 82.10 | 84.61 | 81.87 | 84.15 | 84.76 | 87.80 | 82.32 | 82.13 | 93.90 | - |
| | Qwen-C 7B | 81.81 | 81.95 | 81.77 | 81.82 | 81.91 | 84.57 | 81.80 | 81.71 | 79.27 | 82.32 | 86.59 | 81.52 | 93.90 | - |
| | Dpsk-C 6.7B | 67.95 | 69.97 | 68.42 | 71.09 | 68.00 | 73.16 | 69.66 | 72.56 | 73.78 | 77.44 | 71.95 | 67.62 | 86.59 | - |
| MBPP+ | Llama 8B | 56.75 | 54.20 | 55.94 | 57.17 | 56.42 | 60.72 | 56.85 | 60.32 | 60.85 | 56.88 | 54.50 | 55.21 | 76.46 | - |
| | Llama 70B | 64.92 | 60.33 | 63.84 | 61.81 | 64.26 | 65.01 | 64.74 | 64.55 | 68.39 | 63.23 | 65.08 | 63.68 | 83.07 | - |
| | Qwen 32B | 74.94 | 75.42 | 75.77 | 75.40 | 75.41 | 75.59 | 75.84 | 75.13 | 76.06 | 75.40 | 75.40 | 75.45 | 84.13 | - |
| | Qwen 72B | 76.31 | 76.95 | 76.07 | 76.55 | 76.11 | 76.01 | 76.68 | 76.72 | 75.26 | 76.46 | 76.19 | 75.71 | 84.66 | - |
| | Qwen-C 7B | 67.94 | 67.73 | 67.17 | 69.28 | 67.74 | 70.95 | 69.33 | 71.43 | 72.22 | 71.56 | 71.69 | 66.51 | 85.45 | - |
| | Dpsk-C 6.7B | 62.26 | 62.73 | 62.64 | 62.12 | 62.48 | 61.90 | 63.54 | 63.62 | 65.87 | 58.20 | 66.14 | 62.12 | 79.37 | - |
| BigCodeBench | Llama 8B | 29.69 | 30.65 | 28.95 | 32.00 | 29.85 | 34.20 | 30.43 | 32.63 | 34.96 | 35.09 | 31.67 | 28.03 | 56.84 | - |
| | Llama 70B | 42.37 | 42.75 | 42.55 | 42.88 | 42.73 | 43.31 | 42.74 | 43.63 | 43.90 | 43.86 | 45.44 | 41.92 | 62.63 | - |
| | Qwen 32B | 45.01 | 45.61 | 44.70 | 45.44 | 44.93 | 45.80 | 45.65 | 46.05 | 46.07 | 46.23 | 45.53 | 44.66 | 65.18 | - |
| | Qwen 72B | 46.76 | 46.93 | 46.33 | 47.27 | 46.37 | 47.45 | 46.63 | 46.89 | 48.60 | 48.33 | 46.67 | 46.30 | 60.18 | - |
| | Qwen-C 7B | 38.60 | 39.95 | 38.11 | 40.36 | 38.54 | 42.53 | 39.49 | 41.05 | 42.37 | 42.06 | 42.37 | 37.89 | 61.58 | - |
| | Dpsk-C 6.7B | 34.22 | 34.80 | 33.07 | 35.66 | 33.97 | 38.65 | 35.31 | 36.84 | 38.99 | 38.86 | 34.91 | 32.49 | 58.95 | - |
| AlpacaEval | Llama 8B | 28.42 | 29.95 | 27.80 | 29.50 | 27.64 | 29.43 | 27.36 | 33.62 | 36.05 | 38.98 | 27.26 | 26.80 | 55.07 | - |
| | Llama 70B | 36.13 | 36.86 | 35.01 | 36.55 | 35.50 | 35.60 | 33.99 | 39.10 | 42.05 | 45.08 | 35.48 | 34.27 | 61.90 | - |
| | Qwen 32B | 37.08 | 37.08 | 36.40 | 37.21 | 36.17 | 36.54 | 35.98 | 39.29 | 42.42 | 45.17 | 32.92 | 35.80 | 61.69 | - |
| | Qwen 72B | 54.40 | 54.75 | 53.80 | 54.99 | 54.05 | 54.54 | 54.08 | 55.95 | 60.34 | 62.56 | 52.01 | 56.76 | 76.52 | - |
| IFEval | Llama 8B | 67.85 | 70.34 | 69.13 | 70.63 | 69.15 | 71.80 | 70.41 | 72.46 | 71.07 | 73.01 | 72.83 | 66.43 | 87.80 | - |
| | Llama 70B | 82.26 | 82.27 | 82.26 | 82.29 | 81.99 | 82.72 | 81.45 | 81.15 | 82.62 | 81.89 | 84.29 | 79.94 | 92.42 | - |
| | Qwen 32B | 80.40 | 80.19 | 80.30 | 79.71 | 79.89 | 82.64 | 80.95 | 82.38 | 80.41 | 81.42 | 80.22 | 77.38 | 90.02 | - |
| | Qwen 72B | 83.25 | 83.72 | 83.29 | 84.29 | 83.63 | 84.87 | 83.91 | 84.57 | 86.04 | 83.46 | 83.55 | 81.44 | 92.24 | - |

Table 4: Judge reranking performance under the Likert scale single-instance rating protocol.

| | | Prom 7B | SFR 8B | Thm 8B | SFR 12B | Prom 8x7B | SFR 70B | Llama 8B | OB$_{RM}$ 8B | S$_{RM}$ 8B | S$_{RM}$ 27B | Greedy | Random | Oracle | Maj |
|---|---|---|---|---|---|---|---|---|---|---|---|---|---|---|---|
| GSM8k | Llama 8B | 83.28 | 85.45 | 82.10 | 85.32 | 85.69 | 86.51 | 84.99 | 90.03 | 88.78 | 90.37 | 85.67 | 82.08 | 96.44 | 90.83 |
| | Llama 70B | 95.63 | 95.78 | 95.53 | 95.71 | 95.88 | 95.35 | 95.68 | 95.87 | 96.06 | 96.51 | 95.53 | 95.11 | 98.48 | 96.82 |
| | Qwen 32B | 95.26 | 95.47 | 94.97 | 95.32 | 95.39 | 95.27 | 95.49 | 96.06 | 95.91 | 95.79 | 95.22 | 95.22 | 98.56 | 96.06 |
| | Qwen 72B | 95.57 | 95.55 | 95.15 | 95.53 | 95.67 | 95.50 | 95.13 | 95.83 | 96.29 | 96.13 | 95.68 | 95.52 | 97.88 | 96.29 |
| | Qwen-M 7B | 91.61 | 95.25 | 91.85 | 95.14 | 93.78 | 95.34 | 95.05 | 95.45 | 95.53 | 95.68 | 95.15 | 91.65 | 97.88 | 95.83 |
| | Dpsk-M 7B | 79.94 | 83.42 | 79.97 | 81.21 | 82.41 | 85.32 | 81.35 | 85.90 | 87.72 | 87.83 | 80.97 | 79.75 | 95.53 | 87.49 |
| MATH | Llama 8B | 22.79 | 25.16 | 21.19 | 25.96 | 23.97 | 26.96 | 24.59 | 29.25 | 28.44 | 29.83 | 24.70 | 21.98 | 53.47 | 33.08 |
| | Llama 70B | 43.35 | 45.71 | 42.54 | 46.26 | 44.26 | 44.98 | 46.02 | 46.98 | 47.73 | 50.04 | 43.81 | 42.07 | 68.35 | 52.49 |
| | Qwen 32B | 54.11 | 54.49 | 51.81 | 55.16 | 55.09 | 53.55 | 54.49 | 55.97 | 59.34 | 57.78 | 57.10 | 52.24 | 78.17 | 62.61 |
| | Qwen 72B | 61.71 | 62.97 | 61.85 | 62.84 | 63.08 | 62.46 | 62.49 | 64.09 | 66.01 | 65.94 | 62.99 | 61.37 | 82.78 | 69.26 |
| | Qwen-M 7B | 50.09 | 67.07 | 54.68 | 67.42 | 61.01 | 67.41 | 63.83 | 67.86 | 67.71 | 68.58 | 66.39 | 55.28 | 80.14 | 70.24 |
| | Dpsk-M 7B | 18.35 | 20.05 | 16.46 | 18.45 | 18.76 | 22.45 | 18.64 | 23.79 | 24.58 | 24.43 | 18.66 | 17.04 | 47.21 | 25.68 |
| CHAMP | Llama 8B | 29.77 | 27.78 | 28.74 | 32.05 | 31.61 | 31.18 | 30.70 | 31.85 | 30.00 | 30.74 | 29.26 | 27.30 | 60.00 | - |
| | Llama 70B | 47.96 | 49.85 | 47.19 | 49.40 | 49.44 | 47.41 | 46.37 | 52.22 | 50.74 | 50.56 | 47.41 | 46.07 | 71.48 | - |
| | Qwen 32B | 69.64 | 70.62 | 70.85 | 70.51 | 70.47 | 70.62 | 71.35 | 69.63 | 68.70 | 71.30 | 75.19 | 70.67 | 85.56 | - |
| | Qwen 72B | 67.32 | 67.79 | 66.58 | 67.14 | 67.74 | 67.98 | 66.39 | 68.15 | 68.70 | 67.41 | 71.48 | 67.11 | 85.56 | - |
| | Qwen-M 7B | 61.98 | 68.10 | 67.15 | 66.89 | 67.19 | 67.43 | 64.68 | 67.78 | 70.00 | 68.15 | 62.96 | 46.26 | 81.85 | - |
| | Dpsk-M 7B | 27.55 | 34.02 | 29.36 | 31.84 | 29.61 | 37.14 | 28.84 | 39.44 | 41.48 | 34.81 | 26.30 | 28.78 | 72.22 | - |
| Humaneval+ | Llama 8B | 57.33 | 58.53 | 56.96 | 60.50 | 57.78 | 61.74 | 59.59 | 60.37 | 64.33 | 67.07 | 63.35 | 56.70 | 79.88 | - |
| | Llama 70B | 73.72 | 73.47 | 73.39 | 72.85 | 73.34 | 73.94 | 72.43 | 77.44 | 78.66 | 77.44 | 75.61 | 73.23 | 90.85 | - |
| | Qwen 32B | 83.05 | 79.01 | 82.53 | 82.58 | 82.60 | 82.51 | 81.00 | 84.76 | 81.71 | 85.37 | 81.10 | 82.56 | 93.29 | - |
| | Qwen 72B | 82.29 | 84.02 | 82.16 | 84.59 | 82.07 | 83.60 | 83.58 | 84.15 | 84.76 | 87.80 | 82.32 | 82.13 | 93.90 | - |
| | Qwen-C 7B | 81.96 | 82.20 | 81.93 | 81.98 | 82.29 | 82.39 | 80.37 | 81.71 | 79.27 | 82.32 | 86.59 | 81.52 | 93.90 | - |
| | Dpsk-C 6.7B | 66.34 | 68.83 | 67.95 | 69.63 | 67.95 | 71.56 | 65.71 | 72.56 | 73.78 | 77.44 | 71.95 | 67.62 | 86.59 | - |
| MBPP+ | Llama 8B | 58.09 | 55.70 | 55.82 | 56.76 | 56.56 | 60.33 | 56.80 | 60.32 | 60.85 | 56.88 | 54.50 | 55.21 | 76.46 | - |
| | Llama 70B | 64.85 | 62.64 | 63.90 | 62.10 | 64.13 | 64.22 | 62.40 | 64.55 | 68.39 | 63.23 | 65.08 | 63.68 | 83.07 | - |
| | Qwen 32B | 75.16 | 75.38 | 75.66 | 74.74 | 75.51 | 75.39 | 75.67 | 75.13 | 76.06 | 75.40 | 75.40 | 75.45 | 84.13 | - |
| | Qwen 72B | 76.46 | 76.32 | 75.93 | 75.93 | 76.48 | 76.85 | 77.33 | 76.72 | 75.26 | 76.46 | 76.19 | 75.71 | 84.66 | - |
| | Qwen-C 7B | 68.89 | 67.77 | 66.85 | 70.15 | 67.96 | 71.83 | 66.79 | 71.43 | 72.22 | 71.56 | 71.69 | 66.51 | 85.45 | - |
| | Dpsk-C 6.7B | 64.08 | 60.50 | 62.67 | 61.15 | 62.50 | 63.07 | 62.49 | 63.62 | 65.87 | 58.20 | 66.14 | 62.12 | 79.37 | - |
| BigCodeBench | Llama 8B | 29.48 | 30.40 | 28.88 | 31.20 | 29.90 | 32.27 | 29.78 | 32.63 | 34.96 | 35.09 | 31.67 | 28.03 | 56.84 | - |
| | Llama 70B | 42.66 | 42.27 | 42.55 | 41.94 | 42.66 | 42.88 | 42.72 | 43.63 | 43.90 | 43.86 | 45.44 | 41.92 | 62.63 | - |
| | Qwen 32B | 44.77 | 45.63 | 44.77 | 45.83 | 45.53 | 45.31 | 45.58 | 46.05 | 46.07 | 46.23 | 45.53 | 44.66 | 65.18 | - |
| | Qwen 72B | 46.25 | 47.04 | 46.32 | 47.20 | 46.95 | 46.99 | 46.79 | 46.89 | 48.60 | 48.33 | 46.67 | 46.30 | 60.18 | - |
| | Qwen-C 7B | 37.97 | 39.98 | 38.07 | 39.61 | 39.09 | 41.07 | 39.36 | 41.05 | 42.37 | 42.06 | 42.37 | 37.89 | 61.58 | - |
| | Dpsk-C 6.7B | 33.61 | 33.83 | 32.90 | 35.20 | 34.66 | 37.64 | 33.63 | 36.84 | 38.99 | 38.86 | 34.91 | 32.49 | 58.95 | - |
| AlpacaEval | Llama 8B | 29.02 | 30.78 | 27.86 | 30.16 | 28.79 | 29.32 | 29.08 | 33.62 | 36.05 | 38.98 | 27.26 | 26.80 | 55.07 | - |
| | Llama 70B | 36.05 | 38.19 | 35.64 | 36.60 | 36.48 | 36.17 | 36.32 | 39.10 | 42.05 | 45.08 | 35.48 | 34.27 | 61.90 | - |
| | Qwen 32B | 37.06 | 38.06 | 36.34 | 37.62 | 37.77 | 36.84 | 37.08 | 39.29 | 42.42 | 45.17 | 32.92 | 35.80 | 61.69 | - |
| | Qwen 72B | 55.17 | 55.85 | 53.91 | 55.04 | 55.16 | 54.96 | 55.03 | 55.95 | 60.34 | 62.56 | 52.01 | 56.76 | 76.52 | - |
| IFEval | Llama 8B | 67.99 | 69.37 | 69.11 | 69.49 | 69.75 | 70.48 | 69.32 | 72.46 | 71.07 | 73.01 | 72.83 | 66.43 | 87.80 | - |
| | Llama 70B | 81.50 | 80.94 | 81.96 | 82.20 | 80.84 | 82.35 | 80.84 | 81.15 | 82.62 | 81.89 | 84.29 | 79.94 | 92.42 | - |
| | Qwen 32B | 79.34 | 79.97 | 79.94 | 79.46 | 80.17 | 80.21 | 79.42 | 82.38 | 80.41 | 81.42 | 80.22 | 77.38 | 90.02 | - |
| | Qwen 72B | 84.12 | 83.55 | 83.19 | 83.90 | 82.39 | 84.14 | 83.01 | 84.57 | 86.04 | 83.46 | 83.55 | 81.44 | 92.24 | - |

Table 5: Judge reranking performance under the additive scale single-instance rating protocol.

**Additional visualizations for pairwise protocols.** In Fig. 20, we visualize the performance of judges capable of both single-rating evaluation and pairwise evaluation to present a wholistic picture of reranking performance under different protocols. For the majority of benchmark and judge combinations, pairwise reranking tends to perform the best, with additive and Likert single-rating protocols performing roughly equally.

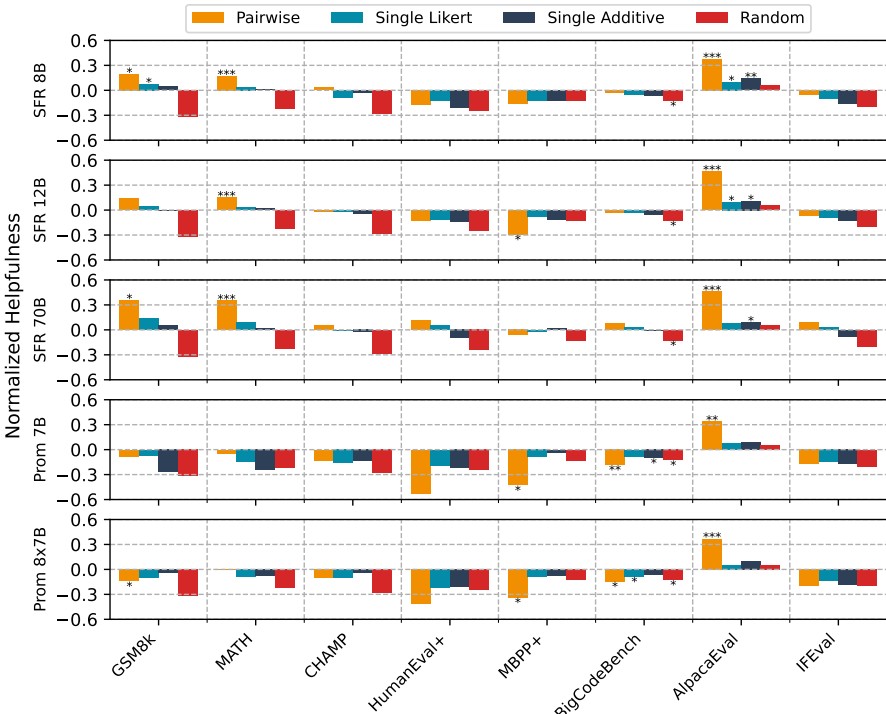

Figure 20: The reranking performance for different datasets, for each judge that supports both pairwise and single-rating protocols.

In Fig. 21, we present an expanded version of Fig. 4 by plotting the minimum, maximum, mean, and distribution of normalized helpfulness for all pairwise judges. A similar set of plots is shown in Fig. 22 for single-rating protocols. Judges tend to reach higher maximum performance under pairwise prompts, while the minimum possible performance is relatively stable between the protocols.

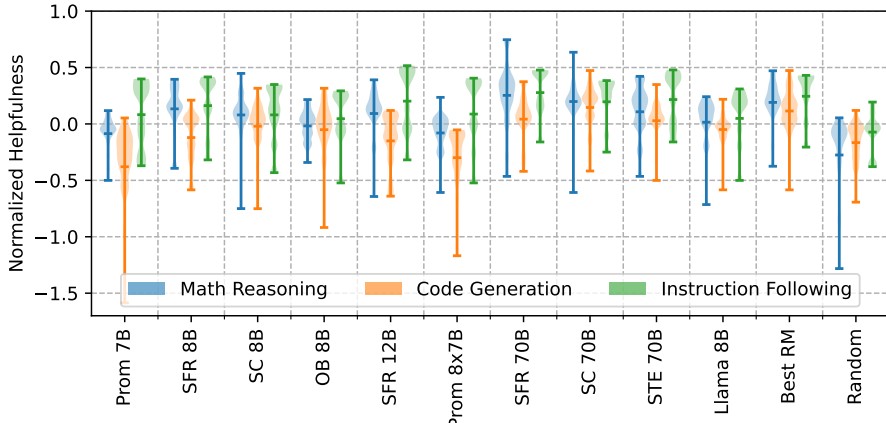

Figure 21: The violin plot showing the full distributions of judges' pairwise reranking performance. An equivalent bar chart containing only the mean is presented in Fig. 4.

**Judges are able to improve AlpacaEval length-controlled win-rates as well.** It is well established that LLM-judges are susceptible to length bias (Zeng et al., 2023; Park et al., 2024), with judges preferring longer responses, even if they do not follow user instructions. As a result, AlpacaEval designed a *length-controlled* win rate (Dubois et al., 2024a), which fits a generalized linear model in an attempt to normalize response preferences based on length of responses. As we show in Tab. 6, pairwise reranking can lead to large gains in length-controlled win rate as well, highlighting that the responses judges

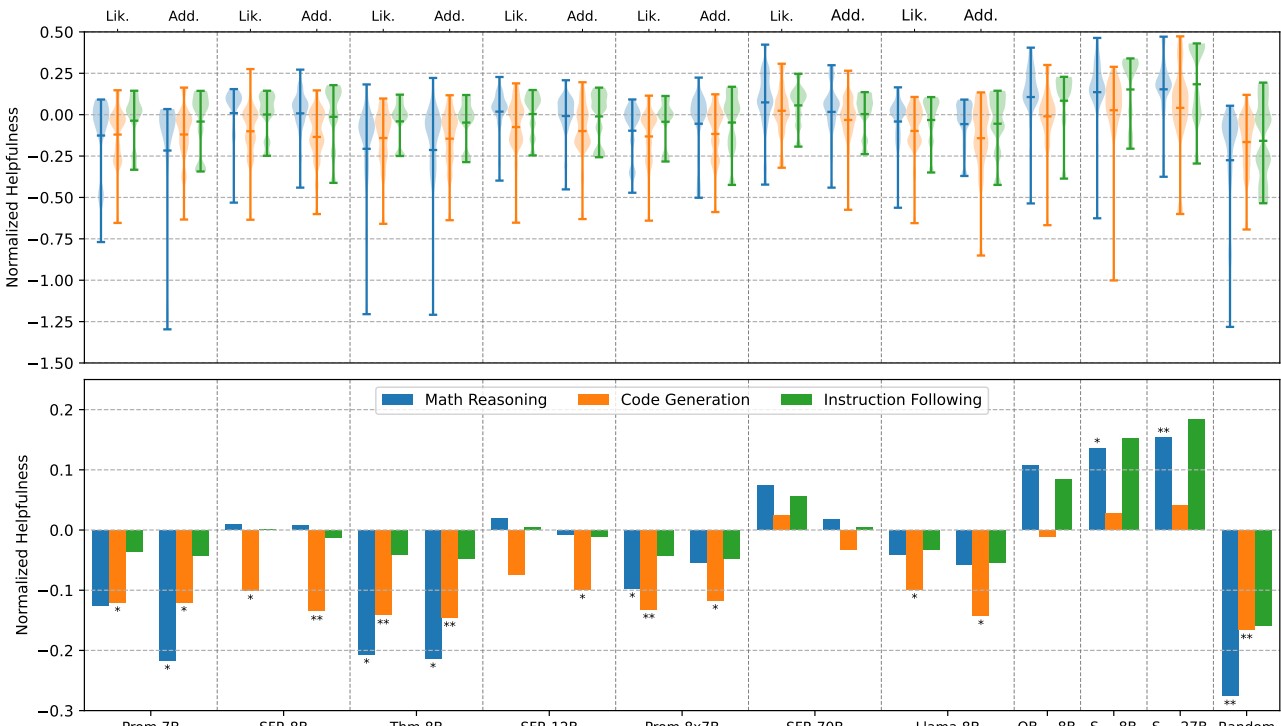

Figure 22: Normalized helpfulness of the single-rating protocol (with both the Likert and additive prompt) across task categories of each judge, compared to Llama-3.1 8B with judge prompt, best reward model and random reranking, averaged across generator models and datasets. The top panel shows the full distribution while the bottom panel only shows the mean.

select are high quality, regardless of length. In these settings, the Skywork-Critic models excel. Single-rating protocols also result in performance gains over greedy responses, but like the standard win rate setting, the gains are far less pronounced. Full results for Likert single-rating are shown in Tab. 7 and additive single-rating are shown in Tab. 8.

|  | LC | Prom 7B | SFR 8B | SC 8B | OB 8B | SFR 12B | Prom 8x7B | SFR 70B | SC 70B | STE 70B | Llama 8B | Best RM | Greedy | Random | Oracle |
|---|---|---|---|---|---|---|---|---|---|---|---|---|---|---|---|
| Llama 8B | ✗ | 34.94 | 37.15 | 36.98 | 34.23 | 39.93 | 37.31 | 40.54 | 37.86 | 37.63 | 32.00 | 37.86 | 27.26 | 26.80 | 55.07 |
|  | ✓ | 31.65 | 32.83 | 36.33 | 31.81 | 35.94 | 32.94 | 37.20 | 37.09 | 33.23 | 29.53 | 37.09 | 25.33 | 25.05 | 52.59 |
| Llama 70B | ✗ | 46.02 | 44.88 | 42.88 | 40.99 | 45.92 | 45.20 | 46.87 | 43.90 | 45.69 | 40.17 | 45.69 | 35.48 | 34.27 | 61.90 |
|  | ✓ | 46.12 | 43.67 | 46.14 | 43.11 | 45.24 | 44.09 | 46.88 | 47.38 | 41.08 | 41.38 | 47.38 | 36.85 | 36.57 | 63.19 |
| Qwen 32B | ✗ | 44.18 | 43.82 | 42.70 | 41.36 | 47.78 | 44.59 | 45.97 | 43.66 | 46.70 | 41.84 | 46.70 | 32.92 | 35.80 | 61.69 |
|  | ✓ | 46.94 | 47.17 | 47.38 | 45.41 | 50.07 | 46.21 | 48.61 | 49.13 | 47.48 | 45.21 | 49.13 | 38.40 | 40.52 | 64.75 |
| Qwen 72B | ✗ | 58.94 | 62.22 | 55.98 | 58.35 | 64.16 | 60.51 | 63.69 | 61.43 | 63.03 | 58.73 | 63.03 | 52.01 | 56.76 | 76.52 |
|  | ✓ | 52.27 | 52.41 | 55.48 | 52.35 | 55.09 | 51.56 | 56.07 | 57.42 | 54.24 | 52.13 | 57.42 | 47.18 | 48.56 | 70.85 |

Table 6: Judge reranking performance under the pairwise protocol on AlpacaEval, showing both non-length-controlled (LC) win rate and length-controlled win rate. The non-LC values are the same as those in Tab. 3.

|  | LC | Prom 7B | SFR 8B | Thm 8B | SFR 12B | Prom 8x7B | SFR 70B | Llama 8B | $OB_{RM}$ 8B | $S_{RM}$ 8B | $S_{RM}$ 27B | Greedy | Random | Oracle |
|---|---|---|---|---|---|---|---|---|---|---|---|---|---|---|
| Llama 8B | ✗ | 28.42 | 29.95 | 27.80 | 29.50 | 27.64 | 29.43 | 27.36 | 33.62 | 36.05 | 38.98 | 27.26 | 26.80 | 55.07 |
|  | ✓ | 26.62 | 27.79 | 25.84 | 27.29 | 25.95 | 27.57 | 26.25 | 32.87 | 34,78 | 35.58 | 25.33 | 25.05 | 52.59 |
| Llama 70B | ✗ | 36.13 | 36.86 | 35.01 | 36.55 | 35.50 | 35.60 | 33.99 | 39.10 | 42.05 | 45.08 | 35.48 | 34.27 | 61.90 |
|  | ✓ | 38.70 | 38.75 | 37.43 | 38.60 | 38.01 | 37.95 | 37.84 | 42.41 | 44.90 | 46.18 | 36.85 | 36.57 | 63.19 |
| Qwen 32B | ✗ | 37.08 | 37.08 | 36.40 | 37.21 | 36.17 | 36.54 | 35.98 | 39.29 | 42.42 | 45.17 | 32.92 | 35.80 | 61.69 |
|  | ✓ | 42.05 | 41.57 | 41.29 | 42.00 | 41.11 | 41.40 | 41.21 | 45.14 | 46.36 | 46.37 | 38.40 | 40.52 | 64.75 |
| Qwen 72B | ✗ | 54.40 | 54.75 | 53.80 | 54.99 | 54.05 | 54.54 | 54.08 | 55.95 | 60.34 | 62.56 | 52.01 | 56.76 | 76.52 |
|  | ✓ | 49.26 | 49.40 | 48.57 | 49.80 | 48.90 | 49.54 | 49.50 | 53.49 | 54.99 | 54.44 | 47.18 | 48.56 | 70.85 |

Table 7: Judge reranking performance under the Likert scale single-instance rating protocol on AlpacaEval, showing both non-length-controlled (LC) win rate and length-controlled win rate. The non-LC values are the same as those in Tab. 4.

| | LC | Prom 7B | SFR 8B | Thm 8B | SFR 12B | Prom 8x7B | SFR 70B | Llama 8B | OB$_{RM}$ 8B | S$_{RM}$ 8B | S$_{RM}$ 27B | Greedy | Random | Oracle |
|---|---|---|---|---|---|---|---|---|---|---|---|---|---|---|
| Llama 8B | ✗ | 29.02 | 30.78 | 27.86 | 30.16 | 28.79 | 29.32 | 29.08 | 33.62 | 36.05 | 38.98 | 27.26 | 26.80 | 55.07 |
| | ✓ | 27.15 | 28.54 | 25.94 | 27.97 | 26.58 | 27.13 | 26.84 | 32.87 | 34,78 | 35.58 | 25.33 | 25.05 | 52.59 |
| Llama 70B | ✗ | 36.05 | 38.19 | 35.64 | 36.60 | 36.48 | 36.17 | 36.32 | 39.10 | 42.05 | 45.08 | 35.48 | 34.27 | 61.90 |
| | ✓ | 38.51 | 39.88 | 38.00 | 38.51 | 38.32 | 38.15 | 37.92 | 42.41 | 44.90 | 46.18 | 36.85 | 36.57 | 63.19 |
| Qwen 32B | ✗ | 37.06 | 38.06 | 36.34 | 37.62 | 37.77 | 36.84 | 37.08 | 39.29 | 42.42 | 45.17 | 32.92 | 35.80 | 61.69 |
| | ✓ | 41.68 | 42.14 | 41.29 | 42.22 | 42.07 | 41.19 | 41.46 | 45.14 | 46.36 | 46.37 | 38.40 | 40.52 | 64.75 |
| Qwen 72B | ✗ | 55.17 | 55.85 | 53.91 | 55.04 | 55.16 | 54.96 | 55.03 | 55.95 | 60.34 | 62.56 | 52.01 | 56.76 | 76.52 |
| | ✓ | 49.56 | 50.37 | 48.72 | 49.78 | 49.74 | 49.76 | 49.31 | 53.49 | 54.99 | 54.44 | 47.18 | 48.56 | 70.85 |

Table 8: Judge reranking performance under the additive scale single-instance rating protocol on AlpacaEval, showing both non-length-controlled (LC) win rate and length-controlled win rate. The non-LC values are the same as those in Tab. 5.

**Domain-specific prompts do not lead to judge improvements.** Because the SFR-Judge family is trained to have flexible evaluation criteria, we experiment with prompting SFR-Judge-8B with domain-specific prompts for math reasoning and code generation benchmarks. We craft these prompts with an emphasis on correctness of output and validity of intermediate reasoning steps. The evaluation criteria for both prompts is shown in Fig. 19, with output format kept the same from the original prompt in Fig. 15. We show the impacts of this domain-specific prompting in Fig. 23. Namely, performance drops on all benchmarks, though no difference is statistically significant at $\alpha = 0.05$. This demonstrates that the instruction following to reasoning domain gap discussed in Sec. 4.2 cannot be bridged with specialized prompting, which is more likely to be detrimental instead.

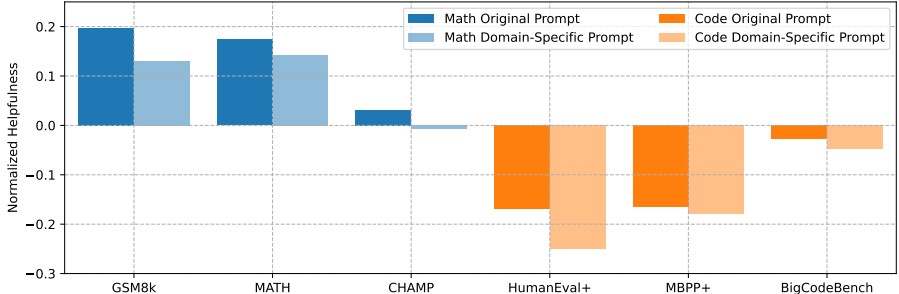

Figure 23: Normalized helpfulness of SFR-Judge-8B across all benchmarks, averaged across all applicable models. Domain-specific evaluation prompting performs worse in all cases.

**Additional visualizations for single-rating protocols.** The performance of judges based on judge-to-generator ratio for single rating protocols is presented in Fig. 24. The positive effect for math is still present, but notably diminished compared to that of the pairwise protocol (Fig. 5), whereas the sizable instruction following advantage found in the pairwise case is eliminated.

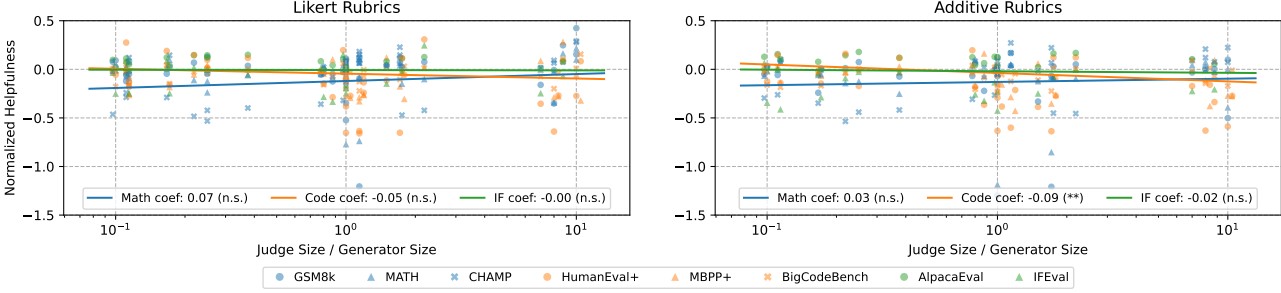

Figure 24: The effect of judge-to-generator size ratio on the normalized helpfulness for under single-rating reranking protocols.

Finally, we plot the distribution of the fraction of tied, top-rated responses for single-rating protocols in Fig. 25. Notably, under the additive rating protocol, the Prometheus models are consistently able reduce the number of top-rated responses compared to other models. Interestingly, this reduction in top-rated responses does not translate to improved average performance relative to other models, as shown in Fig. 7, which may suggest that the judge is assessing based on surface-level factors. Other judges, for the most part, struggle even with finer-grained additive evaluation criteria, tending to assert a large fraction of responses as top-quality, regardless of domain.

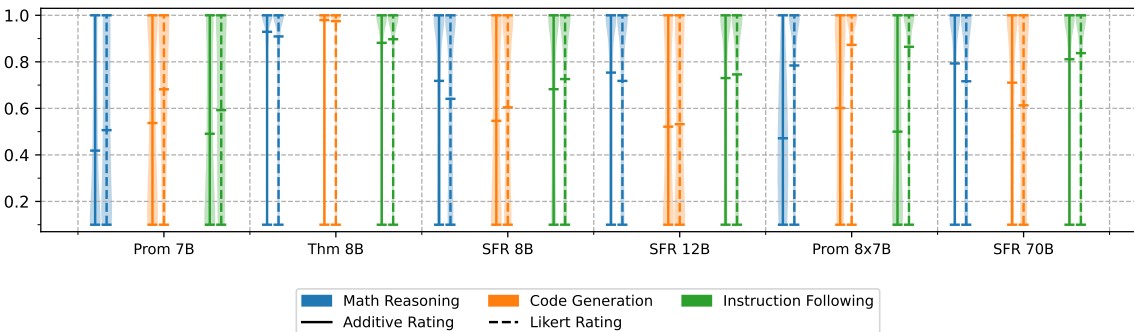

Figure 25: The distribution of fraction of total responses tied as top-rated under single-rating reranking protocols.

## B.2 Step-Level Beam Search

We present full results for our beam search experiments in Tab. 9. As Mao et al. (2024) found out for the CHAMP dataset, models' full solution accuracy is often much lower than their final answer accuracy. Furthermore, even state-of-the-art LLMs such as GPT-4 and 4 Turbo struggle to identify mistakes in intermediate steps. Thus, the oracle accuracy for math datasets, defined based on final answer accuracy, is likely to be a (severe) over-estimate of the true value. Such issues may also be present for code datasets, when a solution contains bugs not identified by the test cases, though the issue may be less extensive as we empirically find the trees to contain fewer leaf nodes (945 for math vs. 192 for code) and recent efforts in code evaluation have emphasized testing rigor (Liu et al., 2023a; Zhuo et al., 2024).

We additionally report the performance of the Qwen2.5-Math-PRM, a 7B PRM specialized for the math domain. Despite its specialization to math, it also offers competitive performance in coding domain tasks, largely outperforming comparably sized LLM-judge models in beam search.

| | | Lookahead | Prom 7B | SFR 8B | SC 8B | OB 8B | Thm 8B | SFR 12B | Prom 8x7B | SFR 70B | SC 70B | STE 70B | Q$_{PRM}$ 7B | Greedy | Random | Oracle |
|---|---|---|---|---|---|---|---|---|---|---|---|---|---|---|---|---|
| GSM8k | Llama 8B | ✗ | 85.67 | 88.86 | 88.70 | 87.19 | 85.75 | 89.31 | 85.44 | 92.12 | 91.96 | 89.23 | 93.78 | 87.11 | 82.34 | 99.47 |
| | | ✓ | 83.62 | 89.84 | 90.90 | 87.34 | 86.05 | 89.61 | 83.70 | 94.09 | 92.12 | 90.83 | 94.47 | | | |
| | Llama 70B | ✗ | 94.69 | 95.07 | 95.91 | 95.75 | 95.60 | 95.45 | 94.16 | 95.45 | 96.51 | 95.15 | 96.97 | 95.38 | 94.25 | 99.32 |
| | | ✓ | 94.47 | 95.83 | 96.06 | 95.60 | 95.15 | 95.91 | 94.47 | 95.91 | 96.51 | 95.07 | 96.89 | | | |
| | Qwen 32B | ✗ | 94.92 | 95.75 | 95.53 | 95.75 | 95.07 | 95.22 | 95.07 | 95.07 | 95.60 | 94.84 | 97.12 | 95.15 | 95.03 | 99.01 |
| | | ✓ | 95.07 | 95.68 | 95.07 | 95.91 | 94.84 | 96.21 | 94.54 | 96.36 | 96.21 | 95.38 | 96.97 | | | |
| | Qwen 72B | ✗ | 95.45 | 95.38 | 95.60 | 95.98 | 96.06 | 95.30 | 95.15 | 95.91 | 95.91 | 95.60 | 96.59 | 95.91 | 95.66 | 98.56 |
| | | ✓ | 95.38 | 94.92 | 96.29 | 95.68 | 95.60 | 95.60 | 95.22 | 96.21 | 96.29 | 95.75 | 97.35 | | | |
| | Qwen-M 7B | ✗ | 93.56 | 94.69 | 95.00 | 94.62 | 95.00 | 95.15 | 93.86 | 95.07 | 95.15 | 94.84 | 95.75 | 94.24 | 90.96 | 99.17 |
| | | ✓ | 92.95 | 94.39 | 94.69 | 94.54 | 94.84 | 95.07 | 93.63 | 95.53 | 95.15 | 94.92 | 95.75 | | | |
| | Dpsk-M 7B | ✗ | 79.45 | 83.55 | 83.85 | 81.35 | 80.29 | 83.62 | 81.27 | 89.16 | 88.02 | 86.13 | 92.34 | 81.65 | 79.95 | 99.09 |
| | | ✓ | 80.67 | 87.95 | 87.04 | 84.31 | 80.82 | 87.64 | 83.70 | 91.96 | 90.30 | 88.70 | 92.12 | | | |
| MATH | Llama 8B | ✗ | 22.05 | 28.32 | 24.77 | 23.19 | 22.96 | 26.51 | 24.55 | 34.14 | 29.38 | 27.79 | 42.60 | 23.87 | 21.47 | 93.28 |
| | | ✓ | 24.09 | 29.91 | 27.34 | 25.76 | 24.70 | 29.46 | 25.68 | 40.48 | 32.63 | 28.40 | 48.56 | | | |
| | Llama 70B | ✗ | 43.05 | 44.18 | 45.09 | 44.34 | 43.66 | 46.37 | 44.11 | 47.96 | 44.34 | 44.86 | 55.36 | 43.20 | 41.75 | 94.71 |
| | | ✓ | 43.05 | 49.40 | 46.75 | 45.69 | 42.52 | 48.26 | 44.41 | 53.10 | 48.19 | 45.77 | 60.50 | | | |
| | Qwen 32B | ✗ | 54.00 | 56.87 | 54.61 | 54.31 | 52.19 | 56.57 | 53.47 | 59.14 | 57.55 | 54.46 | 66.01 | 57.10 | 53.54 | 93.50 |
| | | ✓ | 52.72 | 58.46 | 60.27 | 55.82 | 53.47 | 59.97 | 54.91 | 64.65 | 61.71 | 58.99 | 68.81 | | | |
| | Qwen 72B | ✗ | 62.24 | 64.95 | 64.65 | 63.67 | 62.01 | 65.18 | 64.20 | 66.09 | 65.63 | 63.22 | 70.92 | 62.99 | 61.35 | 93.88 |
| | | ✓ | 61.78 | 65.26 | 65.63 | 61.33 | 62.61 | 65.48 | 62.39 | 68.20 | 66.92 | 66.01 | 72.89 | | | |
| | Qwen-M 7B | ✗ | 65.18 | 67.37 | 68.13 | 66.39 | 65.86 | 68.58 | 66.77 | 64.95 | 68.96 | 67.60 | 72.28 | 65.79 | 57.32 | 93.81 |
| | | ✓ | 63.90 | 68.73 | 68.58 | 66.01 | 67.22 | 70.02 | 66.54 | 74.09 | 70.69 | 70.69 | 76.21 | | | |
| | Dpsk-M 7B | ✗ | 16.24 | 19.49 | 19.41 | 19.79 | 17.45 | 19.26 | 17.52 | 25.76 | 24.62 | 20.92 | 34.29 | 19.03 | 17.08 | 85.20 |
| | | ✓ | 16.54 | 22.96 | 24.62 | 22.81 | 18.96 | 24.70 | 21.30 | 32.93 | 28.85 | 26.06 | 39.88 | | | |
| CHAMP | Llama 8B | ✗ | 30.37 | 35.93 | 30.37 | 34.07 | 28.52 | 35.93 | 28.89 | 33.86 | 38.52 | 33.70 | 39.63 | 28.52 | 28.97 | 98.52 |
| | | ✓ | 27.78 | 37.41 | 36.30 | 34.81 | 28.15 | 36.67 | 31.11 | 43.33 | 38.89 | 35.19 | 44.81 | | | |
| | Llama 70B | ✗ | 43.33 | 52.59 | 51.48 | 54.07 | 52.59 | 48.52 | 45.19 | 54.81 | 52.96 | 50.74 | 58.15 | 52.96 | 46.76 | 100.0 |
| | | ✓ | 46.67 | 54.81 | 53.70 | 58.52 | 50.37 | 51.11 | 47.04 | 54.81 | 55.93 | 52.96 | 62.59 | | | |
| | Qwen 32B | ✗ | 69.63 | 70.74 | 71.85 | 74.07 | 72.59 | 69.63 | 70.74 | 74.44 | 75.56 | 71.11 | 74.44 | 72.96 | 72.16 | 97.04 |
| | | ✓ | 68.89 | 72.22 | 72.22 | 74.07 | 73.33 | 73.33 | 70.00 | 78.15 | 74.07 | 72.22 | 78.52 | | | |
| | Qwen 72B | ✗ | 73.70 | 69.26 | 71.85 | 70.74 | 72.59 | 69.63 | 74.81 | 75.93 | 73.33 | 72.96 | 77.04 | 72.96 | 71.76 | 97.78 |
| | | ✓ | 69.26 | 70.37 | 71.11 | 72.96 | 70.74 | 69.26 | 70.37 | 74.81 | 72.96 | 74.44 | 75.56 | | | |
| | Qwen-M 7B | ✗ | 65.19 | 67.04 | 66.67 | 66.30 | 70.74 | 66.67 | 67.04 | 67.78 | 67.41 | 65.56 | 70.37 | 64.44 | 49.10 | 96.67 |
| | | ✓ | 56.67 | 65.56 | 64.44 | 64.44 | 63.70 | 64.07 | 62.96 | 73.70 | 69.63 | 68.52 | 75.93 | | | |
| | Dpsk-M 7B | ✗ | 24.07 | 32.96 | 28.52 | 31.48 | 29.26 | 28.15 | 27.41 | 41.11 | 35.93 | 36.67 | 44.44 | 27.41 | 28.08 | 95.93 |
| | | ✓ | 22.96 | 35.93 | 31.11 | 37.41 | 33.33 | 36.30 | 28.15 | 44.07 | 38.89 | 37.78 | 50.37 | | | |
| HumanEval+ | Llama 8B | ✗ | 57.93 | 65.24 | 59.76 | 63.41 | 57.32 | 62.80 | 60.37 | 65.85 | 61.59 | 65.24 | 60.37 | 59.15 | 57.55 | 89.02 |
| | | ✓ | 58.54 | 62.20 | 60.37 | 62.20 | 59.15 | 65.85 | 59.15 | 65.85 | 65.85 | 65.24 | 67.07 | | | |
| | Llama 70B | ✗ | 68.29 | 73.17 | 76.83 | 70.12 | 71.95 | 71.34 | 69.51 | 74.39 | 73.78 | 73.17 | 77.44 | 74.39 | 71.87 | 92.68 |
| | | ✓ | 70.12 | 73.78 | 75.00 | 73.17 | 73.17 | 73.78 | 71.95 | 70.73 | 77.44 | 76.83 | 76.83 | | | |
| | Qwen 32B | ✗ | 82.32 | 82.32 | 84.76 | 81.71 | 83.54 | 85.98 | 81.71 | 85.37 | 85.98 | 85.37 | 83.54 | 82.32 | 81.94 | 95.12 |
| | | ✓ | 81.71 | 79.27 | 82.93 | 81.10 | 84.15 | 84.76 | 79.88 | 84.76 | 87.20 | 84.76 | 81.71 | | | |
| | Qwen 72B | ✗ | 81.10 | 81.71 | 80.49 | 81.10 | 81.71 | 82.32 | 80.49 | 82.32 | 82.32 | 82.32 | 81.71 | 80.49 | 80.38 | 92.07 |
| | | ✓ | 79.88 | 79.27 | 81.10 | 81.10 | 81.71 | 82.93 | 81.10 | 82.32 | 82.93 | 82.93 | 82.93 | | | |
| | Qwen-C 7B | ✗ | 78.66 | 78.66 | 81.71 | 81.71 | 83.54 | 81.10 | 78.66 | 81.71 | 82.93 | 82.32 | 85.98 | 84.15 | 80.12 | 95.12 |
| | | ✓ | 78.05 | 79.88 | 81.10 | 81.71 | 78.66 | 84.15 | 81.10 | 79.88 | 82.32 | 79.27 | 82.32 | | | |
| | Dpsk-C 6.7B | ✗ | 62.80 | 66.46 | 64.63 | 65.85 | 64.02 | 65.85 | 63.41 | 67.68 | 70.73 | 66.46 | 70.73 | 69.51 | 64.94 | 90.85 |
| | | ✓ | 63.41 | 69.51 | 71.95 | 69.51 | 68.90 | 70.73 | 64.63 | 76.83 | 71.95 | 73.17 | 76.22 | | | |
| MBPP+ | Llama 8B | ✗ | 58.99 | 59.52 | 60.58 | 61.11 | 60.05 | 58.20 | 56.08 | 59.79 | 64.81 | 62.17 | 61.11 | 59.52 | 57.12 | 84.13 |
| | | ✓ | 55.82 | 55.29 | 60.32 | 62.96 | 58.73 | 57.14 | 53.44 | 61.38 | 64.81 | 62.70 | 61.90 | | | |
| | Llama 70B | ✗ | 62.70 | 62.43 | 66.40 | 64.81 | 64.02 | 65.08 | 60.05 | 64.55 | 67.99 | 65.87 | 68.25 | 66.14 | 63.64 | 88.62 |
| | | ✓ | 60.32 | 62.70 | 66.40 | 66.40 | 61.38 | 64.55 | 61.38 | 60.85 | 64.81 | 66.40 | 66.40 | | | |
| | Qwen 32B | ✗ | 75.66 | 77.25 | 74.60 | 76.72 | 76.46 | 75.93 | 76.72 | 78.04 | 77.78 | 76.72 | 76.19 | 75.66 | 74.51 | 88.62 |
| | | ✓ | 74.34 | 77.51 | 74.34 | 75.40 | 75.13 | 75.13 | 74.60 | 77.51 | 76.46 | 76.72 | 75.93 | | | |
| | Qwen 72B | ✗ | 75.93 | 76.19 | 76.72 | 76.46 | 77.25 | 77.25 | 75.93 | 77.78 | 76.98 | 77.25 | 77.25 | 76.46 | 75.68 | 86.24 |
| | | ✓ | 76.72 | 76.98 | 76.46 | 75.66 | 76.19 | 76.98 | 76.46 | 77.51 | 75.93 | 76.98 | 75.40 | | | |
| | Qwen-C 7B | ✗ | 70.11 | 72.75 | 74.07 | 71.96 | 70.37 | 71.43 | 71.69 | 73.81 | 74.87 | 73.02 | 74.07 | 71.69 | 64.70 | 90.21 |
| | | ✓ | 68.52 | 71.17 | 71.96 | 68.78 | 69.84 | 69.58 | 69.31 | 73.54 | 74.60 | 71.43 | 70.37 | | | |
| | Dpsk-C 6.7B | ✗ | 61.90 | 64.81 | 61.11 | 61.64 | 60.32 | 64.81 | 63.76 | 67.72 | 64.55 | 67.72 | 65.61 | 66.40 | 61.33 | 84.66 |
| | | ✓ | 60.58 | 62.17 | 64.81 | 63.76 | 63.76 | 62.96 | 61.11 | 63.76 | 66.93 | 64.55 | 64.81 | | | |
| BigCodeBench | Llama 8B | ✗ | 29.04 | 30.44 | 31.32 | 31.67 | 29.56 | 30.79 | 28.51 | 33.86 | 34.12 | 33.51 | 32.81 | 33.33 | 28.88 | 66.58 |
| | | ✓ | 26.40 | 33.51 | 33.51 | 31.93 | 30.00 | 32.37 | 28.68 | 36.05 | 34.04 | 35.09 | 33.68 | | | |
| | Llama 70B | ✗ | 40.00 | 40.96 | 41.58 | 42.72 | 43.77 | 42.37 | 40.44 | 42.28 | 43.68 | 42.98 | 42.37 | 43.07 | 41.98 | 67.11 |
| | | ✓ | 39.56 | 41.75 | 44.47 | 42.89 | 43.16 | 41.40 | 41.05 | 42.46 | 44.82 | 43.42 | 42.46 | | | |
| | Qwen 32B | ✗ | 43.33 | 43.68 | 45.09 | 45.79 | 45.35 | 45.18 | 44.82 | 45.70 | 46.49 | 45.79 | 45.18 | 44.47 | 44.26 | 69.47 |
| | | ✓ | 42.81 | 44.91 | 45.61 | 45.96 | 44.56 | 43.95 | 43.77 | 45.53 | 45.61 | 45.96 | 45.44 | | | |
| | Qwen 72B | ✗ | 45.35 | 45.35 | 46.75 | 46.58 | 46.40 | 46.23 | 45.70 | 46.32 | 46.84 | 45.79 | 46.14 | 45.79 | 46.03 | 63.86 |
| | | ✓ | 45.61 | 46.23 | 46.84 | 47.28 | 46.14 | 46.11 | 45.88 | 46.32 | 46.75 | 46.32 | 45.96 | | | |
| | Qwen-C 7B | ✗ | 37.11 | 40.18 | 40.79 | 41.14 | 38.07 | 39.30 | 38.07 | 40.79 | 41.93 | 40.44 | 42.37 | 41.14 | 38.29 | 70.09 |
| | | ✓ | 35.96 | 40.44 | 41.84 | 41.14 | 40.35 | 40.79 | 38.16 | 42.46 | 43.07 | 41.58 | 42.02 | | | |
| | Dpsk-C 6.7B | ✗ | 30.61 | 35.18 | 35.00 | 35.18 | 33.16 | 34.30 | 32.11 | 36.75 | 36.23 | 36.49 | 35.44 | 35.00 | 31.42 | 65.09 |
| | | ✓ | 29.91 | 34.04 | 35.44 | 35.00 | 33.42 | 34.21 | 32.11 | 36.93 | 36.14 | 37.81 | 34.74 | | | |

Table 9: Performance of the final selected node via the judge-guided beam search procedure.

## B.3 Critique-Based Refinement

We present full results for our critique-based refinement experiments in Tab. 10. Here, we report the judge selected response after reranking all generated responses, as well as the performance after just taking the last generated response.

| Bench | Model | Prom 7B Perf. | Last | Rand. | Orac. | SFR 8B Perf. | Last | Rand. | Orac. | Thm 8B Perf. | Last | Rand. | Orac. | SFR 12B Perf. | Last | Rand. | Orac. | Prom8x7B Perf. | Last | Rand. | Orac. | SFR 70B Perf. | Last | Rand. | Orac. | Greedy | Rand.-RR | Orac.-RR |
|---|---|---|---|---|---|---|---|---|---|---|---|---|---|---|---|---|---|---|---|---|---|---|---|---|---|---|---|---|
| GSM8k | Llama 8B | 85.06 | 84.46 | 84.69 | 87.79 | 85.50 | 85.14 | 85.46 | 86.50 | 85.67 | 84.46 | 84.83 | 87.79 | 85.90 | 85.82 | 85.30 | 88.32 | 84.76 | 84.53 | 84.83 | 86.58 | 90.14 | 89.69 | 88.46 | 90.98 | 85.67 | 82.08 | 96.44 |
| | Llama 70B | 95.07 | 94.77 | 95.05 | 95.60 | 95.45 | 95.45 | 95.49 | 95.53 | 95.07 | 95.07 | 94.94 | 96.13 | 95.00 | 94.77 | 95.23 | 95.91 | 95.22 | 95.22 | 95.39 | 95.53 | 95.60 | 95.45 | 95.45 | 95.91 | 95.53 | 95.11 | 98.48 |
| | Qwen 32B | 95.07 | 95.15 | 95.01 | 95.53 | 95.22 | 95.22 | 95.22 | 95.22 | 95.38 | 95.38 | 95.16 | 95.45 | 94.84 | 95.00 | 95.00 | 95.38 | 95.15 | 95.07 | 95.10 | 95.22 | 95.22 | 95.30 | 95.27 | 95.45 | 95.22 | 95.22 | 98.56 |
| | Qwen 72B | 95.75 | 95.68 | 95.65 | 95.83 | 95.68 | 95.68 | 95.68 | 95.68 | 95.53 | 95.53 | 95.58 | 95.68 | 95.53 | 95.53 | 95.63 | 95.75 | 95.68 | 95.68 | 95.68 | 95.68 | 95.75 | 95.75 | 95.70 | 95.83 | 95.68 | 95.52 | 97.88 |
| MATH | Llama 8B | 25.30 | 24.92 | 25.02 | 27.64 | 25.13 | 25.00 | 25.08 | 25.68 | 26.36 | 26.59 | 25.85 | 28.93 | 25.68 | 25.91 | 25.58 | 28.78 | 25.45 | 25.38 | 25.34 | 27.42 | 29.83 | 28.47 | 27.74 | 32.18 | 24.70 | 21.98 | 53.47 |
| | Llama 70B | 43.96 | 44.18 | 44.02 | 45.17 | 44.00 | 44.18 | 44.01 | 44.41 | 43.58 | 43.50 | 43.88 | 46.68 | 44.18 | 43.96 | 43.73 | 46.30 | 44.11 | 43.88 | 43.83 | 44.71 | 44.94 | 44.64 | 44.28 | 45.39 | 43.81 | 42.07 | 68.35 |
| | Qwen 32B | 57.02 | 57.25 | 57.08 | 58.31 | 57.10 | 57.10 | 57.10 | 57.10 | 57.55 | 57.70 | 57.26 | 58.53 | 56.50 | 56.65 | 56.79 | 58.76 | 56.87 | 56.72 | 56.88 | 57.78 | 58.08 | 58.01 | 57.27 | 59.14 | 57.10 | 52.24 | 78.17 |
| | Qwen 72B | 62.92 | 62.76 | 62.80 | 63.22 | 63.26 | 63.29 | 63.14 | 63.29 | 63.14 | 63.07 | 63.10 | 63.82 | 62.54 | 62.61 | 62.70 | 63.90 | 62.92 | 62.99 | 62.91 | 63.29 | 63.29 | 63.07 | 63.04 | 63.97 | 62.99 | 61.37 | 82.78 |
| CHAMP | Llama 8B | 29.63 | 30.00 | 30.45 | 36.30 | 27.78 | 27.41 | 28.80 | 38.89 | 27.39 | 27.41 | 27.94 | 31.11 | 29.26 | 28.89 | 29.41 | 39.26 | 26.67 | 26.67 | 27.39 | 32.22 | 36.30 | 35.93 | 32.86 | 41.85 | 29.26 | 27.30 | 60.00 |
| | Llama 70B | 48.15 | 47.78 | 47.52 | 53.60 | 49.26 | 48.52 | 48.75 | 57.78 | 47.41 | 47.41 | 47.41 | 48.89 | 47.78 | 46.67 | 46.82 | 55.56 | 48.15 | 47.78 | 47.69 | 51.11 | 48.52 | 48.52 | 48.33 | 52.22 | 47.41 | 46.07 | 71.48 |
| | Qwen 32B | 73.33 | 73.33 | 74.04 | 76.30 | 74.81 | 74.07 | 74.58 | 75.56 | 75.33 | 75.19 | 75.25 | 75.56 | 75.19 | 74.81 | 74.49 | 75.93 | 74.44 | 74.81 | 74.83 | 75.19 | 74.81 | 75.19 | 75.12 | 75.56 | 75.19 | 70.67 | 85.56 |
| | Qwen 72B | 71.85 | 71.48 | 71.53 | 71.85 | 70.74 | 71.48 | 71.24 | 72.22 | 71.11 | 71.11 | 71.23 | 71.48 | 71.85 | 71.85 | 71.74 | 72.22 | 71.11 | 71.11 | 71.22 | 71.48 | 71.48 | 71.48 | 71.54 | 71.85 | 71.48 | 67.11 | 85.56 |
| HumanEval+ | Llama 8B | 59.76 | 58.54 | 60.38 | 63.41 | 64.63 | 64.02 | 64.27 | 67.07 | 63.41 | 63.41 | 63.41 | 63.41 | 59.15 | 58.54 | 59.86 | 65.24 | 62.80 | 62.20 | 62.80 | 63.41 | 65.85 | 65.85 | 64.47 | 67.68 | 63.35 | 56.70 | 79.88 |
| | Llama 70B | 71.34 | 71.34 | 73.08 | 75.61 | 71.34 | 71.95 | 73.58 | 76.22 | 75.00 | 75.00 | 75.41 | 75.61 | 74.39 | 73.78 | 74.11 | 79.27 | 75.61 | 75.00 | 75.30 | 75.61 | 75.61 | 75.61 | 75.00 | 77.44 | 75.61 | 73.23 | 90.85 |
| | Qwen 32B | 81.10 | 82.32 | 81.59 | 82.32 | 81.10 | 79.27 | 80.00 | 81.71 | 80.28 | 79.27 | 80.15 | 81.10 | 79.27 | 80.49 | 80.60 | 83.54 | 79.88 | 79.88 | 80.49 | 81.10 | 81.10 | 81.10 | 81.44 | 82.93 | 81.10 | 82.56 | 93.29 |
| | Qwen 72B | 82.93 | 82.32 | 82.47 | 83.54 | 83.54 | 83.54 | 82.97 | 84.76 | 81.71 | 81.71 | 82.01 | 82.32 | 81.71 | 81.71 | 81.97 | 84.76 | 82.32 | 82.32 | 82.32 | 82.32 | 82.93 | 83.54 | 82.13 | 83.54 | 82.32 | 82.13 | 93.90 |
| MBPP+ | Llama 8B | 51.32 | 51.06 | 52.57 | 56.35 | 46.30 | 44.97 | 47.36 | 59.26 | 54.62 | 54.76 | 54.55 | 55.29 | 49.21 | 44.97 | 46.46 | 61.11 | 54.23 | 53.97 | 54.25 | 55.29 | 56.88 | 56.08 | 54.97 | 59.52 | 54.50 | 55.21 | 76.46 |
| | Llama 70B | 64.55 | 64.81 | 64.78 | 66.93 | 58.20 | 57.41 | 59.52 | 69.05 | 65.48 | 65.34 | 65.34 | 65.61 | 58.73 | 60.05 | 60.49 | 69.31 | 64.02 | 64.02 | 64.51 | 65.34 | 64.81 | 65.87 | 64.84 | 67.46 | 65.08 | 63.68 | 83.07 |
| | Qwen 32B | 74.34 | 74.07 | 74.31 | 75.93 | 68.52 | 69.05 | 70.58 | 78.84 | 75.75 | 75.66 | 75.69 | 75.93 | 70.63 | 70.90 | 70.87 | 78.31 | 73.54 | 73.81 | 74.60 | 76.19 | 74.34 | 75.40 | 74.55 | 77.25 | 75.40 | 75.45 | 84.13 |
| | Qwen 72B | 75.66 | 75.40 | 75.84 | 77.25 | 70.37 | 69.58 | 71.12 | 78.57 | 76.46 | 76.46 | 76.32 | 76.46 | 70.63 | 70.63 | 72.17 | 78.57 | 75.13 | 75.13 | 75.75 | 76.72 | 73.81 | 74.07 | 74.45 | 78.04 | 76.19 | 75.71 | 84.66 |
| BigCodeBench | Llama 8B | 31.67 | 31.05 | 31.32 | 32.81 | 30.26 | 29.30 | 29.69 | 35.35 | 31.62 | 31.58 | 31.61 | 32.81 | 32.63 | 32.19 | 31.92 | 37.54 | 30.09 | 30.18 | 30.62 | 34.39 | 34.82 | 34.65 | 33.41 | 37.98 | 31.67 | 28.03 | 56.84 |
| | Llama 70B | 45.18 | 45.18 | 45.41 | 46.23 | 43.07 | 43.07 | 43.85 | 47.19 | 45.75 | 45.70 | 45.57 | 45.88 | 43.07 | 42.72 | 43.27 | 48.07 | 43.77 | 43.86 | 44.50 | 47.02 | 45.09 | 45.35 | 45.16 | 46.40 | 45.44 | 41.92 | 62.63 |
| | Qwen 32B | 45.09 | 45.18 | 45.27 | 46.14 | 45.09 | 44.91 | 45.06 | 47.63 | 45.53 | 45.53 | 45.53 | 45.53 | 44.21 | 44.21 | 44.67 | 46.93 | 44.56 | 44.39 | 44.91 | 45.79 | 45.70 | 45.96 | 45.70 | 47.19 | 45.53 | 44.66 | 65.18 |
| | Qwen 72B | 46.58 | 46.67 | 46.63 | 47.19 | 44.91 | 44.91 | 45.70 | 47.63 | 46.74 | 46.75 | 46.74 | 46.75 | 46.14 | 46.05 | 46.26 | 47.63 | 45.70 | 45.61 | 46.10 | 47.11 | 47.28 | 47.11 | 46.93 | 48.16 | 46.67 | 46.30 | 60.18 |
| AlpacaEval | Llama 8B | 30.20 | 27.94 | 28.12 | 39.01 | 30.78 | 30.38 | 28.56 | 38.24 | 28.68 | 29.18 | 28.32 | 34.07 | 33.28 | 30.34 | 29.35 | 39.11 | 30.04 | 29.33 | 29.26 | 36.83 | 30.22 | 28.60 | 28.02 | 36.35 | 27.26 | 26.80 | 55.07 |
| | Llama 70B | 37.00 | 36.15 | 36.21 | 41.49 | 36.19 | 35.23 | 35.85 | 39.52 | 35.04 | 35.18 | 35.80 | 36.64 | 35.60 | 35.26 | 35.76 | 37.65 | 35.43 | 35.48 | 35.92 | 36.97 | 35.37 | 34.79 | 35.52 | 36.69 | 35.48 | 34.27 | 61.90 |
| | Qwen 32B | 37.14 | 37.01 | 35.25 | 39.67 | 32.26 | 32.26 | 32.92 | 32.92 | 32.14 | 32.13 | 32.38 | 34.68 | 32.58 | 33.44 | 33.33 | 34.68 | 32.57 | 32.47 | 31.83 | 32.88 | 32.73 | 32.57 | 32.94 | 34.31 | 32.92 | 35.80 | 61.69 |
| | Qwen 72B | 56.21 | 55.83 | 53.82 | 56.97 | 52.46 | 51.00 | 51.22 | 53.84 | 52.46 | 52.31 | 51.85 | 52.58 | 52.57 | 52.47 | 51.83 | 52.88 | 53.14 | 53.19 | 52.30 | 52.86 | 52.97 | 51.68 | 51.78 | 53.48 | 52.01 | 50.76 | 76.52 |
| IFEval | Llama 8B | 67.65 | 68.39 | 69.45 | 77.45 | 74.86 | 73.38 | 73.05 | 79.85 | 71.93 | 73.20 | 72.16 | 76.52 | 74.49 | 75.05 | 73.06 | 80.78 | 70.61 | 70.98 | 71.27 | 75.79 | 76.52 | 76.52 | 74.72 | 79.48 | 72.83 | 66.43 | 87.80 |
| | Llama 70B | 83.36 | 82.44 | 83.50 | 87.25 | 84.10 | 83.36 | 83.58 | 87.99 | 84.53 | 84.47 | 84.39 | 85.03 | 85.40 | 84.29 | 84.52 | 86.69 | 83.92 | 84.24 | 84.13 | 85.03 | 85.58 | 85.21 | 84.80 | 85.77 | 84.29 | 79.94 | 92.42 |
| | Qwen 32B | 79.67 | 79.85 | 79.95 | 82.62 | 82.26 | 82.26 | 81.76 | 85.40 | 80.35 | 80.96 | 80.57 | 81.70 | 81.52 | 81.33 | 80.92 | 82.62 | 79.67 | 79.85 | 80.08 | 81.15 | 82.44 | 82.62 | 81.32 | 83.18 | 80.22 | 77.38 | 90.02 |
| | Qwen 72B | 82.99 | 82.62 | 83.32 | 85.58 | 85.21 | 84.47 | 84.84 | 86.51 | 82.86 | 82.81 | 83.13 | 83.92 | 84.47 | 84.10 | 83.86 | 85.58 | 84.10 | 84.66 | 84.19 | 84.66 | 84.29 | 84.47 | 84.01 | 85.40 | 83.55 | 81.44 | 92.24 |

Table 10: Critique-based refinement result for the (1, 9)-refinement setup. Random and Oracle for each judge are computed from the greedy seed response and nine responses generated during refinement. Random-RR and Oracle-RR are computed from the greedy seed response and nine sampled ones used in the reranking experiments.

**Graphical illustration.** Fig. 26 graphically depicts the refinement process, compared to the reranking process, and illustrates the definition of various metrics used in Sec. 4.4.

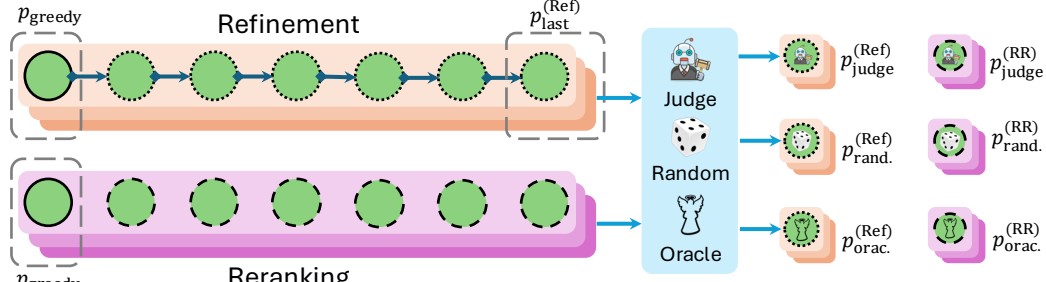

Figure 26: Illustration various metrics, all aggregated over the dataset. $p_{\text{greedy}}$ is calculated on the greedy response (seed in refinement and first in reranking), and $p_{\text{last}}^{(\text{Ref})}$ on the last refined response. The judge, a random selector and an oracle selects respective responses from each refinement and reranking process and compute their metrics.

**Qualitative case study.** Here, we provide examples of ineffective judge model critiques. We analyzed over 100 critique-response pairs and found two primary failure modes that point to lack of judge actionability: (1) False positives due to judges focusing on surface-level details rather than catching critical errors or (2) False negatives due to judges over scrutinizing stylistic errors over substance. Examples of false positive and false negative failure modes are presented in Fig. 27 and Fig. 28, respectively. In both cases, judges fail to assess based on validity and correctness of the response, offering feedback that does not appropriately progress or stop the refinement cycle.

**Dataset**: MATH
**Generator**: Qwen2.5-72B-Instruct
**Judge**: SFR-Judge-70B
**Refinement Round**: 1
**Type of Error**: False positive. Generator produces the wrong output, but the judge does not catch the mistake.

**Question:**
The $5 \times 5$ grid shown contains a collection of squares with sizes from $1 \times 1$ to $5 \times 5$. How many of these squares contain the black center square?

**Generator Output:**
## Step 1: Identify the sizes of squares that can contain the black center square
The black center square is located at the center of the $5 \times 5$ grid. We need to count the number of squares of different sizes that can contain this center square.

## Step 2: Count the number of squares of each size that contain the center square
- **$1 \times 1$ squares**: Only the center square itself contains the black center square. There is 1 such square.
- **$2 \times 2$ squares**: The center square can be part of 4 different $2 \times 2$ squares (one for each possible position of the top-left corner of the $2 \times 2$ square that includes the center).
- **$3 \times 3$ squares**: The center square can be part of 4 different $3 \times 3$ squares (one for each possible position of the top-left corner of the $3 \times 3$ square that includes the center).
- **$4 \times 4$ squares**: The center square can be part of 4 different $4 \times 4$ squares (one for each possible position of the top-left corner of the $4 \times 4$ square that includes the center).
- **$5 \times 5$ squares**: The entire $5 \times 5$ grid is the only $5 \times 5$ square, and it contains the center square. There is 1 such square.

## Step 3: Sum the number of squares of each size

$$1 + 4 + 4 + 4 + 1 = 14$$

Therefore, the final answer is: $\boxed{14}$. I hope it is correct.

**Judge Critique:**
The response precisely executes the instruction by following the required format for complex problems and providing a clear step-by-step solution. The explanation and calculations are concise and easy to follow. The response also accurately counts the number of squares of each size that contain the center square and correctly sums them up to obtain the final answer. The conclusion is also in line with the instruction.

**Ground-Truth Solution:**
All of the squares of size $5 \times 5$, $4 \times 4$, and $3 \times 3$ contain the black square and there are

$$1^2 + 2^2 + 3^2 = 14$$

of these. In addition, 4 of the $2 \times 2$ squares and 1 of the $1 \times 1$ squares contain the black square, for a total of $14 + 4 + 1 = \boxed{19}$.

Figure 27: Example of a false positive judge critique. The judge focuses on surface-level details, such as ease of reading, rather than verifying the logical errors in the model response. As a result, the refinement process stops without correcting the mistake.

**Dataset**: MATH
**Generator**: Llama-3.1-8B-Instruct
**Judge**: SFR-Judge-70B
**Refinement Round**: 1-6; 1 shown
**Type of Error**: False negative. Generator produces the correct output, but the judge overemphasizes details irrelevant to response correctness.

**Question:**
In the diagram below, we have $\sin \angle RPQ = \frac{7}{25}$. What is $\cos \angle RPS$? [Tikz omitted]

**Generator Output:**
## Step 1: We are given that $\sin \angle RPQ = \frac{7}{25}$ and we need to find $\cos \angle RPS$.

## Step 2: To approach this problem, we can use the relationship between sine and cosine functions in a right-angled triangle.

## Step 3: Since $\sin \angle RPQ = \frac{7}{25}$, we can use the Pythagorean identity to find $\cos \angle RPQ$.

## Step 4: The Pythagorean identity states that $\sin^2 \theta + \cos^2 \theta = 1$ for any angle $\theta$.

## Step 5: Therefore, we can find $\cos \angle RPQ$ by rearranging the Pythagorean identity: $\cos^2 \angle RPQ = 1 - \sin^2 \angle RPQ$.

## Step 6: Substituting the given value of $\sin \angle RPQ = \frac{7}{25}$, we get $\cos^2 \angle RPQ = 1 - \left(\frac{7}{25}\right)^2 = 1 - \frac{49}{625} = \frac{576}{625}$.

## Step 7: Taking the square root of both sides, we find $\cos \angle RPQ = \sqrt{\frac{576}{625}} = \frac{24}{25}$.

## Step 8: Now, we need to find $\cos \angle RPS$. Since $\angle RPS$ is the supplement of $\angle RPQ$, we can use the relationship $\cos(\theta + 180°) = -\cos \theta$.

## Step 9: Therefore, $\cos \angle RPS = -\cos \angle RPQ = -\frac{24}{25}$.

Therefore, the final answer is: $\boxed{-\dfrac{24}{25}}$.

**Judge Critique**:
The response precisely executes the instruction by providing a step-by-step solution to the problem. It clearly explains each step and uses the required format. The response is also accurate, as it correctly applies the Pythagorean identity and the relationship between sine and cosine functions in a right-angled triangle. Additionally, the response is harmless and does not contain any unnecessary information. However, the response contains more steps than necessary, as the problem can be solved in fewer steps.

**Ground-Truth Solution**:
For any angle $x$, we have $\cos(180° - x) = -\cos x$, so $\cos \angle RPS = \cos(180° - \angle RPQ) = -\cos \angle RPQ$.
Since $\sin^2 \angle RPQ + \cos^2 \angle RPQ = 1$, we have $\cos^2 \angle RPQ = 1 - \left(\frac{7}{25}\right)^2 = \frac{576}{625}$. Since $\angle RPQ$ is acute, we have $\cos \angle RPQ = \frac{24}{25}$, which gives us $\cos \angle RPS = -\cos \angle RPQ = \boxed{-\dfrac{24}{25}}$.

Figure 28: Example of a false negative judge critique. The judge focuses on surface-level details, such as response length, ignoring the fact that the outcome is correct.

# C   Tables of Normalized Results

Table 11-15 presents the normalized helpfulness and refinement ratio statistics for reranking, beam search and refinement.

| | | Prom 7B | SFR 8B | SC 8B | OB 8B | SFR 12B | Prom 8x7B | SFR 70B | SC 70B | STE 70B | Llama 8B | Best RM | Greedy | Random | Oracle | Maj |
|---|---|---|---|---|---|---|---|---|---|---|---|---|---|---|---|---|
| GSM8k | Llama 8B | -0.042 | 0.317 | 0.296 | 0.014 | 0.338 | -0.042 | 0.747 | 0.528 | 0.409 | 0.134 | 0.436 | 0.000 | -0.333 | 1.000 | 0.479 |
| | Llama 70B | 0.102 | 0.129 | 0.153 | -0.105 | 0.051 | -0.285 | 0.258 | 0.359 | -0.207 | 0.000 | 0.332 | 0.000 | -0.142 | 1.000 | 0.437 |
| | Qwen 32B | -0.066 | 0.296 | 0.093 | 0.138 | 0.114 | -0.114 | 0.207 | 0.159 | 0.000 | -0.021 | 0.251 | 0.000 | 0.000 | 1.000 | 0.251 |
| | Qwen 72B | -0.277 | -0.068 | -0.036 | 0.032 | -0.209 | -0.173 | 0.000 | 0.241 | 0.032 | -0.209 | 0.277 | 0.000 | -0.073 | 1.000 | 0.277 |
| | Qwen-M 7B | -0.168 | 0.110 | 0.139 | -0.168 | 0.165 | -0.249 | 0.194 | 0.194 | -0.029 | -0.139 | 0.194 | 0.000 | -1.282 | 1.000 | 0.249 |
| | Dpsk-M 7B | -0.036 | 0.396 | 0.448 | 0.099 | 0.391 | 0.005 | 0.740 | 0.635 | 0.422 | 0.218 | 0.471 | 0.000 | -0.084 | 1.000 | 0.448 |
| MATH | Llama 8B | -0.052 | 0.184 | 0.037 | 0.005 | 0.184 | 0.016 | 0.417 | 0.178 | 0.184 | 0.155 | 0.178 | 0.000 | -0.095 | 1.000 | 0.291 |
| | Llama 70B | -0.019 | 0.175 | 0.139 | -0.034 | 0.141 | 0.080 | 0.326 | 0.187 | 0.234 | 0.163 | 0.254 | 0.000 | -0.071 | 1.000 | 0.354 |
| | Qwen 32B | -0.111 | 0.093 | 0.039 | -0.168 | 0.140 | -0.097 | 0.330 | 0.129 | 0.118 | 0.043 | 0.106 | 0.000 | -0.231 | 1.000 | 0.262 |
| | Qwen 72B | -0.034 | 0.145 | 0.061 | -0.088 | 0.103 | -0.034 | 0.263 | 0.195 | 0.221 | 0.118 | 0.153 | 0.000 | -0.082 | 1.000 | 0.317 |
| | Qwen-M 7B | 0.028 | 0.269 | 0.099 | -0.066 | 0.241 | 0.049 | 0.385 | 0.247 | 0.187 | 0.241 | 0.159 | 0.000 | -0.808 | 1.000 | 0.280 |
| | Dpsk-M 7B | -0.093 | 0.172 | 0.098 | 0.084 | 0.101 | 0.045 | 0.386 | 0.217 | 0.267 | 0.122 | 0.207 | 0.000 | -0.057 | 1.000 | 0.246 |
| CHAMP | Llama 8B | 0.048 | 0.036 | 0.048 | 0.060 | 0.156 | 0.060 | 0.156 | 0.120 | 0.132 | 0.036 | 0.084 | 0.000 | -0.064 | 1.000 | - |
| | Llama 70B | -0.061 | 0.169 | 0.138 | 0.046 | 0.061 | -0.077 | 0.031 | 0.200 | 0.015 | 0.046 | 0.200 | 0.000 | -0.056 | 1.000 | - |
| | Qwen 32B | -0.500 | -0.393 | -0.750 | -0.215 | -0.643 | -0.608 | -0.465 | -0.608 | -0.465 | -0.715 | -0.375 | 0.000 | -0.436 | 1.000 | - |
| | Qwen 72B | -0.394 | -0.368 | -0.105 | -0.342 | -0.210 | -0.237 | -0.289 | -0.079 | -0.263 | -0.210 | -0.197 | 0.000 | -0.310 | 1.000 | - |
| | Qwen-M 7B | 0.118 | 0.392 | 0.275 | 0.216 | 0.255 | 0.236 | 0.510 | 0.373 | 0.392 | 0.196 | 0.373 | 0.000 | -0.884 | 1.000 | - |
| | Dpsk-M 7B | 0.016 | 0.355 | 0.266 | 0.177 | 0.282 | -0.016 | 0.363 | 0.298 | 0.290 | 0.097 | 0.331 | 0.000 | 0.054 | 1.000 | - |
| HumanEval+ | Llama 8B | -0.586 | -0.291 | 0.041 | -0.106 | -0.143 | -0.475 | 0.262 | 0.188 | 0.041 | -0.106 | 0.225 | 0.000 | -0.402 | 1.000 | - |
| | Llama 70B | -0.560 | -0.280 | 0.160 | 0.000 | 0.119 | -0.320 | 0.000 | 0.240 | 0.120 | 0.000 | 0.200 | 0.000 | -0.156 | 1.000 | - |
| | Qwen 32B | -0.100 | 0.100 | 0.000 | 0.250 | 0.100 | -0.100 | 0.000 | 0.200 | 0.000 | 0.000 | 0.350 | 0.000 | 0.120 | 1.000 | - |
| | Qwen 72B | 0.053 | 0.211 | 0.316 | 0.316 | 0.053 | -0.053 | 0.369 | 0.473 | 0.053 | 0.000 | 0.473 | 0.000 | -0.016 | 1.000 | - |
| | Qwen-C 7B | -1.585 | -0.584 | -0.751 | -0.918 | -0.584 | -1.168 | -0.334 | -0.417 | -0.501 | -0.584 | -0.584 | 0.000 | -0.694 | 1.000 | - |
| | Dpsk-C 6.7B | -0.375 | -0.167 | 0.208 | -0.292 | -0.292 | -0.375 | 0.374 | 0.333 | 0.167 | -0.167 | 0.375 | 0.000 | -0.296 | 1.000 | - |
| MBPP+ | Llama 8B | -0.313 | -0.012 | 0.133 | 0.204 | -0.097 | -0.241 | 0.108 | 0.373 | 0.349 | 0.096 | 0.289 | 0.000 | 0.032 | 1.000 | - |
| | Llama 70B | -0.544 | -0.353 | -0.044 | 0.014 | -0.500 | -0.500 | -0.265 | 0.235 | 0.059 | 0.014 | 0.184 | 0.000 | -0.078 | 1.000 | - |
| | Qwen 32B | -0.182 | 0.061 | 0.061 | 0.121 | -0.061 | -0.182 | 0.030 | 0.302 | 0.000 | -0.031 | 0.076 | 0.000 | 0.006 | 1.000 | - |
| | Qwen 72B | -0.188 | 0.063 | 0.032 | 0.000 | -0.125 | -0.093 | 0.125 | 0.093 | 0.032 | 0.218 | 0.063 | 0.000 | -0.057 | 1.000 | - |
| | Qwen-C 7B | -0.384 | -0.192 | -0.115 | -0.115 | -0.346 | -0.346 | 0.058 | 0.250 | 0.116 | -0.019 | 0.039 | 0.000 | -0.376 | 1.000 | - |
| | Dpsk-C 6.7B | -0.960 | -0.560 | -0.380 | -0.360 | -0.640 | -0.660 | -0.420 | -0.160 | -0.320 | -0.020 | -0.020 | 0.000 | -0.304 | 1.000 | - |
| BigCodeBench | Llama 8B | -0.216 | 0.031 | 0.063 | 0.042 | -0.017 | -0.133 | 0.174 | 0.219 | 0.181 | -0.052 | 0.136 | 0.000 | -0.145 | 1.000 | - |
| | Llama 70B | -0.291 | -0.189 | -0.148 | -0.184 | -0.219 | -0.316 | -0.163 | -0.112 | -0.107 | -0.123 | -0.090 | 0.000 | -0.205 | 1.000 | - |
| | Qwen 32B | -0.103 | 0.036 | -0.014 | -0.032 | 0.013 | -0.085 | 0.111 | 0.067 | 0.058 | -0.018 | 0.036 | 0.000 | -0.044 | 1.000 | - |
| | Qwen 72B | -0.065 | 0.013 | 0.058 | 0.090 | 0.071 | -0.072 | 0.064 | 0.045 | 0.071 | -0.065 | 0.143 | 0.000 | -0.027 | 1.000 | - |
| | Qwen-C 7B | -0.315 | -0.110 | -0.137 | -0.059 | -0.128 | -0.210 | 0.050 | 0.078 | 0.014 | -0.114 | 0.000 | 0.000 | -0.233 | 1.000 | - |
| | Dpsk-C 6.7B | -0.109 | 0.051 | 0.146 | 0.109 | 0.095 | -0.054 | 0.219 | 0.215 | 0.183 | 0.080 | 0.170 | 0.000 | -0.101 | 1.000 | - |
| AlpacaEval | Llama 8B | 0.276 | 0.356 | 0.350 | 0.251 | 0.456 | 0.361 | 0.478 | 0.381 | 0.373 | 0.170 | 0.421 | 0.000 | -0.017 | 1.000 | - |
| | Llama 70B | 0.399 | 0.356 | 0.280 | 0.209 | 0.395 | 0.368 | 0.431 | 0.319 | 0.386 | 0.178 | 0.363 | 0.000 | -0.046 | 1.000 | - |
| | Qwen 32B | 0.391 | 0.379 | 0.340 | 0.293 | 0.517 | 0.406 | 0.454 | 0.373 | 0.479 | 0.310 | 0.426 | 0.000 | 0.100 | 1.000 | - |
| | Qwen 72B | 0.283 | 0.417 | 0.162 | 0.259 | 0.496 | 0.347 | 0.477 | 0.384 | 0.450 | 0.274 | 0.430 | 0.000 | 0.194 | 1.000 | - |
| IFEval | Llama 8B | -0.371 | -0.173 | -0.185 | -0.062 | -0.062 | -0.259 | 0.124 | -0.100 | -0.049 | -0.198 | 0.012 | 0.000 | -0.337 | 1.000 | - |
| | Llama 70B | -0.273 | -0.319 | -0.432 | -0.523 | -0.319 | -0.523 | -0.160 | -0.250 | -0.160 | -0.501 | -0.205 | 0.000 | -0.379 | 1.000 | - |
| | Qwen 32B | 0.039 | 0.151 | 0.019 | -0.076 | -0.018 | 0.000 | 0.245 | 0.245 | 0.208 | -0.038 | 0.220 | 0.000 | -0.080 | 1.000 | - |
| | Qwen 72B | -0.087 | 0.131 | 0.108 | 0.021 | 0.152 | 0.000 | 0.174 | 0.218 | 0.044 | 0.195 | 0.293 | 0.000 | -0.015 | 1.000 | - |

Table 11: Normalized helpfulness of judge reranking (pairwise protocol), computed from raw data presented in Tab. 3.

| | | Prom 7B | SFR 8B | Thm 8B | SFR 12B | Prom 8x7B | SFR 70B | Llama 8B | OB$_{RM}$ 8B | S$_{RM}$ 8B | S$_{RM}$ 27B | Greedy | Random | Oracle | Maj |
|---|---|---|---|---|---|---|---|---|---|---|---|---|---|---|---|
| GSM8k | Llama 8B | -0.125 | 0.110 | -0.348 | 0.045 | -0.160 | 0.247 | -0.090 | 0.405 | 0.289 | 0.436 | 0.000 | -0.333 | 1.000 | 0.479 |
| | Llama 70B | 0.092 | 0.081 | -0.051 | 0.024 | -0.037 | -0.020 | 0.020 | 0.115 | 0.180 | 0.332 | 0.000 | -0.142 | 1.000 | 0.437 |
| | Qwen 32B | 0.069 | 0.129 | -0.045 | 0.036 | 0.024 | 0.075 | 0.039 | 0.251 | 0.207 | 0.171 | 0.000 | 0.000 | 1.000 | 0.251 |
| | Qwen 72B | 0.077 | -0.059 | -0.123 | -0.045 | -0.055 | -0.027 | 0.009 | 0.068 | 0.277 | 0.205 | 0.000 | -0.073 | 1.000 | 0.277 |
| | Qwen-M 7B | -0.524 | 0.077 | -1.205 | 0.073 | -0.352 | 0.154 | -0.150 | 0.110 | 0.139 | 0.194 | 0.000 | -1.282 | 1.000 | 0.249 |
| | Dpsk-M 7B | -0.005 | 0.122 | -0.094 | 0.124 | -0.008 | 0.423 | 0.020 | 0.339 | 0.464 | 0.471 | 0.000 | -0.084 | 1.000 | 0.448 |
| MATH | Llama 8B | -0.013 | 0.069 | -0.118 | 0.057 | -0.002 | 0.164 | -0.013 | 0.158 | 0.130 | 0.178 | 0.000 | -0.095 | 1.000 | 0.291 |
| | Llama 70B | 0.051 | 0.076 | -0.090 | 0.099 | 0.037 | 0.095 | 0.054 | 0.129 | 0.160 | 0.254 | 0.000 | -0.071 | 1.000 | 0.354 |
| | Qwen 32B | -0.110 | -0.122 | -0.237 | -0.060 | -0.149 | -0.100 | -0.097 | -0.054 | 0.106 | 0.032 | 0.000 | -0.231 | 1.000 | 0.262 |
| | Qwen 72B | -0.013 | 0.031 | -0.070 | 0.009 | -0.047 | 0.040 | -0.008 | 0.056 | 0.153 | 0.149 | 0.000 | -0.082 | 1.000 | 0.317 |
| | Qwen-M 7B | -0.769 | 0.070 | -0.737 | 0.068 | -0.340 | 0.101 | -0.164 | 0.107 | 0.096 | 0.159 | 0.000 | -0.808 | 1.000 | 0.280 |
| | Dpsk-M 7B | -0.050 | 0.109 | -0.082 | 0.041 | -0.019 | 0.218 | 0.057 | 0.180 | 0.207 | 0.202 | 0.000 | -0.057 | 1.000 | 0.246 |
| CHAMP | Llama 8B | -0.069 | -0.141 | -0.028 | 0.031 | 0.016 | 0.075 | 0.055 | 0.084 | 0.024 | 0.048 | 0.000 | -0.064 | 1.000 | - |
| | Llama 70B | -0.016 | 0.120 | -0.017 | 0.145 | 0.022 | 0.064 | 0.011 | 0.200 | 0.138 | 0.131 | 0.000 | -0.056 | 1.000 | - |
| | Qwen 32B | -0.484 | -0.531 | -0.422 | -0.398 | -0.472 | -0.422 | -0.562 | -0.536 | -0.626 | -0.375 | 0.000 | -0.436 | 1.000 | - |
| | Qwen 72B | -0.464 | -0.241 | -0.288 | -0.290 | -0.359 | -0.319 | -0.212 | -0.237 | -0.197 | -0.289 | 0.000 | -0.310 | 1.000 | - |
| | Qwen-M 7B | 0.080 | 0.155 | 0.183 | 0.228 | 0.066 | 0.290 | 0.165 | 0.255 | 0.373 | 0.275 | 0.000 | -0.884 | 1.000 | - |
| | Dpsk-M 7B | 0.009 | 0.120 | 0.062 | 0.146 | 0.092 | 0.279 | 0.123 | 0.286 | 0.331 | 0.185 | 0.000 | 0.054 | 1.000 | - |
| Humaneval+ | Llama 8B | -0.287 | -0.109 | -0.382 | -0.114 | -0.355 | 0.092 | -0.351 | -0.180 | 0.059 | 0.225 | 0.000 | -0.402 | 1.000 | - |
| | Llama 70B | -0.056 | -0.179 | -0.130 | -0.160 | -0.147 | -0.098 | -0.116 | 0.120 | 0.200 | 0.120 | 0.000 | -0.156 | 1.000 | - |
| | Qwen 32B | 0.148 | 0.016 | 0.098 | 0.117 | 0.115 | 0.308 | 0.057 | 0.300 | 0.050 | 0.350 | 0.000 | 0.120 | 1.000 | - |
| | Qwen 72B | -0.027 | 0.275 | -0.013 | 0.189 | -0.019 | 0.198 | -0.039 | 0.158 | 0.211 | 0.473 | 0.000 | -0.016 | 1.000 | - |
| | Qwen-C 7B | -0.654 | -0.635 | -0.659 | -0.653 | -0.640 | -0.274 | -0.655 | -0.668 | -1.001 | -0.584 | 0.000 | -0.694 | 1.000 | - |
| | Dpsk-C 6.7B | -0.273 | -0.135 | -0.241 | -0.059 | -0.270 | 0.083 | -0.156 | 0.042 | 0.125 | 0.375 | 0.000 | -0.296 | 1.000 | - |
| MBPP+ | Llama 8B | 0.102 | -0.014 | 0.066 | 0.122 | 0.087 | 0.283 | 0.107 | 0.265 | 0.289 | 0.108 | 0.000 | 0.032 | 1.000 | - |
| | Llama 70B | -0.009 | -0.264 | -0.069 | -0.182 | -0.046 | -0.004 | -0.019 | -0.029 | 0.184 | -0.103 | 0.000 | -0.078 | 1.000 | - |
| | Qwen 32B | -0.053 | 0.002 | 0.042 | 0.000 | 0.001 | 0.022 | 0.050 | -0.031 | 0.076 | 0.000 | 0.000 | 0.006 | 1.000 | - |
| | Qwen 72B | 0.014 | 0.090 | -0.014 | 0.043 | -0.009 | -0.021 | 0.058 | 0.063 | -0.110 | 0.032 | 0.000 | -0.057 | 1.000 | - |
| | Qwen-C 7B | -0.273 | -0.288 | -0.328 | -0.175 | -0.287 | -0.054 | -0.172 | -0.019 | 0.039 | -0.009 | 0.000 | -0.376 | 1.000 | - |
| | Dpsk-C 6.7B | -0.293 | -0.258 | -0.265 | -0.304 | -0.277 | -0.320 | -0.197 | -0.190 | -0.020 | -0.600 | 0.000 | -0.304 | 1.000 | - |
| BigCodeBench | Llama 8B | -0.079 | -0.041 | -0.108 | 0.013 | -0.072 | 0.101 | -0.049 | 0.038 | 0.131 | 0.136 | 0.000 | -0.145 | 1.000 | - |
| | Llama 70B | -0.179 | -0.156 | -0.168 | -0.149 | -0.158 | -0.124 | -0.157 | -0.105 | -0.090 | -0.092 | 0.000 | -0.205 | 1.000 | - |
| | Qwen 32B | -0.026 | 0.004 | -0.042 | -0.005 | -0.031 | 0.014 | 0.006 | 0.026 | 0.027 | 0.036 | 0.000 | -0.044 | 1.000 | - |
| | Qwen 72B | 0.007 | 0.019 | -0.025 | 0.044 | -0.022 | 0.058 | -0.003 | 0.016 | 0.143 | 0.123 | 0.000 | -0.027 | 1.000 | - |
| | Qwen-C 7B | -0.196 | -0.126 | -0.222 | -0.105 | -0.199 | 0.008 | -0.150 | -0.069 | 0.000 | -0.016 | 0.000 | -0.233 | 1.000 | - |
| | Dpsk-C 6.7B | -0.029 | -0.005 | -0.077 | 0.031 | -0.039 | 0.156 | 0.017 | 0.080 | 0.170 | 0.164 | 0.000 | -0.101 | 1.000 | - |
| AlpacaEval | Llama 8B | 0.042 | 0.097 | 0.019 | 0.081 | 0.014 | 0.078 | 0.004 | 0.229 | 0.316 | 0.421 | 0.000 | -0.017 | 1.000 | - |
| | Llama 70B | 0.025 | 0.052 | -0.018 | 0.040 | 0.001 | 0.005 | -0.056 | 0.137 | 0.249 | 0.363 | 0.000 | -0.046 | 1.000 | - |
| | Qwen 32B | 0.145 | 0.145 | 0.121 | 0.149 | 0.113 | 0.126 | 0.106 | 0.221 | 0.330 | 0.426 | 0.000 | 0.100 | 1.000 | - |
| | Qwen 72B | 0.098 | 0.112 | 0.073 | 0.122 | 0.083 | 0.103 | 0.084 | 0.161 | 0.340 | 0.430 | 0.000 | 0.194 | 1.000 | - |
| IFEval | Llama 8B | -0.333 | -0.166 | -0.247 | -0.147 | -0.246 | -0.069 | -0.162 | -0.025 | -0.118 | 0.012 | 0.000 | -0.428 | 1.000 | - |
| | Llama 70B | -0.250 | -0.248 | -0.250 | -0.246 | -0.283 | -0.193 | -0.349 | -0.386 | -0.205 | -0.295 | 0.000 | -0.535 | 1.000 | - |
| | Qwen 32B | 0.018 | -0.003 | 0.008 | -0.052 | -0.034 | 0.247 | 0.074 | 0.220 | 0.019 | 0.122 | 0.000 | -0.290 | 1.000 | - |
| | Qwen 72B | -0.035 | 0.020 | -0.030 | 0.085 | 0.009 | 0.152 | 0.041 | 0.117 | 0.287 | -0.010 | 0.000 | -0.243 | 1.000 | - |

Table 12: Normalized helpfulness of judge reranking (single-rating protocol with Likert scale prompt), computed from raw data presented in Tab. 4.

| | | Prom 7B | SFR 8B | Thm 8B | SFR 12B | Prom 8x7B | SFR 70B | Llama 8B | OB$_{RM}$ 8B | S$_{RM}$ 8B | S$_{RM}$ 27B | Greedy | Random | Oracle | Maj |
|---|---|---|---|---|---|---|---|---|---|---|---|---|---|---|---|
| GSM8k | Llama 8B | -0.222 | -0.020 | -0.331 | -0.032 | 0.002 | 0.078 | -0.063 | 0.405 | 0.289 | 0.436 | 0.000 | -0.333 | 1.000 | 0.479 |
| | Llama 70B | 0.034 | 0.085 | 0.000 | 0.061 | 0.119 | -0.061 | 0.051 | 0.115 | 0.180 | 0.332 | 0.000 | -0.142 | 1.000 | 0.437 |
| | Qwen 32B | 0.012 | 0.075 | -0.075 | 0.030 | 0.051 | 0.015 | 0.081 | 0.251 | 0.207 | 0.171 | 0.000 | 0.000 | 1.000 | 0.251 |
| | Qwen 72B | -0.050 | -0.059 | -0.241 | -0.068 | -0.005 | -0.082 | -0.250 | 0.068 | 0.277 | 0.205 | 0.000 | -0.073 | 1.000 | 0.277 |
| | Qwen-M 7B | -1.297 | 0.037 | -1.209 | -0.004 | -0.502 | 0.070 | -0.037 | 0.110 | 0.139 | 0.194 | 0.000 | -1.282 | 1.000 | 0.249 |
| | Dpsk-M 7B | -0.071 | 0.168 | -0.069 | 0.016 | 0.099 | 0.299 | 0.026 | 0.339 | 0.464 | 0.471 | 0.000 | -0.084 | 1.000 | 0.448 |
| MATH | Llama 8B | -0.066 | 0.016 | -0.122 | 0.044 | -0.025 | 0.079 | -0.004 | 0.158 | 0.130 | 0.178 | 0.000 | -0.095 | 1.000 | 0.291 |
| | Llama 70B | -0.019 | 0.077 | -0.052 | 0.100 | 0.018 | 0.048 | 0.090 | 0.129 | 0.160 | 0.254 | 0.000 | -0.071 | 1.000 | 0.354 |
| | Qwen 32B | -0.142 | -0.124 | -0.251 | -0.092 | -0.095 | -0.168 | -0.124 | -0.054 | 0.106 | 0.032 | 0.000 | -0.231 | 1.000 | 0.262 |
| | Qwen 72B | -0.065 | -0.001 | -0.058 | -0.008 | 0.005 | -0.027 | -0.025 | 0.056 | 0.153 | 0.149 | 0.000 | -0.082 | 1.000 | 0.317 |
| | Qwen-M 7B | -1.185 | 0.049 | -0.852 | 0.075 | -0.391 | 0.074 | -0.186 | 0.107 | 0.096 | 0.159 | 0.000 | -0.808 | 1.000 | 0.280 |
| | Dpsk-M 7B | -0.011 | 0.049 | -0.077 | -0.007 | 0.004 | 0.133 | -0.001 | 0.180 | 0.207 | 0.202 | 0.000 | -0.057 | 1.000 | 0.246 |
| CHAMP | Llama 8B | 0.017 | -0.048 | -0.017 | 0.091 | 0.076 | 0.062 | 0.047 | 0.084 | 0.024 | 0.048 | 0.000 | -0.064 | 1.000 | - |
| | Llama 70B | 0.023 | 0.101 | -0.009 | 0.083 | 0.084 | 0.000 | -0.043 | 0.200 | 0.138 | 0.131 | 0.000 | -0.056 | 1.000 | - |
| | Qwen 32B | -0.535 | -0.441 | -0.419 | -0.451 | -0.455 | -0.441 | -0.370 | -0.536 | -0.626 | -0.375 | 0.000 | -0.436 | 1.000 | - |
| | Qwen 72B | -0.295 | -0.262 | -0.348 | -0.308 | -0.266 | -0.249 | -0.362 | -0.237 | -0.197 | -0.289 | 0.000 | -0.310 | 1.000 | - |
| | Qwen-M 7B | -0.052 | 0.272 | 0.222 | 0.208 | 0.224 | 0.237 | 0.091 | 0.255 | 0.373 | 0.275 | 0.000 | -0.884 | 1.000 | - |
| | Dpsk-M 7B | 0.027 | 0.168 | 0.067 | 0.121 | 0.072 | 0.236 | 0.055 | 0.286 | 0.331 | 0.185 | 0.000 | 0.054 | 1.000 | - |
| Humaneval+ | Llama 8B | -0.364 | -0.292 | -0.387 | -0.172 | -0.337 | -0.097 | -0.227 | -0.180 | 0.059 | 0.225 | 0.000 | -0.402 | 1.000 | - |
| | Llama 70B | -0.124 | -0.140 | -0.146 | -0.181 | -0.149 | -0.110 | -0.209 | 0.120 | 0.200 | 0.120 | 0.000 | -0.156 | 1.000 | - |
| | Qwen 32B | 0.160 | -0.171 | 0.117 | 0.121 | 0.123 | 0.116 | -0.008 | 0.300 | 0.050 | 0.350 | 0.000 | 0.120 | 1.000 | - |
| | Qwen 72B | -0.003 | 0.147 | -0.014 | 0.196 | -0.022 | 0.111 | 0.109 | 0.158 | 0.211 | 0.473 | 0.000 | -0.016 | 1.000 | - |
| | Qwen-C 7B | -0.633 | -0.601 | -0.637 | -0.631 | -0.588 | -0.575 | -0.851 | -0.668 | -1.001 | -0.584 | 0.000 | -0.694 | 1.000 | - |
| | Dpsk-C 6.7B | -0.383 | -0.213 | -0.273 | -0.158 | -0.273 | -0.027 | -0.426 | 0.042 | 0.125 | 0.375 | 0.000 | -0.296 | 1.000 | - |
| MBPP+ | Llama 8B | 0.163 | 0.055 | 0.060 | 0.103 | 0.094 | 0.265 | 0.105 | 0.265 | 0.289 | 0.108 | 0.000 | 0.032 | 1.000 | - |
| | Llama 70B | -0.013 | -0.136 | -0.066 | -0.166 | -0.053 | -0.048 | -0.149 | -0.029 | 0.184 | -0.103 | 0.000 | -0.078 | 1.000 | - |
| | Qwen 32B | -0.027 | -0.002 | 0.030 | -0.076 | 0.013 | -0.001 | 0.031 | -0.031 | 0.076 | 0.000 | 0.000 | 0.006 | 1.000 | - |
| | Qwen 72B | 0.032 | 0.015 | -0.031 | -0.031 | 0.034 | 0.078 | 0.135 | 0.063 | -0.110 | 0.032 | 0.000 | -0.057 | 1.000 | - |
| | Qwen-C 7B | -0.203 | -0.285 | -0.352 | -0.112 | -0.271 | 0.010 | -0.356 | -0.019 | 0.039 | -0.009 | 0.000 | -0.376 | 1.000 | - |
| | Dpsk-C 6.7B | -0.156 | -0.426 | -0.262 | -0.377 | -0.275 | -0.232 | -0.276 | -0.190 | -0.020 | -0.600 | 0.000 | -0.304 | 1.000 | - |
| BigCodeBench | Llama 8B | -0.087 | -0.050 | -0.111 | -0.019 | -0.070 | 0.024 | -0.075 | 0.038 | 0.131 | 0.136 | 0.000 | -0.145 | 1.000 | - |
| | Llama 70B | -0.162 | -0.184 | -0.168 | -0.204 | -0.162 | -0.149 | -0.158 | -0.105 | -0.090 | -0.092 | 0.000 | -0.205 | 1.000 | - |
| | Qwen 32B | -0.039 | 0.005 | -0.039 | 0.015 | 0.000 | -0.011 | 0.003 | 0.026 | 0.027 | 0.036 | 0.000 | -0.044 | 1.000 | - |
| | Qwen 72B | -0.031 | 0.027 | -0.026 | 0.039 | 0.021 | 0.024 | 0.009 | 0.016 | 0.143 | 0.123 | 0.000 | -0.027 | 1.000 | - |
| | Qwen-C 7B | -0.229 | -0.124 | -0.224 | -0.144 | -0.171 | -0.068 | -0.157 | -0.069 | 0.000 | -0.016 | 0.000 | -0.233 | 1.000 | - |
| | Dpsk-C 6.7B | -0.054 | -0.045 | -0.084 | 0.012 | -0.010 | 0.114 | -0.053 | 0.080 | 0.170 | 0.164 | 0.000 | -0.101 | 1.000 | - |
| AlpacaEval | Llama 8B | 0.063 | 0.127 | 0.022 | 0.104 | 0.055 | 0.074 | 0.065 | 0.229 | 0.316 | 0.421 | 0.000 | -0.017 | 1.000 | - |
| | Llama 70B | 0.022 | 0.103 | 0.006 | 0.042 | 0.038 | 0.026 | 0.032 | 0.137 | 0.249 | 0.363 | 0.000 | -0.046 | 1.000 | - |
| | Qwen 32B | 0.144 | 0.179 | 0.119 | 0.163 | 0.169 | 0.136 | 0.145 | 0.221 | 0.330 | 0.426 | 0.000 | 0.100 | 1.000 | - |
| | Qwen 72B | 0.129 | 0.157 | 0.078 | 0.124 | 0.129 | 0.120 | 0.123 | 0.161 | 0.340 | 0.430 | 0.000 | 0.194 | 1.000 | - |
| IFEval | Llama 8B | -0.323 | -0.231 | -0.248 | -0.223 | -0.206 | -0.157 | -0.234 | -0.025 | -0.118 | 0.012 | 0.000 | -0.428 | 1.000 | - |
| | Llama 70B | -0.343 | -0.412 | -0.287 | -0.257 | -0.424 | -0.239 | -0.424 | -0.386 | -0.205 | -0.295 | 0.000 | -0.535 | 1.000 | - |
| | Qwen 32B | -0.090 | -0.026 | -0.029 | -0.078 | -0.005 | -0.001 | -0.082 | 0.220 | 0.019 | 0.122 | 0.000 | -0.290 | 1.000 | - |
| | Qwen 72B | 0.066 | 0.000 | -0.041 | 0.040 | -0.133 | 0.068 | -0.062 | 0.117 | 0.287 | -0.010 | 0.000 | -0.243 | 1.000 | - |

Table 13: Normalized helpfulness of judge reranking (single-rating protocol with additive prompt), computed from raw data presented in Tab. 5.

| | | Lookahead | Prom 7B | SFR 8B | SC 8B | OB 8B | Thm 8B | SFR 12B | Prom 8x7B | SFR 70B | SC 70B | STE 70B | Q$_{PRM}$ 7B | Greedy | Random | Oracle |
|---|---|---|---|---|---|---|---|---|---|---|---|---|---|---|---|---|
| GSM8k | Llama 8B | ✗ | -0.117 | 0.142 | 0.129 | 0.006 | -0.110 | 0.178 | -0.135 | 0.405 | 0.392 | 0.172 | 0.540 | 0.000 | -0.386 | 1.000 |
| | | ✓ | -0.282 | 0.221 | 0.307 | 0.019 | -0.086 | 0.202 | -0.276 | 0.565 | 0.405 | 0.301 | 0.595 | | | |
| | Llama 70B | ✗ | -0.175 | -0.079 | 0.135 | -0.058 | 0.056 | 0.018 | -0.310 | 0.018 | 0.287 | -0.058 | 0.404 | 0.000 | -0.287 | 1.000 |
| | | ✓ | -0.231 | 0.114 | 0.173 | 0.056 | -0.058 | 0.135 | -0.231 | 0.135 | 0.287 | -0.079 | 0.383 | | | |
| | Qwen 32B | ✗ | -0.060 | 0.155 | 0.098 | 0.155 | -0.021 | 0.018 | -0.021 | -0.021 | 0.117 | -0.080 | 0.510 | 0.000 | -0.031 | 1.000 |
| | | ✓ | -0.021 | 0.137 | -0.021 | 0.197 | -0.080 | 0.275 | -0.158 | 0.313 | 0.275 | 0.060 | 0.472 | | | |
| | Qwen 72B | ✗ | -0.174 | -0.200 | -0.117 | 0.026 | 0.057 | -0.230 | -0.287 | 0.000 | 0.000 | -0.117 | 0.257 | 0.000 | -0.094 | 1.000 |
| | | ✓ | -0.200 | -0.374 | 0.143 | | -0.087 | -0.117 | -0.260 | 0.113 | 0.143 | -0.060 | 0.543 | | | |
| | Qwen-M 7B | ✗ | -0.138 | 0.091 | 0.154 | 0.077 | 0.154 | 0.185 | -0.077 | 0.168 | 0.185 | 0.122 | 0.306 | 0.000 | -0.665 | 1.000 |
| | | ✓ | -0.262 | 0.030 | 0.091 | 0.061 | 0.122 | 0.168 | -0.124 | 0.262 | 0.185 | 0.138 | 0.306 | | | |
| | Dpsk-M 7B | ✗ | -0.126 | 0.109 | 0.126 | -0.017 | -0.078 | 0.113 | -0.022 | 0.431 | 0.365 | 0.257 | 0.613 | 0.000 | -0.097 | 1.000 |
| | | ✓ | -0.056 | 0.361 | 0.309 | 0.153 | -0.048 | 0.343 | 0.118 | 0.591 | 0.496 | 0.404 | 0.600 | | | |
| MATH | Llama 8B | ✗ | -0.026 | 0.064 | 0.013 | -0.010 | -0.013 | 0.038 | 0.010 | 0.148 | 0.079 | 0.056 | 0.270 | 0.000 | -0.035 | 1.000 |
| | | ✓ | 0.003 | 0.087 | 0.050 | 0.027 | 0.012 | 0.081 | 0.026 | 0.239 | 0.126 | 0.065 | 0.356 | | | |
| | Llama 70B | ✗ | -0.003 | 0.019 | 0.037 | 0.022 | 0.009 | 0.062 | 0.018 | 0.092 | 0.022 | 0.032 | 0.236 | 0.000 | -0.028 | 1.000 |
| | | ✓ | -0.003 | 0.120 | 0.069 | 0.048 | -0.013 | 0.098 | 0.023 | 0.192 | 0.097 | 0.050 | 0.336 | | | |
| | Qwen 32B | ✗ | -0.085 | -0.006 | -0.068 | -0.077 | -0.135 | -0.015 | -0.100 | 0.056 | 0.012 | -0.073 | 0.245 | 0.000 | -0.098 | 1.000 |
| | | ✓ | -0.120 | 0.037 | 0.087 | -0.035 | -0.100 | 0.079 | -0.060 | 0.207 | 0.127 | 0.052 | 0.322 | | | |
| | Qwen 72B | ✗ | -0.024 | 0.063 | 0.054 | 0.022 | -0.032 | 0.071 | 0.039 | 0.100 | 0.085 | 0.007 | 0.257 | 0.000 | -0.053 | 1.000 |
| | | ✓ | -0.039 | 0.073 | 0.085 | -0.054 | -0.012 | 0.081 | -0.019 | 0.169 | 0.127 | 0.098 | 0.320 | | | |
| | Qwen-M 7B | ✗ | -0.022 | 0.056 | 0.084 | 0.021 | 0.002 | 0.100 | 0.035 | -0.030 | 0.113 | 0.065 | 0.232 | 0.000 | -0.302 | 1.000 |
| | | ✓ | -0.067 | 0.105 | 0.100 | 0.008 | 0.051 | 0.151 | 0.027 | 0.296 | 0.175 | 0.175 | 0.372 | | | |
| | Dpsk-M 7B | ✗ | -0.042 | 0.007 | 0.006 | 0.011 | -0.024 | 0.003 | -0.023 | 0.102 | 0.084 | 0.029 | 0.231 | 0.000 | -0.029 | 1.000 |
| | | ✓ | -0.038 | 0.059 | 0.084 | 0.057 | -0.001 | 0.086 | 0.034 | 0.210 | 0.148 | 0.106 | 0.315 | | | |
| CHAMP | Llama 8B | ✗ | 0.026 | 0.106 | 0.026 | 0.079 | 0.000 | 0.106 | 0.005 | 0.076 | 0.143 | 0.074 | 0.159 | 0.000 | 0.006 | 1.000 |
| | | ✓ | -0.011 | 0.127 | 0.111 | 0.090 | -0.005 | 0.116 | 0.037 | 0.212 | 0.148 | 0.095 | 0.233 | | | |
| | Llama 70B | ✗ | -0.205 | -0.008 | -0.031 | 0.024 | -0.008 | -0.094 | -0.165 | 0.039 | 0.000 | -0.047 | 0.110 | 0.000 | -0.132 | 100.0 |
| | | ✓ | -0.134 | 0.039 | 0.016 | 0.118 | -0.055 | -0.039 | -0.126 | 0.039 | 0.063 | 0.000 | 0.205 | | | |
| | Qwen 32B | ✗ | -0.138 | -0.092 | -0.046 | 0.046 | -0.015 | -0.138 | -0.092 | 0.061 | 0.108 | -0.077 | 0.061 | 0.000 | -0.033 | 1.000 |
| | | ✓ | -0.169 | -0.031 | -0.031 | 0.046 | 0.015 | 0.015 | -0.123 | 0.216 | 0.046 | -0.031 | 0.231 | | | |
| | Qwen 72B | ✗ | 0.030 | -0.149 | -0.045 | -0.089 | -0.015 | -0.134 | 0.075 | 0.120 | 0.015 | 0.000 | 0.164 | 0.000 | -0.048 | 1.000 |
| | | ✓ | -0.149 | -0.104 | -0.075 | 0.000 | -0.089 | -0.149 | -0.104 | 0.075 | 0.000 | 0.060 | 0.105 | | | |
| | Qwen-M 7B | ✗ | 0.023 | 0.081 | 0.069 | 0.058 | 0.195 | 0.069 | 0.081 | 0.104 | 0.092 | 0.035 | 0.184 | 0.000 | -0.476 | 1.000 |
| | | ✓ | -0.241 | 0.035 | 0.000 | 0.000 | -0.023 | -0.011 | -0.046 | 0.287 | 0.161 | 0.127 | 0.357 | | | |
| | Dpsk-M 7B | ✗ | -0.049 | 0.081 | 0.016 | 0.059 | 0.027 | 0.011 | 0.000 | 0.200 | 0.124 | 0.135 | 0.249 | 0.000 | 0.010 | 1.000 |
| | | ✓ | -0.065 | 0.124 | 0.054 | 0.146 | 0.086 | 0.130 | 0.011 | 0.243 | 0.168 | 0.151 | 0.335 | | | |
| HumanEval+ | Llama 8B | ✗ | -0.041 | 0.204 | 0.020 | 0.143 | -0.061 | 0.122 | 0.041 | 0.224 | 0.082 | 0.204 | 0.041 | 0.000 | -0.054 | 1.000 |
| | | ✓ | -0.020 | 0.102 | 0.041 | 0.102 | 0.000 | 0.224 | 0.000 | 0.224 | 0.224 | 0.204 | 0.265 | | | |
| | Llama 70B | ✗ | -0.334 | -0.067 | 0.133 | -0.233 | -0.133 | -0.167 | -0.267 | 0.000 | -0.033 | -0.067 | 0.167 | 0.000 | -0.138 | 1.000 |
| | | ✓ | -0.233 | -0.033 | 0.033 | -0.067 | -0.067 | -0.033 | -0.133 | -0.200 | 0.167 | 0.133 | 0.133 | | | |
| | Qwen 32B | ✗ | 0.000 | 0.000 | 0.191 | -0.048 | 0.095 | 0.286 | -0.048 | 0.238 | 0.286 | 0.238 | 0.095 | 0.000 | -0.030 | 1.000 |
| | | ✓ | -0.048 | -0.238 | 0.048 | -0.095 | 0.143 | 0.191 | -0.191 | 0.191 | 0.381 | 0.191 | -0.048 | | | |
| | Qwen 72B | ✗ | 0.053 | 0.105 | 0.000 | 0.053 | 0.105 | 0.158 | 0.000 | 0.158 | 0.158 | 0.158 | 0.105 | 0.000 | -0.009 | 1.000 |
| | | ✓ | -0.053 | -0.105 | 0.053 | 0.053 | 0.105 | 0.211 | 0.053 | 0.158 | 0.211 | 0.211 | 0.211 | | | |
| | Qwen-C 7B | ✗ | -0.500 | -0.500 | -0.222 | -0.222 | -0.056 | -0.278 | -0.500 | -0.222 | -0.111 | -0.167 | 0.167 | 0.000 | -0.367 | 1.000 |
| | | ✓ | -0.556 | -0.389 | -0.278 | -0.222 | -0.500 | 0.000 | -0.278 | -0.389 | -0.167 | -0.445 | -0.167 | | | |
| | Dpsk-C 6.7B | ✗ | -0.314 | -0.143 | -0.229 | -0.229 | -0.257 | -0.172 | -0.286 | -0.086 | 0.057 | -0.143 | 0.057 | 0.000 | -0.214 | 1.000 |
| | | ✓ | -0.286 | 0.000 | 0.114 | 0.000 | -0.029 | 0.057 | -0.229 | 0.343 | 0.114 | 0.172 | 0.314 | | | |
| MBPP+ | Llama 8B | ✗ | -0.022 | 0.000 | 0.043 | 0.065 | 0.022 | -0.054 | -0.140 | 0.011 | 0.215 | 0.108 | 0.065 | 0.000 | -0.098 | 1.000 |
| | | ✓ | -0.150 | -0.172 | 0.033 | 0.140 | -0.032 | -0.097 | -0.247 | 0.076 | 0.215 | 0.129 | 0.097 | | | |
| | Llama 70B | ✗ | -0.153 | -0.165 | 0.012 | -0.059 | -0.094 | -0.047 | -0.271 | -0.071 | 0.082 | -0.012 | 0.094 | 0.000 | -0.111 | 1.000 |
| | | ✓ | -0.259 | -0.153 | 0.012 | 0.000 | -0.212 | -0.071 | -0.212 | -0.235 | -0.059 | 0.012 | 0.012 | | | |
| | Qwen 32B | ✗ | 0.000 | 0.123 | -0.082 | 0.082 | 0.062 | 0.021 | 0.082 | 0.184 | 0.164 | 0.082 | 0.041 | 0.000 | -0.089 | 1.000 |
| | | ✓ | -0.102 | 0.143 | -0.102 | -0.020 | -0.041 | -0.041 | -0.082 | 0.143 | 0.062 | 0.082 | 0.021 | | | |
| | Qwen 72B | ✗ | -0.054 | -0.028 | 0.027 | 0.000 | 0.081 | 0.081 | -0.054 | 0.135 | 0.053 | 0.081 | 0.081 | 0.000 | -0.080 | 1.000 |
| | | ✓ | 0.027 | 0.053 | 0.000 | -0.082 | -0.028 | 0.053 | 0.000 | 0.107 | -0.054 | 0.053 | -0.108 | | | |
| | Qwen-C 7B | ✗ | -0.085 | 0.057 | 0.129 | 0.015 | -0.071 | -0.014 | 0.000 | 0.114 | 0.172 | 0.072 | 0.129 | 0.000 | -0.377 | 1.000 |
| | | ✓ | -0.171 | -0.028 | 0.015 | -0.157 | -0.100 | -0.114 | -0.129 | 0.100 | 0.157 | -0.014 | -0.071 | | | |
| | Dpsk-C 6.7B | ✗ | -0.246 | -0.087 | -0.290 | -0.261 | -0.333 | -0.087 | -0.145 | 0.072 | -0.101 | 0.072 | -0.043 | 0.000 | -0.278 | 1.000 |
| | | ✓ | -0.319 | -0.232 | -0.087 | -0.145 | -0.145 | -0.188 | -0.290 | -0.145 | 0.029 | -0.101 | -0.087 | | | |
| BigCodeBench | Llama 8B | ✗ | -0.129 | -0.087 | -0.060 | -0.050 | -0.113 | -0.076 | -0.145 | 0.016 | 0.024 | 0.005 | -0.016 | 0.000 | -0.134 | 1.000 |
| | | ✓ | -0.208 | 0.005 | 0.005 | -0.042 | -0.100 | -0.029 | -0.140 | 0.082 | 0.021 | 0.053 | 0.011 | | | |
| | Llama 70B | ✗ | -0.128 | -0.088 | -0.062 | -0.015 | 0.029 | -0.029 | -0.109 | -0.033 | 0.025 | -0.004 | -0.029 | 0.000 | -0.045 | 1.000 |
| | | ✓ | -0.146 | -0.055 | 0.058 | -0.007 | 0.004 | -0.069 | -0.084 | -0.025 | 0.073 | 0.015 | -0.025 | | | |
| | Qwen 32B | ✗ | -0.046 | -0.032 | 0.025 | 0.053 | 0.035 | 0.028 | 0.014 | 0.049 | 0.081 | 0.053 | 0.028 | 0.000 | -0.008 | 1.000 |
| | | ✓ | -0.066 | 0.018 | 0.046 | 0.060 | 0.004 | -0.021 | -0.028 | 0.042 | 0.046 | 0.060 | 0.039 | | | |
| | Qwen 72B | ✗ | -0.024 | -0.024 | 0.053 | 0.044 | 0.034 | 0.024 | -0.005 | 0.029 | 0.058 | 0.000 | 0.019 | 0.000 | 0.013 | 1.000 |
| | | ✓ | -0.010 | 0.024 | 0.058 | 0.082 | 0.019 | 0.018 | 0.005 | 0.029 | 0.053 | 0.029 | 0.009 | | | |
| | Qwen-C 7B | ✗ | -0.139 | -0.033 | -0.012 | 0.000 | -0.106 | -0.064 | -0.106 | -0.012 | 0.027 | -0.024 | 0.042 | 0.000 | -0.098 | 1.000 |
| | | ✓ | -0.179 | -0.024 | 0.024 | 0.000 | -0.027 | -0.012 | -0.103 | 0.046 | 0.067 | 0.015 | 0.030 | | | |
| | Dpsk-C 6.7B | ✗ | -0.146 | 0.006 | 0.000 | 0.006 | -0.061 | -0.023 | -0.096 | 0.058 | 0.041 | 0.050 | 0.015 | 0.000 | -0.119 | 1.000 |
| | | ✓ | -0.169 | -0.032 | 0.015 | 0.000 | -0.053 | -0.026 | -0.096 | 0.064 | 0.038 | 0.093 | -0.009 | | | |

Table 14: Normalized helpfulness of judge beam search, computed from raw data presented in Tab. 9.

| | | Prom 7B | | | SFR 8B | | | Thm 8B | | | SFR 12B | | | Prom8x7B | | | SFR 70B | | |
|---|---|---|---|---|---|---|---|---|---|---|---|---|---|---|---|---|---|---|---|
| | | $\delta^{(RR)}$ | $\delta^{(G)}$ | $\delta^{(Eff)}$ | $\delta^{(RR)}$ | $\delta^{(G)}$ | $\delta^{(Eff)}$ | $\delta^{(RR)}$ | $\delta^{(G)}$ | $\delta^{(Eff)}$ | $\delta^{(RR)}$ | $\delta^{(G)}$ | $\delta^{(Eff)}$ | $\delta^{(RR)}$ | $\delta^{(G)}$ | $\delta^{(Eff)}$ | $\delta^{(RR)}$ | $\delta^{(G)}$ | $\delta^{(Eff)}$ |
| GSM8k | Llama 8B | 0.998 | 0.993 | 0.993 | 0.960 | 0.998 | 0.960 | 1.043 | 1.000 | 1.000 | 0.962 | 1.003 | 0.962 | 0.995 | 0.989 | 0.989 | 0.962 | 1.052 | 0.962 |
| | Llama 70B | 0.992 | 0.995 | 0.992 | 0.995 | 0.999 | 0.995 | 0.995 | 0.995 | 0.995 | 0.993 | 0.994 | 0.993 | 1.006 | 0.997 | 0.997 | 0.993 | 1.001 | 0.993 |
| | Qwen 32B | 1.001 | 0.998 | 0.998 | 0.990 | 1.000 | 0.990 | 1.004 | 1.002 | 1.002 | 0.992 | 0.996 | 0.992 | 1.003 | 0.999 | 0.999 | 0.993 | 1.000 | 0.993 |
| | Qwen 72B | 1.007 | 1.001 | 1.001 | 1.002 | 1.000 | 1.000 | 1.004 | 0.998 | 0.998 | 1.003 | 0.998 | 0.998 | 1.004 | 1.000 | 1.000 | 1.001 | 1.001 | 1.001 |
| MATH | Llama 8B | 1.091 | 1.024 | 1.024 | 0.838 | 1.017 | 0.838 | 1.244 | 1.067 | 1.067 | 0.857 | 1.040 | 0.857 | 1.012 | 1.030 | 1.012 | 0.813 | 1.208 | 0.813 |
| | Llama 70B | 1.014 | 1.003 | 1.003 | 0.915 | 1.004 | 0.915 | 1.024 | 0.995 | 0.995 | 0.934 | 1.008 | 0.934 | 0.964 | 1.007 | 0.964 | 0.867 | 1.026 | 0.867 |
| | Qwen 32B | 1.041 | 0.999 | 0.999 | 0.967 | 1.000 | 0.967 | 1.111 | 1.008 | 1.008 | 0.941 | 0.989 | 0.941 | 1.033 | 0.996 | 0.996 | 0.907 | 1.017 | 0.907 |
| | Qwen 72B | 1.010 | 0.999 | 0.999 | 0.961 | 1.004 | 0.961 | 1.021 | 1.002 | 1.002 | 0.962 | 0.993 | 0.962 | 1.010 | 0.999 | 0.999 | 0.928 | 1.005 | 0.928 |
| CHAMP | Llama 8B | 0.964 | 1.013 | 0.964 | 0.915 | 0.949 | 0.915 | 0.953 | 0.936 | 0.936 | 0.859 | 1.000 | 0.859 | 0.857 | 0.911 | 0.857 | 1.065 | 1.241 | 1.065 |
| | Llama 70B | 1.048 | 1.016 | 1.016 | 0.957 | 1.039 | 0.957 | 1.005 | 1.000 | 1.000 | 0.977 | 1.008 | 0.977 | 1.057 | 1.016 | 1.016 | 1.008 | 1.023 | 1.008 |
| | Qwen 32B | 1.048 | 0.975 | 0.975 | 1.052 | 0.995 | 0.995 | 1.063 | 1.002 | 1.002 | 1.097 | 1.000 | 1.000 | 1.081 | 0.990 | 0.990 | 1.063 | 0.995 | 0.995 |
| | Qwen 72B | 1.090 | 1.005 | 1.005 | 1.067 | 0.990 | 0.990 | 1.068 | 0.995 | 0.995 | 1.049 | 1.005 | 1.005 | 1.043 | 0.995 | 0.995 | 1.060 | 1.000 | 1.000 |
| HumanEval+ | Llama 8B | 1.114 | 0.943 | 0.943 | 1.104 | 1.020 | 1.020 | 1.113 | 1.001 | 1.001 | 0.970 | 0.934 | 0.934 | 1.132 | 0.991 | 0.991 | 0.973 | 1.039 | 0.973 |
| | Llama 70B | 1.064 | 0.944 | 0.944 | 1.000 | 0.944 | 0.944 | 1.022 | 0.992 | 0.992 | 0.961 | 0.984 | 0.961 | 1.069 | 1.000 | 1.000 | 1.000 | 1.000 | 1.000 |
| | Qwen 32B | 1.015 | 1.000 | 1.000 | 0.985 | 1.000 | 0.985 | 0.973 | 0.990 | 0.973 | 0.963 | 0.977 | 0.963 | 1.000 | 0.985 | 0.985 | 1.000 | 1.000 | 1.000 |
| | Qwen 72B | 1.000 | 1.007 | 1.000 | 0.986 | 1.015 | 0.986 | 0.995 | 0.993 | 0.993 | 0.985 | 0.993 | 0.985 | 1.007 | 1.000 | 1.000 | 0.958 | 1.007 | 0.958 |
| MBPP+ | Llama 8B | 1.078 | 0.942 | 0.942 | 0.854 | 0.850 | 0.850 | 0.979 | 1.002 | 0.979 | 0.939 | 0.903 | 0.903 | 1.102 | 0.995 | 0.995 | 1.000 | 1.044 | 1.000 |
| | Llama 70B | 1.167 | 0.992 | 0.992 | 0.991 | 0.894 | 0.894 | 1.025 | 1.006 | 1.006 | 1.047 | 0.902 | 0.902 | 1.142 | 0.984 | 0.984 | 1.074 | 0.996 | 0.996 |
| | Qwen 32B | 1.007 | 0.986 | 0.986 | 0.902 | 0.909 | 0.902 | 1.001 | 1.005 | 1.001 | 0.943 | 0.937 | 0.937 | 0.996 | 0.975 | 0.975 | 0.983 | 0.986 | 0.983 |
| | Qwen 72B | 1.014 | 0.993 | 0.993 | 0.917 | 0.924 | 0.917 | 1.007 | 1.004 | 1.004 | 0.940 | 0.927 | 0.927 | 0.996 | 0.986 | 0.986 | 0.955 | 0.969 | 0.955 |
| BigCodeBench | Llama 8B | 1.207 | 1.000 | 1.000 | 0.932 | 0.955 | 0.932 | 1.095 | 0.998 | 0.998 | 1.045 | 1.030 | 1.030 | 1.062 | 0.950 | 0.950 | 0.966 | 1.099 | 0.966 |
| | Llama 70B | 1.117 | 0.994 | 0.994 | 1.021 | 0.948 | 0.948 | 1.075 | 1.007 | 1.007 | 1.034 | 0.948 | 0.948 | 1.094 | 0.963 | 0.963 | 1.058 | 0.992 | 0.992 |
| | Qwen 32B | 1.036 | 0.990 | 0.990 | 0.975 | 0.990 | 0.975 | 1.017 | 1.000 | 1.000 | 0.965 | 0.971 | 0.965 | 1.016 | 0.979 | 0.979 | 0.958 | 1.004 | 0.958 |
| | Qwen 72B | 1.017 | 0.998 | 0.998 | 0.959 | 0.962 | 0.959 | 1.009 | 1.001 | 1.001 | 0.969 | 0.989 | 0.969 | 1.000 | 0.979 | 0.979 | 0.995 | 1.013 | 0.995 |
| AlpacaEval | Llama 8B | 0.864 | 1.108 | 0.864 | 0.829 | 1.129 | 0.829 | 1.029 | 1.052 | 1.029 | 0.833 | 1.221 | 0.833 | 0.805 | 1.102 | 0.805 | 0.745 | 1.109 | 0.745 |
| | Llama 70B | 0.804 | 1.043 | 0.804 | 0.806 | 1.020 | 0.806 | 0.983 | 0.988 | 0.983 | 0.775 | 1.003 | 0.775 | 0.784 | 0.999 | 0.784 | 0.755 | 0.997 | 0.755 |
| | Qwen 32B | 0.841 | 1.128 | 0.841 | 0.736 | 0.980 | 0.736 | 0.884 | 0.976 | 0.884 | 0.696 | 1.011 | 0.696 | 0.731 | 0.990 | 0.731 | 0.712 | 0.994 | 0.712 |
| | Qwen 72B | 0.954 | 1.081 | 0.954 | 0.843 | 1.009 | 0.843 | 0.973 | 1.009 | 0.973 | 0.819 | 1.011 | 0.819 | 0.878 | 1.022 | 0.878 | 0.832 | 1.018 | 0.832 |
| IFEval | Llama 8B | 1.005 | 0.929 | 0.929 | 1.066 | 1.028 | 1.028 | 1.041 | 0.988 | 0.988 | 1.036 | 1.023 | 1.023 | 1.024 | 0.970 | 0.970 | 1.025 | 1.051 | 1.025 |
| | Llama 70B | 1.016 | 0.989 | 0.989 | 1.029 | 0.998 | 0.998 | 1.031 | 1.003 | 1.003 | 1.045 | 1.013 | 1.013 | 1.048 | 0.996 | 0.996 | 1.031 | 1.015 | 1.015 |
| | Qwen 32B | 0.988 | 0.993 | 0.988 | 1.007 | 1.025 | 1.007 | 1.005 | 1.002 | 1.002 | 1.018 | 1.016 | 1.016 | 0.993 | 0.993 | 0.993 | 0.998 | 1.028 | 0.998 |
| | Qwen 72B | 1.002 | 0.993 | 0.993 | 1.006 | 1.020 | 1.006 | 0.996 | 0.992 | 0.992 | 0.996 | 1.011 | 0.996 | 1.007 | 1.007 | 1.007 | 0.991 | 1.009 | 0.991 |

Table 15: The three improvement ratios for the (1, 9)-critique-based refinement setup, computed from raw data presented in Tab. 10.

