# OpenReview forum: "Evaluating Judges as Evaluators: The JETTS Benchmark of LLM-as-Judges as Test-Time Scaling Evaluators"
_ICML.cc/2025/Conference — ICML 2025 poster_

### Official Review · Reviewer_Mn8F · 2025-03-13

**Overall Recommendation:** 2

**Summary:**

This paper studies LLM judges as evaluators for test-time scaling and introduces a new benchmark JETTS (Judge Evaluation for Test-Time Scaling). The benchmark assesses different models across three tasks: 1) Response reranking: ) Response reranking, where judges select the best from multiple candidate responses; 2) Step-level beam search, where judges evaluate and rank partial responses during generation; and 3) Critique-based refinement, where judges provide feedback for response improvement. Key findings demonstrate that while existing LLM judges show promise in some test-time scaling scenarios, they have significant limitations, especially in domains requiring complex reasoning.

**Claims And Evidence:**

The claims made in the paper are generally supported. However, several areas would benefit from stronger or more conclusive evidence: 1) In the critique-based refinement findings, the authors demonstrate that refinements rarely surpass both reranking and greedy baselines, but their explanation that critiques are "not actionable enough" lacks sufficient support. A qualitative analysis of critique content with examples would strengthen this claim by illustrating specifically why generators struggle to utilize the feedback effectively. 2) While the judge-to-generator size ratio findings present coefficients (0.19 for math, 0.06 for instruction following, 0.00 for code), it's difficult to determine if these differences are statistically significant. This is particularly important when making claims about domain-specific patterns.

**Essential References Not Discussed:**

The paper overlooks prior work on critique-based refinement. Most notably, it fails to cite CriticBench (Lin et al., 2024) and CriticEval (Lan et al., 2024), which directly evaluate LLMs' abilities to generate and utilize critiques.

**Experimental Designs Or Analyses:**

I analyzed several experimental designs in the JETTS paper. The normalized helpfulness metric and task diversity framework both appear sound. However, I identified several validity issues: 1) The random tie-breaking method used for single-rating protocols introduces unquantified variability that affects result reliability; 2) The critique quality analysis lacks a systematic methodology—the paper claims critiques aren't actionable enough but provides no content analysis to support this conclusion.

**Methods And Evaluation Criteria:**

This paper proposes a new benchmark so no new methods are introduced. The evaluation criteria is comprehensive covering various tasks, datasets, and metrics. But I do feel it is quite overwhelming and important findings are not highlighted. The comparison between Likert and Additive rating protocols, for example, appears to be included primarily for completeness rather than yielding substantive insights. Such peripheral comparisons would be better placed in an appendix to maintain focus on the more significant findings. Additionally, the benchmark's exclusive focus on open-source models represents a limitation.

**Other Comments Or Suggestions:**

The font size on the images is too small to read clearly.

**Other Strengths And Weaknesses:**

Please see my comments in the sections above.

**Questions For Authors:**

Please see my comments in the sections above.

**Relation To Broader Scientific Literature:**

The JETTS benchmark connects two popular research areas: LLM-as-a-judge and test-time scaling. The benchmark reveals significant limitations: LLM judges often fail to improve generator outputs in beam search and refinement tasks, particularly for code generation, and struggle when evaluating larger generator models. These findings challenge the optimistic assumptions in previous research about using LLMs as reliable judges for test-time scaling [1, 2].

[1] Zheng L, Chiang W L, Sheng Y, et al. Judging llm-as-a-judge with mt-bench and chatbot arena[J]. Advances in Neural Information Processing Systems, 2023, 36: 46595-46623.

[2] Snell C, Lee J, Xu K, et al. Scaling llm test-time compute optimally can be more effective than scaling model parameters[J]. arXiv preprint arXiv:2408.03314, 2024.

**Theoretical Claims:**

The paper makes no theoretical claims.

---

> ### Author Rebuttal · Authors · 2025-03-28
>
> We thank reviewer Mn8F for the constructive review and are delighted that they found our evaluation criteria comprehensive. We respond point by point below.
>
> > The random tie-breaking method used for single-rating protocols introduces unquantified variability that affects result reliability;
>
> We believe this is a misunderstanding: We did not employ random tie-breaking in the single-rating protocol precisely because of the prevalence of tied highest-scores. As explained in Sec. 3.2 (Line 129 right), we report the min, average, and max performances from tied responses, allowing us to account for the possible *range* of performance, as plotted in Fig. 6.
>
> > The paper overlooks prior work on critique-based refinement … CriticBench (Lin et al., 2024) and CriticEval (Lan et al., 2024) …
>
> Thank you for pointing us to these works. We will include these works and a longer discussion of critique-related works. Both works use a single round of refinement, while in JETTS, the judge and the generator jointly decide on the number of rounds that the refinement is carried out (including no refinement at all). Despite the difference, all works arrive at similar conclusions: models struggle to improve the response from judge critiques.
>
> For CriticBench [1], the first two “Correction” columns in Table 1 of Page 5 show that only GPT-4 can significantly improve model responses, and most other models generate worse responses than the original ones, as indicated by the red background colors.
>
> For CriticEval [2], as shown in Table 5 of Page 7, the quality of refined responses (the CR metric) using judge-generated critiques is much lower than those using human-annotated feedbacks. Furthermore, as the authors did not share the original model performance (to the best of our knowledge), it is unclear whether the refined responses are actually better than the original ones.
>
> > The critique quality analysis lacks a systematic methodology…
>
> We agree that additional qualitative analysis would be beneficial. We include a case study for **Reviewer i886**, and point the reviewer there due to space limitations.
>
> >2) While the judge-to-generator size ratio findings present coefficients … it's difficult to determine if these differences are statistically significant.
>
> For the regression analysis in Fig. 4, we have the following p-values for the slope and intercept.
>
> | | |p-value|
> |-:|-:|-:|
> | Math|Slope|**9.3e-10**|
> | | Intercept (at size-ratio=0.1)|**1.6e-3**|
> | Code|Slope|0.93|
> | | Intercept (at size-ratio=0.1)|**0.038**|
> | Instruction Following | Slope|0.26|
> | | Intercept (at size-ratio=0.1)|0.064|
> |
>
> For math, both the slope and the (negative) intercept are statistically significant, suggesting both that a large size ratio helps with performance, and the very small ratios hurt. For code, the slope is not significant but the (negative) intercept is, suggesting that all size ratios lead to negative helpfulness. For instruction following, while both slope and the intercept are slightly positive, neither is statistically significant. Thus, claiming a positive effect helpfulness would need more data to support.
>
> We present a similar analysis for the results in Fig. 6 in our response to reviewer gykS, and will include such analyses for all results in the final version.
>
> > The comparison between Likert and Additive rating protocols … would be better placed in an appendix…
>
> Thank you for the suggestion. We will make changes in the final version. Given new results of large judge beam search (see our response to reviewer gykS) and GPT-4o-as-judge (see below), we will also holistically assess the significance of each result and re-organize the main body and appendix as necessary.
>
> > Additionally, the benchmark's exclusive focus on open-source models represents a limitation.
>
> While JETTS focuses on benchmarking specialized LLM judge models, we started experiments with GPT-4o as the judge, using SFRJudge prompts (Fig. 14-15 on Page 15-16),  Llama-3.1-8B-Instruct as the generator, and report normalized helpfulness in reranking and relative improvement over greedy in refinement. We present preliminary results below and will update our paper with full results when experiments conclude. (Skywork-70B cannot generate critiques and hence cannot be used for refinement.)
>
> | Judge | Reranking: MATH | Reranking: BigCodeBench | Reranking: AlpacaEval | Refinement: MATH | Refinement: BigCodeBench | Refinement: AlpacaEval |
> |--|--|--|--|--|--|--|
> | GPT-4o |0.174|0.300|0.359|0.98|1.07|1.10 |
> | SFRJudge-70B|0.418|0.174|0.478|1.12|1.10|1.11 |
> | Skywork-70B|0.185|0.219|0.381|N/A|N/A|N/A |
> |
>
> Except for BigCodeBench Reranking, GPT-4o consistently lags behind SFRJudge-70B and Skywork-70B. This suggests that general-purpose high-performance LLMs also struggle with such fine-grained judging tasks, making JETTS a valuable resource in assessing judging capability progress of future LLMs.
>
> [1] ​​https://arxiv.org/pdf/2402.14809
>
> [2] https://arxiv.org/pdf/2402.13764

---

### Official Review · Reviewer_i886 · 2025-03-14

**Overall Recommendation:** 4

**Summary:**

This paper introduces a benchmark designed to assess the feasibility of using large language model (LLM) judges as evaluators in test-time scaling scenarios. The study compares LLM-judges to traditional reward models (RMs) and process reward models (PRMs) in three key tasks: Response Reranking, Step-Level Beam Search, Critique-Based Refinement.

**Claims And Evidence:**

I think most of the claims are well supported by the evidence.

**Essential References Not Discussed:**

I think most of the related works are well cited.

**Experimental Designs Or Analyses:**

The experimental design is mostly sound, particularly in its use of diverse benchmarks, structured evaluation tasks, and efficiency trade-off analyses.

**Methods And Evaluation Criteria:**

The proposed methods and evaluation criteria in JETTS are generally well-designed for assessing LLM-judges as test-time evaluators.

**Other Comments Or Suggestions:**

N/A

**Other Strengths And Weaknesses:**

N/A

**Questions For Authors:**

What are the possible reasons for the Critique-Based Refinement Task being largely ineffective, despite the success of self-reflection and similar methods in other tasks? If this is the case, what potential solutions could improve its effectiveness?

**Relation To Broader Scientific Literature:**

I think this paper is well designed and interesting overall. It has good contribution that ties test-time scaling settings and llm-as-judge.

**Theoretical Claims:**

N/A

---

> ### Author Rebuttal · Authors · 2025-03-28
>
> We thank the reviewer for their considerate review, and are happy that you found our paper well designed and interesting.
>
> > What are the possible reasons for the Critique-Based Refinement Task being largely ineffective, despite the success of self-reflection and similar methods…? …what potential solutions could improve its effectiveness
>
> We suspect that the lack of success is because judge critiques **lack actionability**. Generally, this means that LLM-as-judge critiques tend to focus on surface-level details (e.g., formatting) rather than correctness.
>
> Recent critique benchmarks had similar findings: Models struggle to improve their performance using critiques generated from external critic models. In particular, two papers suggested by Reviewer Mn8F support this claim: CriticBench [1] and CriticEval [2] both highlight that critiques hold some promise, but in general, only critiques from extremely powerful models, like GPT-4, lead to performance gains. Our work further shows that this holds also for multi-round critique-based refinement, whereas previous work focused only on one round.
>
> Furthermore, even for self-reflection and self-correction without an external evaluator, the evidence of its utility has been mixed, with papers finding that LLMs can’t self-correct reasoning [3], or small LLMs need strong verifiers to do so [4]; See Sec 4.2 of [5] for a comprehensive review. Thus, we believe that future work is needed to further identify the fundamental mechanism and reason for reported success, e.g., to resolve the paradox posed by [3]: If an LLM possesses the ability to self-correct, why doesn’t it simply offer the correct answer in its initial attempt?
>
> To remedy this, we believe that judge training should place a higher emphasis on critique quality, likely borrowing ideas from process supervision or critique-generation models, e.g., using human annotated critiques [6] or RL-training from verifiable feedback [7]. We will update the paper with these discussions.
>
> [1] ​​https://arxiv.org/pdf/2402.14809
>
> [2] https://arxiv.org/pdf/2402.13764
>
> [3] https://arxiv.org/pdf/2310.01798
>
> [4] https://arxiv.org/pdf/2404.17140
>
> [5] https://arxiv.org/abs/2308.03188
>
> [6] https://arxiv.org/abs/2407.00215
>
> [7] https://arxiv.org/abs/2502.03492
>
> ## Case study
> We manually inspected over 100 judge critiques. Overwhelmingly, we found that judges fail in two ways: (1) Failure to catch mistakes by critiquing only stylistic elements of the response and (2) Overemphasis on stylistic details while ignoring a correct response. In both cases, **judge critiques lack actionability**: Judges focus on superfluous details. In the former, the judge fails to guide the generator to a correct response, while in the latter, critiques needlessly prolong the refinement process without substantive improvements.
>
> Below, we present two representative examples of both scenarios. We use MATH, as each response can quickly be verified using the final answer. Due to space limitations, we omit the model’s intermediate reasoning steps and parts of the judge critique with [...]. We will dedicate an entire appendix section in our updated paper with complete examples.
>
> ### Example 1
> Incorrect response, but the judge focuses on surface level details rather than catching the error. As a result, the generator does not revise further.
>
> Ground-truth solution: 19
>
> Generator: Qwen-72B
> ```
> ## Step 1: Identify the sizes of squares that can contain the black center square [...]
> Therefore, the final answer is: $\boxed{14}$. I hope it is correct.
> ```
> Judge: SFRJudge-70B
> ```
> The response precisely executes the instruction by following the required format for complex problems and providing a clear step-by-step solution. The explanation and calculations are concise and easy to follow. The response also accurately counts the number of squares of each size that contain the center square and correctly sums them up to obtain the final answer. [...] Score: 5
> ```
>
> ### Example 2
> Correct response, but the judge focuses on minor details. As a result, the generator continuously revises, but retains the correct answer in each revision. In this case, we verify that the model’s reasoning steps are correct.
>
> Ground-truth solution: -7
>
> Generator: Llama-8B
> ```
> ## Step 1:  We are given a polynomial equation $x^5 + 7x^4 - 2 = 0$ [...]
> Therefore, the final answer is: $\boxed{-7}$.
> ```
> Judge: SFRJudge-70B
> ```
> The response does not precisely execute the instruction. The instruction asks to provide a concise solution with minimal explanation for simple problems (2 steps or fewer) or use a step-by-step format for complex problems (3 steps or more). However, the given problem is complex and requires the step-by-step format, but the response does not provide brief explanations and calculations [...] Additionally, the response does not provide a clear and detailed explanation of how the expression is simplified to obtain the final value. Score: 2
> ```

---

> > ### Comment · Reviewer_i886 · 2025-04-01
> >
> > Thank you very much for the clarifications. I think the discussion on the Critique-Based Refinement Task could further strengthen the paper. Please make sure to include these to the final manuscript if accepted.

---

> > > ### Author Response · Authors · 2025-04-02
> > >
> > > We will ensure the critique-based refinement discussion is included in our final paper. Thank you for your constructive feedback!

---

### Official Review · Reviewer_Bcf6 · 2025-03-18

**Overall Recommendation:** 4

**Summary:**

The authors propose the JETTS Benchmark for evaluating LLM-as-Judge evaluators for test-time scaling where the judges are used to improve the final output from the generator.  The benchmark covers Best-of-N reranking, (2) step-level beam search, and (3) critique-based refinement across the math reasoning, code generation, and instruction following domains.

**Claims And Evidence:**

Yes. The paper provides citations where needed and the claims stated in the experimental results are backed by the evidence in the benchmark results.

**Essential References Not Discussed:**

Not that I am aware of.

**Experimental Designs Or Analyses:**

Yes.  The benchmark is run using 6 different generator models and 6 different judge models using 8 different datasets.  These are all split across the 3 tasks of math reasoning, code generation and instruction following.  The analysis is very detailed and clearly presented with key take-aways marked in bold.

**Methods And Evaluation Criteria:**

Yes, the approaches used make sense and are clearly described.

**Other Comments Or Suggestions:**

I would suggest adding a reasoning for why the 3 domains were chosen as opposed to others.  There is a section on task and dataset selection and some detail on model selection but no information on domain selection.

**Other Strengths And Weaknesses:**

The paper is well written and motivated and the contributions are clearly described.  I do feel that the 3 domains selected limit the applicability of the benchmark to more common use cases such as Q&A, chat and summarization and would like to see those added in a future form of the benchmark.  I would also like to see a general evaluation of the LLM judges as simple evaluators in order to better understand the impact of using them for the 3 tasks presented.

**Questions For Authors:**

1) Why were the 3 domains selected as opposed to others?

2) The focus of this benchmark is on test-time scaling, but does it make sense to add general evaluation of the judges as evaluators to better understand their performance on the 3 tasks?  For instance, a weak evaluator may also be weak on the tasks but some models that are in general weak evaluators may still be useful for the tasks.

3). Similar to the previous question, there should be a consideration on task performance vs latency and memory as these are key considerations for deploying a model for the three tasks evaluated.  Memory impacts resource constraints while latency impacts the usability of even a great model, according to this benchmark, as a judge.

**Relation To Broader Scientific Literature:**

LLMs are being used as evaluators increasingly often in recent work and they provide key benefits in scaling and control.  This has led to them also being used to improve generated output at inference time as a form of reflective selection and refinement.  The benchmark presented in the paper can help researchers identify the strengths and weaknesses of different models for this purpose.

**Theoretical Claims:**

Yes, although there is not much in theoretical claims as this is a benchmark paper.

---

> ### Author Rebuttal · Authors · 2025-03-28
>
> We thank Reviewer Bcf6 for their thoughtful review and are grateful that you found our work well-motivated.
>
> > Why were the 3 domains selected…
>
> This is an excellent question. We will revise our paper to motivate our choice of domains more concretely:
>
> *Instruction following (IF):* Much recent work in judge benchmarking focuses on IF as a proxy for **chat quality** (e.g., [1,2). As such, we include IF as it is best aligned with what judges excel at. This fact is reflected in our results: Across the board, judges performed the best on IF (Fig. 4).
>
> *Math:* Math has exploded in popularity as a domain to measure progress in **LLM reasoning**. Many existing works focus on scaling inference-time compute for math (e.g., [3,4]), using benchmarks like MATH and GSM8K. Thus, we found it crucial to evaluate the judges for math.
>
> *Code:* We identified code as a challenging domain, with many recent methods (including Alphacode [5] and Reflexion [6]),  using inference-time scaling (e.g., [7, 8]) and trained evaluators (e.g., [9]). These initial works suggest that code is an emerging domain in need of strong test-time evaluators. Moreover, the line-by-line nature of code amenable to beam-search, code reranking has been the focus of small-scale judge experiments in prior work [10], and the coding domain provides a more formal reasoning language for LLMs.
>
> > …does it make sense to add general evaluation of the judges as evaluators to better understand their performance on the 3 tasks?...
>
> Thank you for suggesting to contexualize JETTS performance with existing benchmarks. We compare normalized helpfulness on JETTS reranking (RR) and beam search (BS) against accuracy on RewardBench [2] and AutoJ’s EvalP test-set [11]. The former assesses reward modeling ability, while the latter assesses chat-specific evaluation. We will update our paper with a complete figure and present a subset of results below.
>
>
> |Model| RewardBench (accuracy)|EvalP (accuracy)|JETTS RR|JETTS BS|
> |-:|-:|-:|-:|-:|
> |Prometheus-7B |72.0|56.03|-0.098|-0.102|
> |Prometheus 8x7B |74.5|58.69|-0.077|-0.091|
> |SFRJudge 8B |88.7|60.34|0.024|-0.006|
> |Skywork-Critic 8B |89.0| 56.39|0.040|0.044|
> |SFRJudge 70B |92.7| 63.51|0.177|0.129|
> |Skywork-Critic 70B |93.3|57.26|0.172 |0.126|
> |
>
> The judge performance across benchmarks are generally correlated. However, variation in performance in JETTS is much larger than that in RewardBench or EvalP. For example, in RewardBench, the gap between the 8B and 70B Skywork models is 4.3% accuracy (5% relative improvement from 8B to 70B). On JETTS RR, the gap is 0.132 normalized helpfulness, or a 330% relative improvement.
>
> We believe JETTS more accurately reflects the difference in “fundamental judging ability” between small/large judges: Based on RewardBench, the practical choice is to use an 8B judge rather than a 70B judge for reranking/beam-search (4% acc. drop vs for 9x fewer params). However, JETTS, which realistically mimics inference-time scaling tasks, advises the opposite: the choice of 70B judge yields far more gains than the 8B judge.
>
> We found this discussion to be rich, and will update our paper accordingly.
>
> > …there should be a consideration on task performance vs latency and memory as these are key considerations…
>
> We agree that latency and memory are important metrics. Previous works quantify the test time scaling with respect to a compute budget (e.g., Figure 3 of [4]), but use scalar reward models that make the budget easy to quantify (i.e., reward score is only a function of input size). By comparison, LLM judge models can generate critiques/CoT reasoning, making it non-trivial to equalize the compute quantity. Instead, we equalize the experiment setup (e.g., number of responses to rerank or beam width) and leave “compute-optimal” judging to future work.
>
> The reranking strategy, however, does show a performance-efficiency trade-off (Line 247 left). The O(n^2) pairwise round-robin delivers larger gains than the O(n) single-instance rating but requires more time: 23.58 vs. 5.64 seconds/sample for reranking Llama-8B’s BigCodeBench responses using GPT-4o as a judge in our new experiment for Reviewer Mn8F. This difference is more significant for beam search, where each beam search step requires a reranking step. We will update the paper to highlight this trade-off, and additionally include statistics about GPU VRAM needed to run each judge.
>
> [1] https://arxiv.org/abs/2310.07641
>
> [2] https://arxiv.org/abs/2403.13787
>
> [3] https://arxiv.org/abs/2408.03314
>
> [4] https://arxiv.org/abs/2502.06703
>
> [5] https://www.science.org/stoken/author-tokens/ST-905/full
>
> [6] https://arxiv.org/abs/2303.11366
>
> [7] https://arxiv.org/abs/2407.21787
>
> [8] https://arxiv.org/abs/2501.14723
>
> [9] https://arxiv.org/abs/2410.17621
>
> [10] https://arxiv.org/abs/2407.10817
>
> [11] https://arxiv.org/abs/2310.05470

---

> > ### Comment · Reviewer_Bcf6 · 2025-04-08
> >
> > I confirm that I have read the author response and my questions have been answered. I will update my score.

---

> > > ### Author Response · Authors · 2025-04-08
> > >
> > > Thank you for reading our response and we are glad to have addressed your questions. We look forward to incorporating these discussions in the final version of the paper.

---

### Official Review · Reviewer_gykS · 2025-03-24

**Overall Recommendation:** 3

**Summary:**

This paper proposes a benchmark called JETTS (Judge Evaluation for Test-Time Scaling) to evaluate the performance of LLM-as-judges in test-time scaling scenarios. The benchmark consists of three tasks: response reranking, step-level beam search, and critique-based refinement.
The main findings of the paper are:
1. LLM judges can be helpful in certain domains, such as instruction following, but not in others, like math and code generation.
2. Despite more time-efficient, single-rating evaluation protocol performance results in evaluation that is too lenient. Judges often rate a significant fraction of the N responses a top score.
3. Current chain-of-thought reasoning generated by LLM judges is insufficient for self-improvement.

The main contributions by the paper are:
1. The JETTS benchmark, which provides a systematic evaluation framework for LLM judges in test-time scaling scenarios.
2. The comparision of pairwise and pointwise protocols, and the analysis of their trade-offs.
3. The investigation of the effectiveness of chain-of-thought reasoning in LLM judges and its limitations.
Overall, the paper highlights the challenges and opportunities in using LLM judges for test-time scaling and provides a foundation for future research in this area.

**Claims And Evidence:**

The submission presents several claims about the performance and limitations of LLM-as-judges in test-time scaling scenarios. While the paper provides some evidence to support these claims, there are areas where the evidence is not clear or convincing due to lack of in-depth experimental results or improper exeprimental setup.

Specifically, on Line 218, most judges are fine-tuned using a fixed prompt template, but in this paper's setup, a single prompt template is used for Critique-Based Refinement experiments. It would be beneficial to explain why this template was chosen and what effect using different templates might have.

Furthermore, while many numbers are included in the results due to the involvement of multiple generators and judges, it is difficult to determine whether the observed trends or patterns are statistically significant. For example, Figure 6 does not appear to show any significant differences between the various judges, making it less informative.

Minor Suggestion: Adding clear y-axis labels to each plot would improve the overall clarity of the figures.

**Essential References Not Discussed:**

N/A

**Experimental Designs Or Analyses:**

Upon review, most experimental designs and analyses appear to be sound.

**Methods And Evaluation Criteria:**

The proposed methods/metrics overall look intuitive and reasonable.

**Other Comments Or Suggestions:**

## update after rebuttal
The rebuttal has addressed my questions. On the other hand, I agree with some of points from reviewer Mn8F. Thus, I will maintain my score.

**Other Strengths And Weaknesses:**

N/A

**Questions For Authors:**

N/A

**Relation To Broader Scientific Literature:**

Yes. This paper points out the limitations of LLM-as-a-judge models in the test-time scaling setting. Important findings:
1. Although unique to LLM-judges, their natural language critiques are currently ineffective in guiding the generator towards better responses.
2. LLM-judges lag significantly behind the small QPRM in the task of step-level beam search.

**Theoretical Claims:**

N/A

---

> ### Author Rebuttal · Authors · 2025-03-28
>
> We thank Reviewer gykS for their thoughtful review. In particular, we are happy that you found our metrics intuitive and our experimental setup sound. We respond point-by-point to questions and comments below.
>
> > Specifically, on Line 218, most judges are fine-tuned using a fixed prompt template, but in this paper's setup, a single prompt template is used for Critique-Based Refinement experiments. It would be beneficial to explain why this template was chosen and what effect using different templates might have.
>
> We want to clarify our setup. Each judge model is asked to produce a critique and judgment **using its corresponding prompt template. That is, the judge prompt template is not fixed across all judges**. This means that each judge’s instructions and output format follow ones that are used to train the judge.
>
> However, we use a fixed prompt template to prompt the generator model, which for the critique experiments are all general-purpose instruction-tuned LLMs. This fixed prompt, shown in Figure 17, takes in the judge’s critique and score (which we parse out separately from the judge response), the previous response, and the original user query, and tasks the generator (i.e., instruct model) to refine its answer. Upon review of Section 3.4, we realize that we did not make this explicitly clear, and will update the final paper to clarify. Thanks!
>
> > Furthermore, while many numbers are included in the results due to the involvement of multiple generators and judges, it is difficult to determine whether the observed trends or patterns are statistically significant. For example, Figure 6 does not appear to show any significant differences between the various judges, making it less informative.
>
>
> For the single-instance rating reranking protocol shown in Figure 6, we tested whether the min, average and max performances (i.e., the three ticks for each model, task and likert/additive prompt combination) are statistically significantly different from 0 (using a one-sample t-test with p-value threshold of 0.05). Not surprisingly, both the min and max are statistically significantly different from 0 in all cases. However,the average performances are significantly different from 0 for only a handful of math and code cases, as indicated by an “x” in the table below. *Quite concerningly, in all such cases, the average performance is negative, indicating that we have strong evidence that they perform worse than the simple greedy baseline, suggesting the unreliability of the single-rating method.*
>
> |               |         |         |        |         |           |         |
> |--------------:|:-------:|:-------:|:------:|:-------:|:---------:|:-------:|
> |               | Prom 7B | SFR 8B  | Thm 8B | SFR 12B | Prom 8x7B | SFR 70B |
> | Math Likert   | x       |         |        |         | x         |         |
> | Math Additive | x       |         |        |         |           |         |
> | Code Likert   | x       | x       | x      |         | x         |         |
> | Code Additive | x       | x       | x      | x       |           |         |
> | Instruction Following Likert   |         |         |        |         |     |         |
> | Instruction Following Additive |         |         |        |         |     |         |
> |
>
> Furthermore, we present statistical analysis for the linear regression in Figure 4 in our response to reviewer Mn8F, and will include these analyses for all results in the final version.
>
> > Minor Suggestion: Adding clear y-axis labels to each plot would improve the overall clarity of the figures.
>
> We agree with the reviewer. As we cannot upload an updated paper version, we will update our figures for our final paper.
>
> > LLM-judges lag significantly behind the small QPRM in the task of step-level beam search.
>
> We are excited to share some new results. Since submission, we obtained access to additional compute resources which were used to evaluate the **large judge models** on beam search (the “C!” entries in the Figure 1 result summary). Here, we provide a summary of our results, with our final paper to be updated with more comprehensive analysis.
>
> | Model                | Performance |
> |---------------------:|------------:|
> | Prometheus-7B        | -0.102      |
> | SFRJudge 8B          | -0.006      |
> | Skywork-Critic 8B    | 0.044       |
> | OffsetBias 8B        | 0.005       |
> | Themis 8B            | -0.026      |
> | SFRJudge 12B         | 0.040       |
> | **Prometheus 8x7B**      | -0.091      |
> | **SFRJudge 70B**         | 0.129       |
> | **Skywork-Critic 70B**   | 0.126       |
> | **Self-taught-eval-70B** | 0.074  |
> | Qwen PRM 7B              | 0.178  |
> | Random               | -0.141 |
> | |
>
> As we can see, all large judges (bolded), except for Prometheus 8x7B, perform much better than smaller ones, with SFRJudge and Skywork-Critic 70B being the best. However, they still lag behind the much smaller 7B Qwen PRM, suggesting that finer-grained step-level judging has much room for improvement.

---

### Decision · Program_Chairs · 2025-05-01

**Decision:**

Accept (poster)

**Comment:**

**Summary:** The paper proposes ​​Judge Evaluation for Test-Time Scaling (JETTS) benchmark, which aims to provide an evaluation of LLM-judges for test-time scaling. The tasks evaluated are response reranking, step-level beam search and critique based response refinement tasks. They also provide insights on how LLM-judges compare with respect to reward models and process reward models and discuss performance across different domains. They also show that their explanations are ineffective for guiding generators for self-improving answers.

**Strengths:**
- LLM-judges are being commonly used and their connection with test-time scaling will be relevant to the ICML community.
- Claims are well supported by rigorous experiments. Experimental designs vetted.
- The paper is well written and organized.

**Weaknesses:**
- The evaluation domains of instruction following, math and code might be limited in scope.

**Suggestions:**
- Showcase the limitations of LLM-judges for critic based refinement through qualitative analysis. This is already done during the rebuttal and reviewers recommend including in the final manuscript.

**Recommendation:**
- Three out of four reviewers recommended acceptance of the paper with scores of 3, 4 and 4. The remaining reviewer recommended a weak rejection (with a score of 2).
- I believe that the paper is investigating an important area with clear research questions and well designed experiments to investigate those questions, and will be of interest to the ICML community. Therefore, I recommend its acceptance.